# Can neural operators always be continuously discretized?

Takashi Furuya[1,*]     Michael Puthawala[2,*]     Maarten V. de Hoop[3]     Matti Lassas[4]

[1]Shimane University, `takashi.furuya0101@gmail.com`
[2]South Dakota State University, `Michael.Puthawala@sdstate.edu`
[3]Rice University, `mdehoop@rice.edu`
[4]University of Helsinki, `matti.lassas@helsinki.fi`

\* These authors contributed equally to this work

## Abstract

We consider the problem of discretization of neural operators between Hilbert spaces in a general framework including skip connections. We focus on bijective neural operators through the lens of diffeomorphisms in infinite dimensions. Framed using category theory, we give a no-go theorem that shows that diffeomorphisms between Hilbert spaces or Hilbert manifolds may not admit any continuous approximations by diffeomorphisms on finite-dimensional spaces, even if the approximations are nonlinear. The natural way out is the introduction of strongly monotone diffeomorphisms and layerwise strongly monotone neural operators which have continuous approximations by strongly monotone diffeomorphisms on finite-dimensional spaces. For these, one can guarantee discretization invariance, while ensuring that finite-dimensional approximations converge not only as sequences of functions, but that their representations converge in a suitable sense as well. Finally, we show that bilipschitz neural operators may always be written in the form of an alternating composition of strongly monotone neural operators, plus a simple isometry. Thus we realize a rigorous platform for discretization of a generalization of a neural operator. We also show that neural operators of this type may be approximated through the composition of finite-rank residual neural operators, where each block is strongly monotone, and may be inverted locally via iteration. We conclude by providing a quantitative approximation result for the discretization of general bilipschitz neural operators.

## 1  Introduction

Neural operators, first introduced in [30], have become more and more prominent in deep learning on function spaces. As opposed to traditional neural networks that learn maps between finite-dimensional Euclidean spaces, neural operators learn maps between infinite-dimensional function spaces yet may be trained and evaluated on finite-dimensional data through a rigorous notion of discretization. Neural operators are widely used in the field of scientific machine learning [6, 21, 38, 55, 56], among others, principally because of their discretization invariance. In this work, we consider the fundamental limits of this discretization. Throughout, we emphasize the importance of continuity under discretization.

A key ingredient in our analysis is the identification of properties of diffeomorphisms that may be induced by (bijective) neural operators, which are diffeomorphisms themselves. Diffeomorphisms exist in many contexts, for example, in generative models. These involve mapping one probability distribution or measure, $\mu$, over some measurable space to another, $X$, via a push forward, that is, $F_\#\mu(U) = \mu(F^{-1}(U))$ for $U \subset X$. If $\mu$ admits a density $d\mu$ then, in finite dimensions, we may

use the change of variables formula to write $d\rho(x) = d\mu(F^{-1}(x))|JF(x)|^{-1}$, where $JF$ is the Jacobian of $F$. Clearly, $F$ must be a bijection with full-rank Jacobian. In other words, $F$ must be a diffeomorphism onto its range. This established diffeomorphisms as natural objects of interest in finite-dimensional machine learning, and helps account for their wide use [20, 25, 31, 44, 54]. In this work, we consider the extension of these efforts from finite to infinite dimensions implemented via neural operators. Although there is no analogue of the change of variables formula in infinite dimensions, we argue that it is, nonetheless, natural to consider the role of diffeomorphisms, and how they may be approximated via diffeomorphisms on finite-dimensional spaces.

The question of when operations between Hilbert spaces may be discretized continuously may be understood through an analogy to computer vision. Consider the task of learning a map from one image space to another, for example, a style transfer problem [16], where the mapping learned does not depend much on the resolution of the images provided. It is natural to think of the map as being defined between (infinite-resolution) *continuum* images, and then its application to images of a specific resolution. In this analogy, $X$ is a function space (over images $m : [0,1]^2 \to \mathbb{R}$ [29]) that is approximated with a finite-dimensional space $\mathbb{R}^d$ and the transformation $F : X \to X$ is approximated by a map $f : \mathbb{R}^d \to \mathbb{R}^d$, where each $f$ acts on images of a particular resolution. An explicit transformation formula can be obtained when $f$ is a diffeomorphism and has a smooth inverse.

We introduce a framework based on a generalized notion of neural operator layers including a skip connection and their restrictions to balls rather than compact sets. With bijective neural operators in mind, we give a perspective based on diffeomorphisms in infinite dimensions between Hilbert manifolds. We give a no-go theorem, framed with category theory, that shows that diffeomorphisms between Hilbert spaces may not admit any continuous approximations by diffeomorphisms on finite-dimensional spaces, *even* if the underlying discretization is nonlinear. In this framing the discretization operation is modeled as a functor from the category of Hilbert spaces and $C^1$-diffeomorphisms on them to their finite-dimensional approximations. A natural way to mitigate the no-go theorem is described by the introduction of strongly monotone diffeomorphisms and layerwise strongly monotone neural operators. We prove that all strongly monotone neural operator layers admit continuous approximations by strongly monotone diffeomorphsisms on finite-dimensional spaces. We then provide various conditions under which a neural operator layer is strongly monotone. Notably, a bilipschitz (and, hence, bijective) neural operator layer can always be represented by a composition of strongly monotone neural operator layers. Hence, such an operator may be continuously discretized. More constructively, any bilipschitz neural operator layer can be approximated by residual finite-rank neural operators, each of which are strongly monotone, plus a simple isometry. Moreover, these finite-rank residual neural operators are (locally) bijective and invertible, and their inverses are limits of compositions of finite-rank neural operators. Our framework may be used "out of the box" to prove quantitative approximation results for discretization of neural operators.

## 1.1 Related work

Neural operators were first introduced in [30]. Alternative designs for mappings between function spaces are the DeepONet [34, 40], and the PCA-Net [7, 12]. In spite of the multitudinous applications of neural operators, the theory of the natural class of injective or bijective neural operators is comparatively underdeveloped; see, for example [2, 14].

Our work is concerned with the of discretization of neural operators through the lens of diffeomorphisms. For recent important work on analyzing the effect of discretization error of Fourier Neural Operators (FNOs) arising from aliasing, see [35]. Our work has connections to infinite-dimensional inference, see e.g. [19], and approximation theory, see e.g. [13] while bridging the gap with the theory of neural operators.

We give a no-go theorem that uses a category theory framing. This contributes to the use of category theory as an emerging tool in the analysis and understanding of neural networks at large. In this sense, we are in league with the recent work generalizing ideas from geometric machine learning using category theory [17].

Discretization obstructions have been encountered in other contexts. Numerical methods that approximate continuous models are known to sometimes fail in surprising ways. A basic example of this is the "locking" phenomenon in the study of the Finite Elements Method (FEM). For example,

linear elements used to model bending of a curved surface or beam lock in such a way that the model exhibits a non-physical stiff response to deformations [4]. Understanding this has been instrumental in developing improved numerical methods, such a high order FEM [52]. Furthermore, in discretized statistical inverse problems [27, 36, 51], the introduction of Besov priors [48, 11] has been found to be essential.

Finally, our work extends prior work (not based on deep learning) in discretization of physical or partial differential equations based forward models in inverse problems. The analogous notion of discretization invariance of solution algorithms of inverse problems was studied in [37, 48, 51] and the lack of it (in imaging methods using Bayesian inversion with $L^1$ priors) in [36, 48]. By considering the neural operator as the physical model, our results state that discretization can be done locally in an appropriate way, together with constructing an inverse.

## 1.2 Our contributions

The key results in this paper comprise the following:

1. We prove a general no-go theorem showing that, under general circumstances, diffeomorphisms between Hilbert spaces may not admit continuous approximation by finite-dimensional diffeomorphisms. In particular, neural operators corresponding to diffeomorphic maps, in general, cannot be approximated by finite-dimensional diffeomorphisms and their associated neural representations.

2. We show that strongly monotone neural operator layers admit continuous approximations by strongly monotone diffeomorphsisms on finite-dimensional spaces.

3. We show that bilipschitz neural operators can be represented in any bounded set as a composition of strongly monotone, diffeomorphic neural operator layers, plus a simple isometry. These can be approximated by finite-rank diffeomorphic neural operators, where each layer is strongly monotone. For these operators we give a quantitative approximation result.

## 2 Definitions and notation

In this section, we give the definitions and notation used throughout the paper. First, we summarize the relevant basic concepts from functional analysis. Then, we introduce generalized neural operators.

### 2.1 Elements of functional analysis

In this work, all Hilbert spaces, $X$, are endowed with their norm topology. We denote by $B_X(r) = B_X(0, r)$ the ball in the space $X$ having the center at zero and radius $r > 0$. We denote by $S(X)$ the set of all finite-dimensional linear subspaces $V \subset X$. The set $S_0(V) \subset S(X)$ is a partially ordered lattice. That is, if $V_1, V_2 \in S_0(X)$ then there is a $V_3 \in S_0(X)$ so that $V_1 \subset V_3$ and $V_2 \subset V_3$. [1] With $Y$ standing for another Hilbert space, we denote by $C^n(X; Y)$ the set of operators, $F \colon X \to Y$, having $n$ continuous (Fréchet) derivatives, and $C^n(X) = C^n(X; X)$.

Next, we define what it means that a nonlinear operator or function $F \colon X \to X$ on an infinite-dimensional Hilbert space, $X$, is approximated by operators or functions on finite-dimensional subspaces $V \subset X$. The key is that as $V$ tends to $X$, the complexity of $F_V$ increases and one may hope that the approximation becomes better. We formalize this in the following definition.

**Definition 1** ($\epsilon_V$ approximators and weak approximators)**.** *(i) Let $r > 0$, $\mathcal{F} \subset C^n(X; X)$ be a family of functions, and $\vec{\varepsilon} = (\varepsilon_V)_{V \in S_0(X)}$ be a sequence such that $\varepsilon_V \to 0$ as $V \to X$. We say that a function*

$$\mathcal{A}_X \colon \ \mathcal{F} \to \bigtimes_{V \in S_0(X)} C(\overline{B_V(0, r)}; V), \quad F \to (F_V)_{V \in S_0(X)}$$

---

[1] Each element of $S_0(X)$ will come to represent a discretization of $X$. The partially ordered lattice condition will come to represent a notion of common refinement of a discretization. This makes it possible to consider "realistic" choices of discretizations. The condition automatically follows for any discretization scheme that has a notion of "common refinement" of two discretizations. Examples include finite-difference schemes, and the finite elements Galerkin discretization that is based on a triangulation of the domain.

*is an $\vec{\varepsilon}$-approximation operation for functions $\mathcal{F}$ in the ball $B_X(0, r)$ taking values in families $\mathcal{F}_V \subset C^1(V; V)$ if $\mathcal{A}_X$ maps a function $F \colon X \to X$, where $F \in \mathcal{F}$, to a sequence of functions $(F_V)_{V \in S_0(X)}$, where $F_V \in \mathcal{F}_V$, such that the following is valid: For all $F \colon X \to X$ satisfying $\|F\|_{C^n(\overline{B_X(0,r)}; X)} \leq M$, we have*

$$\sup_{x \in \overline{B_V(0,r)}} \|F_V(x) - P_V(F(x))\|_X \leq M\varepsilon_V, \tag{1}$$

*where $P_V \colon X \to X$ is the orthogonal projection onto $V$, that is, $\mathrm{Ran}(P_V) = V$.*

*(ii) We say that $\mathcal{A} \colon C^n(X; X) \to \bigtimes_{V \in S_0(X)} C(V; V), \quad F \to (F_V)_{V \in S_0(X)}$ is a weak approximation operation for the family $\mathcal{F} \subset C^n(X; X)$ if for any $F \in \mathcal{F}$ and $r > 0$ it holds that*

$$\lim_{V \to X} \sup_{x \in \overline{B_V(0,r)}} \|F_V(x) - P_V(F(x))\|_X \to 0.$$

Note that the condition (i) is stronger than the condition (ii). An example of an approximation operation for the family $\mathcal{F} = C^n(X)$, that is, an $\vec{\varepsilon}$-approximation operation with all sequences $\vec{\varepsilon} = (\varepsilon_V)_{V \in S_0(X)}$ subject to $\varepsilon_V > 0$, is the linear discretization

$$\mathcal{A}_{\mathrm{lin}}(F) = (F_V)_{V \in S_0(X)}, \quad F_V = P_V \circ (F|_V) \colon V \to V. \tag{2}$$

Nonlinear discretization methods that do not rely on $P_V$ have been used, for example, in the numerical analysis of nonlinear partial differential equations. Here, $X$ becomes an appropriate Sobolev space, and a Galerkin approximation is implemented through finite-dimensional subspaces, $V$, spanned by finite element basis functions. We present an example for the nonlinear equation, $\Delta u(t) - g(u(t)) = \Delta x(t)$ where $g$ is a smooth convex function, when $F \colon x \to u$, in Appendix A.1.

Below, we will study whether a family, $\mathcal{F} \subset \mathrm{Diff}^1(X)$, of $C^1$ diffeomorphisms on $X$ can be approximated by $C^1$ diffeomorphisms, $\mathcal{F}_V \subset \mathrm{Diff}^1(V)$, on finite-dimensional subspaces, $V$. Of course, diffeomorphisms are bijective. Unless stated otherwise, from now on we will omit $C^1$ and implicitly assume that diffeomorphism are $C^1$ diffeomorphisms. We introduce two more notions that will play a key role in the further analysis.

**Definition 2** (Strongly Monotone). *We say that a (nonlinear) operator $F \colon X \to X$ on Hilbert space, $X$, is strongly monotone if there exists a constant $\alpha > 0$ so that*

$$\langle F(x_1) - F(x_2), x_1 - x_2 \rangle_X \geq \alpha \|x_1 - x_2\|_X^2, \quad \text{for all } x_1, x_2 \in X. \tag{3}$$

**Definition 3** (Bilipschitz). *We say that $F$ if bilipschitz there exist constants $c > 0$ and $C < \infty$ so that for all $x_1, x_2 \in X$, $c\|x_1 - x_2\| \leq \|F(x_1) - F(x_2)\| \leq C\|x_1 - x_2\|$.*

## 2.2 A general framework for neural operators

In this paper, we are concerned with the modeling of diffeomorphisms between Hilbert spaces by bijective neural operators. Our working definition of neural operator, which generalizes the traditional notion, is given below. We note the presence of a skip connection, which is essential.

**Definition 4** (Generalized neural operator layer). *For Hilbert space $X$, a layer of a neural operator is a function $F \colon X \to X$ of the form*

$$F(x) = x + T_2 G(T_1 x), \tag{4}$$

*where $T_1 \colon X \to X$ and $T_2 \colon X \to X$ are compact linear operators [2] and $G \colon X \to X$ is a nonlinear operator in $C^1(X)$. A generalized neural operator, $H \colon X \to X$, is given by the composition*

$$H = A_L \circ \sigma \circ F_L \circ A_{L-1} \circ \sigma \circ F_{L-1} \circ \cdots \circ A_1 \circ \sigma \circ F_1, \tag{5}$$

*where each $F_\ell$, $\ell = 1, \ldots, L$ is of the form (4), the $A_\ell \colon X \to X$ are bounded linear operators and $\sigma \colon X \to X$ is a continuous operation (for example, a Nemytskii operator defined by a composition with a suitable activation function in function spaces).*

---

[2]By using mapping properties of monotone operators [5], we can replace this definition by using Hilbert spaces $Y$ and $Z$ that are isometric to $X$, $T_1 \colon X \to Y$ and $T_2 \colon Z \to X$ as compact linear operators, and $G \colon Y \to Z$ as a $C^1$ nonlinear operator.

The generalized neural operators can represent the classical neural operators [30, 33]. For an explicit construction, we refer to Appendix C.1. In the next section, we will study, under what conditions, bounded linear operators $A_\ell$, Nemytskii operators $\sigma$, and neural operator layers $F_\ell$, for which the generalized neural operator consists, can be continuously discretized.

We note that because $G \in C^1(X)$ in Definition 4, it follows that $G \in L^\infty(X)$ and $\text{Lip}_{X \to X}(G) < \infty$. Given a Hilbert space, $X$, a *layer of a strongly monotone neural operator* (respectively, a *layer of a bilipschitz neural operator*,) is a function $F \colon X \to X$ that is strongly monotone (respectively, bilipschitz). Furthermore, a *strongly monotone neural operator* (respectively, a *bilipschitz neural operator*), is a generalized neural operator with strongly monotone (respectively, bilipschitz), layers.

## 3 Category theoretic framework for discretization

"Well-behaved" operators between infinite-dimensional Hilbert spaces may have dramatically different behaviors than corresponding "well-behaved" maps between finite-dimensional Euclidean spaces. This observation applies to discretization and neural operators versus neural networks. In this section we explore this. We first present a no-go theorem, that there are *no* procedures that continuously discretize an isotopy of diffeomorphisms. Next, we introduce strongly monotone neural operator layers, which are strongly monotone diffeomorphisms, and then prove that these allow a continuous approximation functor, that is, continuous approximations by strongly monotone diffeomorphisms on finite dimensional spaces. We finally show that bilipschitz neural operator layers admit a representation via strongly monotone operators and linear maps, allowing for their continuous approximation.

### 3.1 No-go theorem for discretization of diffeomorphisms on Hilbert spaces

In this section, we present our no-go theorem. To formulate the 'impossibility' of something, we must define what is meant by discretization and approximation. Before this, we give an informal statement of the no-go theorem.

**Theorem 1** (No-go Theorem, Informal)**.** *Let $\mathcal{A}$ be an approximation scheme that maps diffeomorphisms $F$ on a Hilbert to a sequence of finite-approximations $F_V$ that are themselves diffeomorphisms. If $F_V$ converges to $F$ as $V \to X$, then $\mathcal{A}$ is not continuous, that is, there are maps $F^{(j)}$ that converge to $F$ as $j \to \infty$, but all $F_V^{(j)}$ are far from $F_V$.*

We want to emphasize that most practical numerical algorithms are continuous so that the output depends (in some suitable sense) continuously on the input. This shows that there are no such numerical schemes that approximate infinite-dimensional diffeomorphisms with finite-dimensional ones. In order to prove our no-go theorem in the most general setting, we phrase it in terms of category theory. Namely, we formulate $\mathcal{A}$ (which will denote the approximation scheme) as a functor from the category of Hilbert spaces and diffeomorphisms thereon, to their finite-rank approximations.

**Definition 5** (Category of Hilbert Space Diffeomorphisms)**.** *We denote by $\mathcal{D}$ the category of Hilbert diffeomorphisms with objects $\mathcal{O}_\mathcal{D}$ that are pairs $(X, F)$ of a Hilbert space $X$ and a (possibly non-linear) $C^1$-diffeomorphism $F \colon X \to X$ and the set of morphisms (or arrows that 'map' objects to other objects) $\mathcal{A}$ that are either*

1. *(induced isomorphisms) Maps $a_\phi$ that are defined for a linear isomorphism $\phi : X_1 \to X_2$ of Hilbert spaces $X_1$ and $X_2$ that maps the objects $(X_1, F_1) \in \mathcal{O}_\mathcal{D}$ to the object $(\phi(X_1), \phi \circ F_1 \circ \phi^{-1}) \in \mathcal{O}_\mathcal{D}$, or*

2. *(induced restrictions) Maps $a_{X_1, X_2}$ that are defined for a Hilbert space $X_1$, its closed subspace $X_2 \subset X_1$, and an object $(X_1, F_1) \in \mathcal{O}_\mathcal{D}$ such that $F_1(X_2) = X_2$. Then $a_{X_1, X_2}$ maps to the object $(X_1, F_1) \in \mathcal{O}_\mathcal{D}$ to the object $(X_2, F_1|_{X_2}) \in \mathcal{O}_\mathcal{D}$.*

**Definition 6** (Category of Approximation Sequences)**.** *We denote by $\mathcal{B}$ the category of approximation sequences, that has objects $\mathcal{O}_\mathcal{B}$ that are of the form $(X, S_0(X), (F_V)_{V \in S_0(X)})$ where $X$ is a Hilbert space,*

$$S_0(X) \subset S(X) = \{V \mid V \subset X \text{ is a finite dimensional linear subspace}\},$$

*are partially ordered lattices, $\bigcup_{V \in S_0(X)} V = X$, and $F_V \colon V \to V$ are $C^1$-diffeomorphisms of spaces $V \in S_0(X)$.*

*The set of morphisms $\mathcal{A}_{\mathcal{B}}$ consists of either*

1. *Maps $A_\phi$ that are defined for a linear isomorphism $\phi : X_1 \to X_2$ of Hilbert spaces $X_1$ and $X_2$, and lattices $S_0(X_1)$ and $S_0(X_2) = \{\phi(V) \mid V \in S_0(X_1)\}$, that maps the objects $(X_1, S(X_1), (F_V)_{V \in S(X_1)})$ to $(X_2, S(X_2), (\phi \circ F_{\phi^{-1}(W)} \circ \phi^{-1})_{W \in S(X_2)})$, or*

2. *Maps $A_{X_1, X_2}$ that are defined for a Hilbert space $X_1$, its closed subspace $X_2 \subset X_1$, and an object $(X_1, S_0(X_1), (F_V)_{V \in S_0(X_1)})$ such that $F(X_2) = X_2$ and $S_0(X_2) = \{V \in S_0(X_1) \mid V \subset X_2\}$ is a partially ordered lattice. Then $A_{X_1, X_2}$ maps the object $(X_1, S_0(X_1), (F_V)_{V \in S_0(X_1)})$ to the object $(X_2, S_0(X_2), (F_V)_{V \in S_0(X_2)})$.*

Next, we define the notion of an approximation or discretization functor. In practice, an approximation functor is an operator which maps a function $F$ in an infinite dimensional space $X$ to a function $F_V$ that operate in finite dimensional subspaces $V$ of $X$ in such a way that functions $F_V$ are close (in a suitable sense) to the function $F$.

**Definition 7** (Approximation Functor). *We define the* approximation functor, *denoted by $\mathcal{A} : \mathcal{D} \to \mathcal{B}$, as the functor that maps each $(X, F) \in \mathcal{O}_{\mathcal{D}}$ to some $(X, S_0(X), (F_V)_{V \in S_0(X)}) \in \mathcal{O}_{\mathcal{B}}$ so that the Hilbert space $X$ stays the same. The approximation functor maps all morphisms $a_\phi$ to $A_\phi$ and morphisms $a_{X_1, X_2}$ to $A_{X_1, X_2}$, and has the following the properties*

(A) *For all $r > 0$ and all $(X, F) \in \mathcal{O}_{\mathcal{D}}$,*

$$\lim_{V \to X} \sup_{x \in \overline{B}_X(0,r) \cap V} \|F_V(x) - F(x)\|_X = 0.$$

*In separable Hilbert spaces this means that when the finite dimensional subspaces $V \subset X$ grow to fill the whole Hilbert space $X$, then the approximations $F_V$ converge uniformly in all bounded subsets to $F$.*

We recall the notation $\lim_{V \to X}$ used above: We consider $(S_0(X), \supset)$ as a partially ordered set and say that real numbers $y_V$ converge to the limit $y$ as $V \to X$, and denote

$$\lim_{V \to X} y_V = y,$$

if for all $\epsilon > 0$ there is $V_0 \in S_0(X)$ such that for $V \in S_0(X)$ satisfying $V \supset V_0$ it holds that $|y_V - y| < \epsilon$.

**Definition 8.** *We say that the approximation functor $\mathcal{A}$ is continuous if the following holds: Let $(X, F), (X, F^{(j)}) \in \mathcal{O}_{\mathcal{D}}$ be such that the Hilbert space $X$ is the same for all these objects and let $(X, S_0(X), (F_V)_{V \in S_0(X)}) = \mathcal{A}(X, F)$ be approximating sequences of $(X, F)$ and $(X, S_0(X), (F_{j,V})_{V \in S_0(X)}) = \mathcal{A}(X, F^{(j)})$ be approximating sequences of $(X, F^{(j)})$. Moreover, assume that $r > 0$ and*

$$\lim_{j \to \infty} \sup_{x \in \overline{B}_X(0,r)} \|F^{(j)}(x) - F(x)\|_X = 0. \tag{6}$$

*Then, for all $V \in S_0(X)$ the approximations $F_V^{(j)}$ of $F^{(j)}$ and $F_V$ of $F$ satisfy*

$$\lim_{j \to \infty} \sup_{x \in V \cap \overline{B}_V(0,r)} \|F_V^{(j)}(x) - F_V(x)\|_X = 0. \tag{7}$$

The theorem below states a negative result, namely that there does not exist continuous approximating functors for diffeomorphisms.

**Theorem 2.** *(No-go theorem for discretization of general diffeomorphisms) There exists no functor $\mathcal{D} \to \mathcal{B}$ that satisfies the property (A) of an approximation functor and is continuous.*

The proof is given in Appendix A.4.1, and is quite involved, but we give an overview of some of the steps here. A generalization of Theorem 2, in the case where the norm topology is replaced by the weak topology, is considered in Appendix D.1.

A key fact is that for finite dimensional diffeomorphisms the space of smooth embeddings consists of two connected components, one orientation preserving and the other orientation reversing. This is

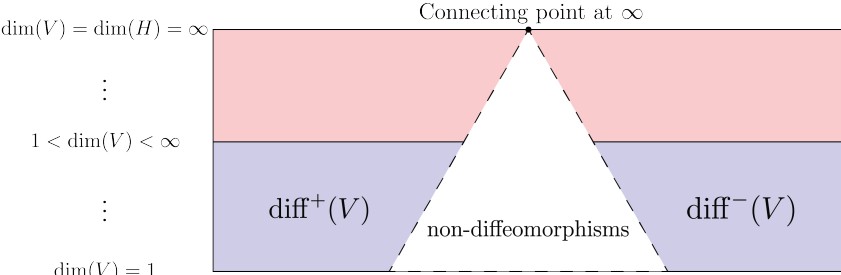

Figure 1: A figure illustrating the proof ideas for Theorem 2. It represents the disconnected components of diffeomorphisms that preserve orientation, notated by $\text{diff}^+$, and reverse orientation, notated, $\text{diff}^-$. The horizontal axis abstractly represents the two disconnected components of $\text{diff}$ for a finite-dimensional vector space $V$. The vertical axis represents the dimension of $V$. Observe how the two components of $\text{diff}$ connect as $\dim(V) \to \infty$, and $V$ becomes a Hilbert space $H$.

not the case in infinite dimensions, see e.g. [32] and [45]. For an illustration of this, see Figure 1. The proof proceeds by contradiction. First, we consider the action of the approximation functor as it operates on an isotopy (path of diffeomorphisms) that connects two diffeomorphisms. The first has only orientation-preserving discretizations, and the second only orientation-reversing discretizations. We then show that the image of the path under the approximation functor yields a disconnected path, as the discretization 'jumps' from the orientation preserving component to the orientation reversing component. This violates continuity. To encode the notions of orientation preserving and orientation reversing that allow for a description of nonlinear discretization theory, we use topological degree theory. This generalizes the familiar notion of orientation that uses the sign of the determinant of the Jacobian matrix.

### 3.2 Strongly monotone diffeomorphisms and their approximation on finite-dimensional subspaces

In this section and the next two we show that, although Theorem 2 precludes continuous approximation of general diffeomorphisms, stronger constraints on the diffeomorphisms allows one to sidestep the topological obstruction. In this section, in summary, we show that the obstruction to continuous approximation vanishes when the diffeomorphisms in question are assumed to be strongly monotone. Key to our positive result is the following technical result that states that the restriction of the domain and codomain of a strongly monotone diffeomorphism always yields another strongly monotone diffeomorphism.

**Lemma 1.** *Let $V \subset X$ be a finite-dimensional subspace of $X$, and let $P_V : X \to X$ be orthogonal projection onto $V$. Let $F : X \to X$ be a strongly monotone $C^1$-diffeomorphism. Then, $P_V F|_V : V \to V$ is strongly monotone, and a $C^1$-diffeomorphism.*

The proof is given in Appendix A.5.1. Lemma 1 implies that the discretization functor $\mathcal{A}_{\text{lin}}$ defined in (2) is a well-defined functor from strongly monotone $C^1$-diffeomorphisms of $X$ to $C^1$-diffeomorphisms of $V$. Note that the discretization functor $\mathcal{A}_{\text{lin}}$ on strongly monotone $C^1$ diffeomorphisms may not be a continuous approximation functor in the strong sense of Definitions 7 and 8, but it is obviously a continuous approximation functor in the weak sense of Definitions 11 and 12. Therefore, we obtain that :

**Proposition 1.** *Let $\mathcal{A}_{\text{lin}}$ be the discretization functor that maps $F$ to $P_V F|_V$ for each finite subspace $V \subset X$. Let $\mathcal{D}_{sm}$ and $\mathcal{B}_{sm}$ be categories where $F : X \to X$ and $F_V : V \to V$ are strongly monotone $C^1$-diffeomorphisms. Then, the functor $\mathcal{A}_{\text{lin}} : \mathcal{D}_{sm} \to \mathcal{B}_{sm}$ satisfies assumption (A') of a weak approximation functor in Definition 11, and is continuous in the weak sense of Definition 12.*

### 3.3 Strongly monotone Nemytskii operators and linear bounded operators and their continuous approximation on finite-dimensional subspaces

By Proposition 1, strongly monotone maps can be continuously discretized in the weak sense. Thus, we concern under what conditions, the maps, of which generalized neural operator consists, can be

strongly monotone. In this subsection, we focus on bounded linear operators and Nemytskii operators (layers of neural operator will be discussed in the next subsection). The following lemma is obviously given by the definition of a strongly monotone map :

**Lemma 2.** *Let $A : X \to X$ be a linear bounded operator and satisfy $\langle Au, u \rangle \geq c_0 \|u\|_X^2$ for some $c_0 > 0$. Then, $A : X \to X$ is strongly monotone.*

Next, assuming that $X = L^2(D; \mathbb{R})$, we define Nemytskii operator by

$$F^\sigma(u) = \sigma \circ u, \tag{8}$$

where $\sigma : \mathbb{R} \to \mathbb{R}$ is continuous. In this case, we can show the following lemma by using [50, Corollary 3.3]:

**Proposition 2.** *Assume that $\sigma$ satisfies that $|\sigma(s)| \leq C_1 |s| + C_2$ and the derivative of $s \to \sigma(s)$ is defined a.e and satisfies $\sigma'(s) \geq \alpha > 0$. Then, $F^\sigma : L^2(D; \mathbb{R}) \to L^2(D; \mathbb{R})$ is strongly monotonous.*

### 3.4 Strongly monotone generalized neural operators and their continuous approximation on finite-dimensional subspaces

As seen in Theorem 3 below, strongly monotone layers of neural operators do not suffer from the same topological obstruction to continuous discretization as general diffeomorphisms. We now give sufficient conditions for the layers of a neural operator to be strongly monotone, and show that these conditions imply that those are diffeomorphisms.

**Lemma 3.** *All strongly monotone layers of neural operators ($F$) defined by* (4) *are diffeomorphisms.*

Also, the following theorem is proven in Appendix A.6.3.

**Theorem 3.** *Let $\mathcal{A}_{\text{lin}}$ be the discretization functor that maps $F$ to $P_V F|_V$ for each finite subspace $V \subset X$. Let $\mathcal{D}_{smn}$ and $\mathcal{B}_{smn}$ be categories where $F \colon X \to X$ and $F_V \colon V \to V$ are strongly monotone $C^1$-functions of the form* (4). *Then, the functor $\mathcal{A}_{\text{lin}} : \mathcal{D}_{smn} \to \mathcal{B}_{smn}$ satisfies assumption (A), and it is continuous in the sense of Definition 8.*

The functor defined in Theorem 3 does not suffer from the same topological obstruction as functors for general diffeomorphisms, shown in the no-go Theorem 2. This is because when $F_V = P_V F|_V$ is strongly monotone, its derivative $D|_x F_V : V \to V$ is a strongly monotone matrix at all points $x \in V$. Therefore it is strictly positive definite (see [47, Prop. 12.3]) and the determinant $\det(D|_x F_V)$ is strictly positive. Due to this, the orientation of the finite-dimensional approximations never switch signs, and the key technique used in the proof of the no-go Theorem 2 does not apply.

A straightforward condition to guarantee strong monotonicity of a neural operator layer is given in

**Lemma 4.** *Let $F : X \to X$ be a layer of neural operator that is of the form $F(u) = u + T_2 G(T_1 u)$, where $T_j : X \to X$, $j = 1, 2$ are compact operators and $G : X \to X$ is a $C^1$-smooth map. Assume that Fréchet derivative $DG|_x$ of $G$ at $x$ satisfies the following for all $x \in X$,*

$$\|DG|_x\|_{X \to X} \leq \tfrac{1}{2} \|T_1\|_{X \to X}^{-1} \|T_2\|_{X \to X}^{-1}.$$

*Then, $F \colon X \to X$ is strongly monotone.*

See Appendixes A.6.1 and A.6.2 for the proofs.

### 3.5 Bilipschitz neural operators are conditionally strongly monotone diffeomorphisms

Now we show an analogous result to Theorem 3, but applied to bilipschitz neural operators. Moreover, we will show that all neural operator $F : X \to X$ that are bilipschitz admit approximations that can be locally inverted using iteration for each point in their range using an iteration.

**Theorem 4.** *Let $X$ be a Hilbert space. Then there is $e \in X$, $\|e\|_X = 1$ such that the following is true: Let $F : X \to X$ be a layer of a bilipschitz neural operator. Then for all $r_1 > 0$ and $\epsilon > 0$ there are a linear invertible map $A_0 : X \to X$, that is either the identity map or a reflection operator*[3]

---

[3]Note that we can write the reflection operator across the hyperplane $\{e\}^\perp$, that is, the operator $x \to x - 2\langle x, e \rangle_X e$ as a diagonal operator $\text{diag}(-1, 1, 1, \dots)$ in a suitable orthogonal basis of $X$.

$x \to x - 2\langle x, e \rangle_X e$, and strongly monotone functions $H_k$ that are also layers of neural operators such that

$$H_k : X \to X, \quad H_k(x) = x + B_k(x), \quad k = 1, 2, \ldots, J,$$

where $B_k : X \to X$ is a compact mapping and satisfies $\mathrm{Lip}(B_k) < \epsilon$ and

$$F(x) = H_J \circ \cdots \circ H_2 \circ H_1 \circ A_0(x), \quad \text{for all } x \in B_X(0, r_1). \tag{9}$$

Moreover, if $F \in C^2(X, X)$, then $J = \mathcal{O}(\epsilon^{-2})$.

The proof of Theorem 4 is in Appendix A.7.1. Theorem 4 shows that we may always decompose a bilipschitz neural operator into the composition of strongly monotone neural operator layers $H_j$ and a reflection operator $A_0$. Each $H_j$ can be discretized using the continuous functor $\mathcal{A}_{\mathrm{lin}}$ from Theorem 3. If we consider the discretization (via the construction in Definition 6) using a collection of subsets $S_0(X) \subset S(X)$ such that all $V \in S_0(X)$ satisfy $e \in V$, then the operator $A_0$ can be discretized by $A_{0,V} = P_V \circ A_0|_V$. These mean that if we write a bilipschitz neural operator as a sufficiently deep neural operator where each layer is either of the form $Id + B_j$, where $\mathrm{Lip}(B_j) < 1$, or a reflection operator $A_0$. In either case, we may use linear discretization to approximate each layer. So, we may discretize $F$ in a ball $B_X(0, R)$ by discretizing each layer $H_j$ and $A_1$ where $F = H_J \circ \cdots \circ H_1 \circ A_1$. We observe that the number of layers, $J$, depends on $R$.

We have observed that operators of the form identity plus a compact term are critical for continuous discretization. This insight motivates the introduction of residual networks as approximators within the framework of finite-rank neural operators. In what follows, we assume that $X$ is a separable Hilbert space, with an orthonormal basis $\varphi = \{\varphi_n\}_{n \in \mathbb{N}}$. For $N \in \mathbb{N}$, we define $E_N : X \to \mathbb{R}^N$ and $D_N : \mathbb{R}^N \to X$ by $E_N u := (\langle u, \varphi_1 \rangle_X, ..., \langle u, \varphi_N \rangle_X) \in \mathbb{R}^N$. $D_N \alpha := \sum_{n \leq N} \alpha_n \varphi_n$. We note that $P_{V_N} = D_N E_N$, where $P_{V_N} : X \to X$ is the projection onto $V_N := \mathrm{span}\{\varphi_n\}_{n \leq N}$. Using $E_N$, $D_N$, we define the class of residual networks in the separable Hilbert space, with $T, N \in \mathbb{N}$ and activation function $\sigma$, as

$$\mathcal{R}_{T, N, \varphi, \sigma}(X) := \Big\{ G : X \to X : G = \bigcirc_{t=1}^{T}(Id_X + D_N \circ NN_t \circ E_N),$$

$$NN_t : \mathbb{R}^N \to \mathbb{R}^N \text{ are neural networks with activation function } \sigma \ (t = 1, ..., T) \Big\}. \tag{10}$$

The following theorem proves a universality result for each of the layers $G$, allowing us to obtain a general universality result for the entire network. The statement of the theorem requires the careful construction of a neural operator-representable function $\Phi$. Giving a full description of $\Phi$ involves introducing a lot of technical notation, and so the presentation here in the main text leaves the details of the construction of $\Phi$ abridged. For the full statement of the theorem and definition of the notation, see Section A.7.2. Intuitively, $\Phi$ is the 'wrapping' of a fixed-point process in a neural operator.

**Theorem 5.** *Let $R > 0$, and let $F : X \to X$ be a layer of a bilipschitz neural operator, as in Defintion 3. Let $\sigma$ be the Rectified Cubic Unit (ReCU) function defined by $\sigma(x) := \max\{0, x\}^3$. Then, for any $\epsilon \in (0, 1)$, there are $T, N \in \mathbb{N}$ and $G \in \mathcal{R}_{T, N, \varphi, \sigma}(X)$ that has the form*

$$G = (Id_X + D_N \circ NN_T \circ E_N) \circ \cdots \circ (Id_X + D_N \circ NN_1 \circ E_N),$$

*such that each map $(Id_X + D_N \circ NN_t \circ E_N)$ is strongly monotone $C^1$-diffeomorphisms on some ball and*

$$\sup_{x \in \overline{B}_X(0, R)} \|F(x) - G \circ A(x)\|_X \leq \epsilon,$$

*where $A : X \to X$ is a linear invertible map that is either the identity map or a reflection operator $x \to x - 2\langle x, e \rangle_X e$ with some unit vector $e \in X$. Further, $G \circ A : B_X(0, R) \to G \circ A(B_X(0, R))$ is invertible, and there is some neural operator $\Phi : G \circ A(B_X(0, R)) \to A(B_X(0, R))$ so that $\left(G \circ A|_{B_X(0, R)}\right)^{-1} = A^{-1} \circ \Phi$.*

The proof is given in Section A.7.2. Neural operator $\Phi$ becomes a better approximation of the inverse operator when it becomes deeper. Theorem 5 means that neural operators are an operator algebra of nonlinear operators that are closed in composition and when the inverse of a neural operator exists, the inverse operator can be locally approximated by neural operators.

In the case when the separable Hilbert space $X$ is the real-valued $L^2$-function space $L^2(D; \mathbb{R})$, residual networks in the separable Hilbert space can be represented as residual neural operators defined by (179). See Lemma 11 for details. Then, we obtain the following

**Corollary 1.** *Let $D \subset \mathbb{R}^d$ be a bounded domain, and let $\varphi = \{\varphi_n\}_{n \in \mathbb{N}}$ be an orthonormal basis in $L^2(D; \mathbb{R})$. Assume that the orthonormal basis $\varphi$ include the constant function. Let $\mathcal{RNO}_{T,N,\varphi,\sigma}(L^2(D; \mathbb{R}))$ be the class of residual neural operators defined in (179). Then, the statement replacing $X$ with $L^2(D; \mathbb{R})$ and $G \in \mathcal{R}_{T,N,\varphi,ReLU}(X)$ with $G \in \mathcal{RNO}_{T,N,\varphi,ReLU}(L^2(D; \mathbb{R}))$ in Theorem 5 holds.*

The proof is given by a combination of Theorem 5 and Lemma 11. We note that the assumption that the orthonormal basis $\varphi$ includes the constant function is satisfied if we choose $\varphi$ to be a Fourier basis, which yields the Fourier neural operator, see e.g., [39, 38].

**Remark 1.** *In this section, we have shown that residual networks in a separable Hilbert space $X$, defined in (10), are universal approximators for layers of bilipschitz neural operators. Additionally, in the specific case where $X = L^2(D; \mathbb{R})$, residual neural operators, defined in Definition 9, also provide universal approximators for layers of bilipschitz neural operators. We note that the residual network we have discussed is locally invertible but not globally. By introducing invertible residual networks on Hilbert space $X$, defined in (166), we can similarly prove that these networks by employing sort activation functions (see [3, Section 4]) are universal approximators for strongly monotone diffeomorphisms with compact support. Specifically, when $X = L^2(D; \mathbb{R})$, invertible residual neural operators, defined in Definition 9, are also universal approximators for strongly monotone diffeomorphisms with compact support. For further details, we refer to Appendix B.*

## 4  Quantitative approximation

Quantitative approximation results for neural networks, see e.g. [57] or [23], can be used to derive quantitative error estimates for discretization operations. Let $F \colon \overline{B}_X(0, r) \to X$ be a non-linear function satisfying $F \in \text{Lip}(\overline{B}_X(0, r); X)$, in $n = 1$, or $F \in C^n(\overline{B}_X(0, r); X)$, if $n \geq 2$. Then, $F$ can be discretized using neural networks in the following way: Let $\varepsilon_V > 0$ be numbers indexed by the linear subspaces $V \subset X$ such that $\varepsilon_V \to 0$ as $V \to X$. When $\vec{\varepsilon} = (\varepsilon_V)_{V \in S(X)}$, in the sense of Definition 1, an $\vec{\varepsilon}$-approximation operation $\mathcal{A}_{NN} \colon F \to (F_V)_{V \in S(X)}$ in the ball $B_X(0, r)$ can be be obtained by defining $F_V = J_V^{-1} \circ F_{V,\theta} \circ J_V \colon V \to V$, where $J_V \colon V \to \mathbb{R}^d$ is an isometric isomorphism, $d = \dim(V)$, $F_{V,\theta} \colon \mathbb{R}^d \to \mathbb{R}^d$ is a feed-forward neural network with ReLU-activation functions with at most $C(d) \log_2((1+r)/\varepsilon_V)$ layers and $C(d)\varepsilon_V^d \log_2((1+r)/\varepsilon_V)$ non-zero elements in the weight matrices. Details of this result are given in Proposition 5 in Appendix A.8.

## 5  Conclusion

In this work, we have studied the problem of discretizing neural operators between Hilbert spaces. Many physical models concern functions $\mathbb{R}^n \to \mathbb{R}$, for example $L^2(\mathbb{R}^n)$, the computational methods based on approximations in finite dimensional spaces should become better when the dimension of the model grows and tends to infinity. We have focused on diffeomorphisms in infinite dimensions, which are crucial to understand in generative modeling. We have shown that the approximation of diffeomorphisms leads to computational difficulties. We used tools from category theory to produce a no-go theorem showing that general diffeomorphisms between Hilbert spaces may not admit any continuous approximations by diffeomorphisms on finite spaces, even if the approximations are allowed to be nonlinear. We then proceeded to give several positive results, showing that diffeomorphisms between Hilbert spaces may be continuously approximated if they are further assumed to be strongly monotone. Moreover, we showed that the difficulties can be avoided by considering a restricted but still practically rich class of diffeomorphisms. This includes bilipschitz neural operators, which may be represented in any bounded set as a composition of strongly monotone neural operators and strongly monotone diffeomorphisms. We then showed that such operators may be inverted locally via an iteration scheme. Finally we gave a simple example on how quantitative stability questions can be obtained for discretization functors, inviting other researchers to study related questions using more sophisticated methods.

## Acknowledgments

TF was supported by JSPS KAKENHI Grant Number JP24K16949, JST CREST JPMJCR24Q5, JST ASPIRE JPMJAP2329, and Grant for Basic Science Research Projects from The Sumitomo Foundation. MP was supported by CAPITAL Services of Sioux Falls, South Dakota and NSF-DMS under grant 3F5083. MVdH was supported by the Simons Foundation under the MATH + X program, the Department of Energy, BES under grant DE-SC0020345, and the corporate members of the Geo-Mathematical Imaging Group at Rice University. A significant part of the work of MVdH was carried out while he was an invited professor at the Centre Sciences des Données at Ecole Normale Supérieure, Paris. M.L. was partially supported by a AdG project 101097198 of the European Research Council, Centre of Excellence of Research Council of Finland and the FAME flagship. Views and opinions expressed are those of the authors only and do not necessarily reflect those of the funding agencies or the EU.

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

# A  Proofs and additional examples

## A.1  Non-linear discretization

In this section we consider the non-linear discretization theory, see [9]. Let $\Omega \subset \mathbb{R}^n$ be a bounded open set with smooth boundary. Let $H^1(\Omega) = \{u \in L^2(\Omega) \mid \nabla u \in L^2(\Omega)\}$ and $H_0^1(\Omega) = \{u \in H^1(\Omega) \mid u|_{\partial\Omega} = 0\}$ be the Sobolev spaces with an integer order of smoothness and $H^s(\Omega)$, $s \in \mathbb{R}$ the Sobolev space with a fractional order of smoothness [53].

For simplicity, we assume in the section that $n = 1$, and that $\Omega \subset \mathbb{R}$ is an interval and we point out that in this case the Sobolev embedding theorem implies $H^1(\Omega) \subset C(\overline{\Omega})$ which significantly simplifies the constructions below.

Let $g \in C^{m+1}(\mathbb{R}; \mathbb{R})$, $m \geq 1$, be the derivative of a convex function $G \in C^{m+2}(\mathbb{R}; \mathbb{R})$ satisfying

$$-c_0 \leq G(r) \leq c_1(1 + r^p), \quad r \in \mathbb{R}, \ c_0, c_1, p > 0. \tag{11}$$

Let $F = F^{(g)} \colon H_0^1(\Omega) \to H_0^1(\Omega)$, $F^{(g)} \colon x \to u$ be the solution operator of the non-linear differential equation

$$\Delta u(t) - g(u(t)) = x(t), \quad t \in \Omega, \tag{12}$$
$$u|_{\partial\Omega} = 0, \tag{13}$$

where $\Delta$ is the Laplace operator operating in the $t$ variable, where the function $x(t)$ is a physical source term. Because the $u$ that solves (12)-(13) is the minimizer of the strictly convex and lower semi-continuous function $H : X \to \mathbb{R}$,

$$H(v) = \|\nabla v\|_{L^2(\Omega)^2}^2 + \int_\Omega (G(v(t)) - x(t)v(t))dt, \tag{14}$$

the Weierstrass theorem, see [50], Theorem II.8.1, implies that (12)-(13) has a unique solution. Moreover, by regularity theory for elliptic equations, see [18], Theorem 8.13, we have that the solution $u$ is in $H^3(\Omega)$. Moreover, as $G$ is convex and satisfies (11), the function $H$ in (14) is coercive, for any $R > 0$ there is $\rho(R) > 0$ such that when $\|x\|_{H_0^1(\Omega)} \leq R$, then $\|u\|_{H^3(\Omega)} \leq \rho(R)$.

The equation (12)-(13) can be written also as an integral equation

$$u(t) + \int_\Omega \Psi(t, t')g(u(t'))dt' = -\int_\Omega \Psi(t, t')x(t')dt', \quad t \in \Omega, \tag{15}$$

where $\Psi(t, t')$ is the Dirichlet Green's function of the negative Laplacian, that is, $-\Delta\Psi(\cdot, t') = \delta_{t'}(\cdot)$, $\Psi(\cdot, t')|_{\partial\Omega} = 0$, and $g_*(u) = g(u)$. We can write (15) as

$$(Id + Q \circ g_*)u = -Q(x), \quad Qu(t) = \int_\Omega \Psi(t, t')x(t')dt', \tag{16}$$

where $i_{H_0^1 \to H^{3/4}} : H_0^1(\Omega) \to H^{3/4}(\Omega)$ and $i_{H^2 \to H_0^1} : H^2(\Omega) \cap H_0^1(\Omega) \to H_0^1(\Omega)$ are compact operators and as $H^{3/4}(\Omega) \subset C(\overline{\Omega})$, the operators $g_* : H^{3/4}(\Omega) \to L^2(\Omega)$ and $Q : L^2(\Omega) \to H^2(\Omega) \cap H_0^1(\Omega)$ are continuous, we see that

$$F = Id + i_{H^2 \to H^1} \circ (Q \circ g_*) \circ i_{H^1 \to H^{3/4}} : H_0^1(\Omega) \to H_0^1(\Omega),$$

is a layer of the neural operator. As $G \in C^{m+1}$, $m \geq 2$, we have that $F : H_0^1(\Omega) \to H_0^1(\Omega)$ is $C^1$-smooth. Moreover, as $G$ is convex, the operator $F : H^1(\Omega) \to H^1(\Omega)$ is strongly monotone and hence by Lemma 3, $F$ has an inverse map $F^{-1} : H^1(\Omega) \to H^1(\Omega)$ that is $C^1$-smooth. Thus, the solution $u$ of (12)-(13) can be represented as

$$u = F^{-1}(Q(x)).$$

Let $V \subset X = H_0^1(\Omega)$ be a finite dimensional space. The Galerkin methods to obtain an approximation for the solution of the boundary value problem (12)-(13) using Finite Element Method. To do this, let $w \in V$ be the solution of the problem

$$\int_\Omega (\nabla w(t) \cdot \nabla \phi(t) + g(w(t))\phi(t))dt = -\int_\Omega x(t) \cdot \phi(t)dt, \quad \text{for all } \phi \in V. \tag{17}$$

When $\phi_1, \ldots, \phi_d \in V$ are a basis of the finite dimensional vector space $V$, and we write $w = \sum_{k=1}^{d} w_k \phi_j$, the problem (17) is equivalent that $(w_1, \ldots, w_d) \in \mathbb{R}^d$ satisfies the system equations $d$ equations

$$\sum_{k=1}^{d} b_{jk} w_k = -\int_\Omega \phi_j(t) g\left(\sum_{k=1}^{d} w_k \phi_k(t)\right) dt - \int_\Omega x(t) \cdot \phi_j(t) dt, \quad \text{for } j = 1, 2, \ldots, d, \quad (18)$$

where

$$b_{jk} = \int_\Omega \nabla \phi_j(t) \cdot \nabla \phi_k(t) dt.$$

When $x \in V$, we define the map $F_V^{(g)} \colon V \to V$ by setting map $F_V^{(g)}(x) = w$, where $w \in V$ is the solution of the problem (17). When

$$\mathcal{F}_{PDE} = \{F^{(g)} \colon X \to X \mid g(s) = \frac{dG}{ds}(s), G \in C^{m+2}(\mathbb{R}) \text{ is convex and satisfies (11)}\},$$

we define the map $\mathcal{A}_{FEM} \colon F^{(g)} \to F_V^{(g)}$ for $F^{(g)} \in \mathcal{F}_{PDE}$. As the the function $g$ in the equation (18) is non-linear, the solution $u$ of the continuum problem (15)-(13) and the solution $w$ of the finite dimensional problem have typically no linear relationship, even if the source $x$ satisfies $x \in V$.

Let $R > 0$ As the embedding $H^2(\Omega) \cap H_0^1(\Omega) \to H_0^1(\Omega)$ is compact and $Q \colon H_0^1(\Omega) \to H^2(\Omega) \cap H_0^1(\Omega)$ is continuous, we see that the image of the closed ball $\overline{B}_X(0, R)$ in the map $Q$ is a precompact set $Q(\overline{B}_X(0, R)) \subset X$. As $F^{-1} \colon X \to X$ is continuous the set of corresponding solutions,

$$Z_R = \{u \in X \mid u = F^{-1}(Q(x)), \; x \in \overline{B}_X(0, R)\},$$

is precompact in $X$. Hence,

$$\lim_{V \to X} \sup_{u \in Z_R} \|(Id - P_V)(u)\|_X = 0. \quad (19)$$

As $G$ is convex, the Fréchet derivative of the map $F = Id + Q \circ g_* \colon X \to X$ is a strictly positive linear operator at all points $x \in X$. This, the uniform convergence (19), and the convergence results for the Galerkin method for semi-linear equations, [49, Theorem 3.2], see also [24], imply that in the space $X = H_0^1(\Omega)$

$$\lim_{V \to X} \sup_{\|x\|_X \leq R} \|F_V^{(g)}(x) - F^{(g)}(x)\|_X = 0. \quad (20)$$

These imply that $\mathcal{A}_{FEM} \colon F^{(g)} \to F_V^{(g)}$ is an approximation operation for the function $F^{(g)} \in \mathcal{F}_{PDE}$.

The map $F_V^{(g)} \colon V \to V$ is called the Galerkin approximation of the problem (12)-(13) and is an example of the non-linear approximation methods. The properties of the Galerkin approximation is studied in detail in [8], in particular sections 4.4 and 4.5.

## A.2 Negative results

On the positive results, in Appendix A.1, we have considered nonlinear discretiation of the operators $u \to -\Delta u + g(u)$. To exemplify the negative result, we consider the following example on the solution operation of differential equations and the non-existence of approximation by diffeomorphic maps: We consider the elliptic (but not not strongly elliptic) problem

$$B_s u := -\frac{d}{dt}\left((1+t)\text{sign}(t-s)\frac{d}{dt}u(t)\right) = f(t), \quad t \in [0,1], \quad (21)$$

with the Dirichlet and Neumann boundary conditions

$$u(0) = 0, \quad \frac{d}{dt}u(t)\bigg|_{t=1} = 0. \quad (22)$$

Here, $0 \leq s \leq 1$ is a parameter of the coeffient function and $\text{sign}(t-s) = 1$ if $t > s$ and $\text{sign}(t-s) < 0$ if $t < s$. We consider the weak solutions of (21) in the space

$$u \in H_{D,N}^1(0,1) := \{v \in H^1(0,1) : v(0) = 0\}.$$

We can write

$$B_s u = -D_t^{(2)} A_s D_t^{(1)} u,$$

where

$$A_s v(t) = (1 + t)\text{sign}(t - s)v(t),$$

parametrized by $0 \leq s \leq$, are multiplication operations that are invertible operators, $A_s : L^2(0, 1) \rightarrow L^2(0, 1)$ (this invertibility makes the equation (21) elliptic). Moreover, $D_t^{(1)}$ and $D_t^{(2)}$ are the operators $v \rightarrow \frac{d}{dt}v$ with the Dirichlet boundary condition $v(0) = 0$ and $v(1) = 0$, respectively. We consider the Hilbert space $X = H_{D,N}^1(0, 1)$; to generate an invertible operator $G_s : X \rightarrow X$ related to (21), we write the source term using an auxiliary function $g$,

$$f(t) = Qg := -\frac{d^2}{dt^2}g(t) + g(t).$$

Then the equation,

$$B_s u = Qg, \tag{23}$$

defines a continuous and invertible operator,

$$G_s : X \rightarrow X, \quad G_s : g \rightarrow u.$$

In fact, $G_s = B_s^{-1} \circ Q$ when the domains of $B_s$ and $Q$ are chosen in a suitable way. The Galerkin method (that is, the standard approximation based on the Finite Element Method) to approximate the equation (23) involves introducing a complete basis $\chi_j(t)$, $j = 1, 2, \ldots$ of the Hilbert space $X$, the orthogonal projection

$$P_n : X \rightarrow X_n := \text{span}\{\chi_j : j = 1, 2, \ldots, n\},$$

and approximate solutions of (23) through solving

$$P_n B_s P_n u_n = P_n Q P_n g_n, \quad u_n \in X_n, \ g_n = P_n g. \tag{24}$$

This means that operator $B_s^{-1} Q : g \rightarrow u$ is approximated by $(P_n B_s P_n)^{-1} P_n Q P_n : g_n \rightarrow u_n$, when $P_n B_s P_n : X_n \rightarrow X_n$ is invertible.

The above corresponds to the Finite Element Method where the matrix defined by the operator $P_n B_s P_n$ is $\mathbf{b}(s) = [b_{jk}(s)]_{j,k=1}^n \in \mathbb{R}^{n \times n}$, where

$$b_{jk}(s) = \int_0^1 (1 + t)\text{sign}(t - s)\frac{d}{dt}\chi_j(t) \cdot \frac{d}{dt}\chi_k(t)dt, \quad j, k = 1, \ldots, n.$$

Since we used the mixed Dirichlet and Neumann boundary conditions in the above boundary value problem, we see that for $s = 0$ all eigenvalues of the matrix $\mathbf{b}(s)$ are strictly positive, and when $s = 1$ all eigenvalues are strictly negative. As the function $s \rightarrow \mathbf{b}(s)$ is a continuous matrix-valued function, we see that there exists $s \in (0, 1)$ such that the matrix $\mathbf{b}(s)$ has a zero eigenvalue and is no invertible. Thus, we have a situation where all operators $B_s^{-1} Q : g \rightarrow u$, $s \in [0, 1]$ are invertible (and thus define diffeomorphisms $X \rightarrow X$) but for any basis $\chi_j(t)$ and any $n$ there exists $s \in (0, 1)$ such that the finite dimensional approximation $\mathbf{b}(s) : \mathbb{R}^n \rightarrow \mathbb{R}^n$ is not invertible. This example shows that there is no FEM-based discretization method for which the finite dimensional approximations of all operators $B_s^{-1} Q$, $s \in (0, 1)$, are invertible. The above example also shows a key difference between finite and infinite dimensional spaces. The operator $A_s : L^2(0, 1) \rightarrow L^2(0, 1)$ has only continuous spectrum and not eigenvalues nor eigenfunctions whereas the finite dimensional matrices have only point spectrum (that is, eigenvalues). The continuous spectrum makes it possible to deform the positive operator $A_s$ with $s = 0$ to a negative operator $A_s$ with $s = 1$ in such a way that all operators $A_s$, $0 \leq s \leq 1$, are invertible but this is not possible to do for finite dimensional matrices. We point out that the map $s \rightarrow A_s$ is not continuous in the operator norm topology but only in the strong operator topology and the fact that $A_0$ can be deformed to $A_1$ in the norm topology by a path that lies in the set of invertible operators is a deeper result. However, the strong operator topology is enough to make the FEM matrix $\mathbf{b}(s)$ to depend continuously on $s$.

## A.3 Application of Theorem 2 to Neural Operators

In this section, we give an example of the application of Theorem 2, as applied to the problem of using a neural operator to solve a linear PDE. The guiding idea is to draw inspiration from Figure 1; the space of diffeomorphisms on $X$ is disconnected if $X$ is a finite-dimensional space, but connected if it is a Hilbert space.

Before constructing the path, we first construct a isotopy of diffeomorphisms in $L^2(\Omega)$ in several steps

Let $\rho \colon \mathbb{R} \to [0,1]$ be a smooth function such that $\rho|_{(-\infty,0]} = 0$, $\rho|_{[1,\infty)} = 1$. Let $R(\theta) \in \mathbb{R}^{2\times 2}$ be the rotation matrix given by

$$R(\theta) := \begin{pmatrix} \cos(\theta) & \sin(\theta) \\ -\sin(\theta) & \cos(\theta) \end{pmatrix}. \tag{25}$$

Then, define the function $R_i \colon \mathbb{R} \to \mathbb{R}^{2\times 2}$ by

$$R_i(t) := R(\rho(t-i)\pi). \tag{26}$$

Note that $R_i(t) \in SO(2)$, for $t \leq i$, $R_i(t) = I$, $t \geq i+1$, $R_i(t) = -I$.

Next, we consider the 'little ell' space, $\ell^2 := \{(\alpha_1, \alpha_2, \dots,) \colon \sum_{i=1}^{\infty} \alpha_i^2 < \infty\}$. Consider the linear operator $\hat{\mathcal{R}} \colon \mathbb{R} \to (\ell^2 \to \ell^2)$ on $\ell^2$ given by a pair-wise coordinate representation given by the infinite dimensional 'matrix' with block structure

$$\hat{\mathcal{R}}(t) := \begin{pmatrix} R_1(t) & & \\ & R_2(t) & \\ & & \ddots \end{pmatrix}. \tag{27}$$

Finally, define $\mathcal{R} \colon [0,1] \to (\ell^2 \to \ell^2)$ by

$$\mathcal{R}(t) := \begin{cases} \hat{\mathcal{R}}\left(\frac{1}{1-t}\right) & t \in [0,1) \\ -I & t \geq 1 \end{cases}, \tag{28}$$

and define the related operator $\tilde{\mathcal{R}} \colon [0,1] \to (\ell^2 \to \ell^2)$ by

$$\tilde{\mathcal{R}}(t) = \begin{pmatrix} -1 & \\ & \mathcal{R}(t) \end{pmatrix}. \tag{29}$$

**Proposition 3** ($\mathcal{R}, \tilde{\mathcal{R}}$ are isotopies of smooth diffeomorphisms in $\ell^2$)**.** *The maps $H_1 \colon \ell^2 \times [0,1] \to \ell^2$ and $H_2 \colon \ell^2 \times [0,1] \to \ell^2$ defined by*

$$H_1(v,t) = \mathcal{R}(t)v, \quad H_2(v,t) = \tilde{\mathcal{R}}(t)v, \tag{30}$$

*are isotopies of smooth diffeomorphisms in $\ell^2$ and agree at $t = 1$.*

*Proof.* We prove that $H_1$ is an isotopy of smooth diffeomorphisms on $\ell^2$, as the proof of the same for $H_2$ is very similar.

We first show that $H_1(\cdot, t) \in \text{diff}(\ell^2)$ for each $t$.

For $t < 1$, $\mathcal{R}$ operates on vectors in $\ell^2$ in only a finite number of indices. In those indices it corresponds to a rotation, and hence is a smooth diffeomorphism. When $t \geq 1$, it corresponds to scalar multiplication by $-1$, and so is too a smooth diffeomorphism. Now we show that $\mathcal{R}(t)$ is continuous on . The only point where continuity may fail is at $t = 1$, and there it may only fail in the

limit from the left. Given a $v \in \ell^2$, we compute

$$\lim_{t \to 1^-} \|\mathcal{R}(t)v - \mathcal{R}(1)v\|_2 = \lim_{t \to 1^-} \left\| \begin{pmatrix} R_1(\frac{1}{1-t})\begin{pmatrix} v_1 \\ v_2 \end{pmatrix} \\ R_2(\frac{1}{1-t})\begin{pmatrix} v_3 \\ v_4 \end{pmatrix} \\ \vdots \\ R_k(\frac{1}{1-t})\begin{pmatrix} v_{2k-1} \\ v_{2k} \end{pmatrix} \\ R_{k+1}(\frac{1}{1-t})\begin{pmatrix} v_{2k+1} \\ v_{2k+2} \end{pmatrix} \\ \vdots \end{pmatrix} + \begin{pmatrix} v_1 \\ v_2 \\ v_3 \\ v_4 \\ \vdots \\ v_{2k-1} \\ v_{2k} \\ v_{2k+1} \\ v_{2k+2} \\ \vdots \end{pmatrix} \right\|_2. \tag{31}$$

Let $k^*(t) := \lfloor \frac{1}{1-t} \rfloor$. Then for any integer $i \leq k$, we have that

$$R_i\left(\frac{1}{1-t}\right) = R\left(\rho\left(\underbrace{\frac{1}{1-t} - i}_{\geq 1}\right)\pi\right) = R(\pi) = -I \in \mathbb{R}^{2\times2}, \tag{32}$$

and likewise when $i \geq k^*(t) + 1$, then $R_i\left(\frac{1}{1-t}\right) = I \in \mathbb{R}^{2\times2}$, and so the r.h.s. of Eqn. 31 becomes

$$\lim_{t \to 1^-} \left\| \begin{pmatrix} -v_1 \\ -v_2 \\ -v_3 \\ -v_4 \\ \vdots \\ R_{k^*(t)}(\frac{1}{1-t})\begin{pmatrix} v_{2k^*(t)-1} \\ v_{2k^*(t)} \end{pmatrix} \\ v_{2k^*(t)+1} \\ v_{2k^*(t)+2} \\ \vdots \end{pmatrix} + \begin{pmatrix} v_1 \\ v_2 \\ v_3 \\ v_4 \\ \vdots \\ v_{2k^*(t)-1} \\ v_{2k^*(t)} \\ v_{2k^*(t)+1} \\ v_{2k^*(t)+2} \\ \vdots \end{pmatrix} \right\|_2 \tag{33}$$

$$= \lim_{t \to 1^-} \left\| \begin{pmatrix} 0 \\ 0 \\ 0 \\ 0 \\ \vdots \\ R_{k^*(t)}(\frac{1}{1-t})\begin{pmatrix} v_{2k^*(t)-1} \\ v_{2k^*(t)} \end{pmatrix} + \begin{pmatrix} v_{2k^*(t)-1} \\ v_{2k^*(t)+2} \end{pmatrix} \\ 2v_{2k^*(t)+1} \\ 2v_{2k^*(t)+2} \\ \vdots \end{pmatrix} \right\|_2 \tag{34}$$

$$\leq 2\sqrt{\sum_{i=k^*(t)}^{\infty} v_i^2}. \tag{35}$$

As $t \to 1^-$, $k^*(t) \to \infty$, and so, by standard estimates, $2\sqrt{\sum_{i=k^*(t)}^{\infty} v_i^2} \to 0$. Hence, $\mathcal{R}(t)$ is continuous on $\ell^2$ at $t = 1$.

Finally, by definition at $t = 1$, we have that $H_1(v, 1) = \mathcal{R}(1) = -I = \mathcal{R}(1) = H_2(v, 1)$. $\square$

Finally, we define $H: \mathbb{R}^\infty \times [0, 1] \to \mathbb{R}^\infty$ by gluing the isotopies $H_1$ and $H_2$ together by

$$H(v, t) := \begin{cases} H_1(v, 2t) & \text{if } t \leq \frac{1}{2} \\ H_2(v, 2 - 2t) & \text{if } t > \frac{1}{2} \end{cases}. \tag{36}$$

**Proposition 4.** *The function $H$ given by Eqn. 36 is an isotopy of diffeomorphisms in $\ell^2$.*

*Proof.* Continuity of $H$ follows from continuity of $\mathcal{R}$, $\tilde{\mathcal{R}}$, and that $\mathcal{R}(1) = \tilde{\mathcal{R}}(1)$. For each $t \in [0, 1]$, $H(\cdot, t) \in \operatorname{diff}(\ell^2)$. $\qquad\square$

## A.4  Proofs from Sec. 3.1

### A.4.1  Proof of Theorem 2

*Proof.* Assume that $\mathcal{A} \colon \mathcal{D} \to \mathcal{B}$ is a functor that satisfies the property (A) of an approximation functor and is continuous. Let us consider the case when $X = \ell^2$.

Let $e \in X$ be a unit vector and $B_e : X \to X$ be a linear map $Bx = x - 2\langle x, e \rangle_X e$. In other words, $B$ is a diagonal matrix $\operatorname{diag}(-1, 1, \dots)$ in some orthogonal basis where $e$ is the first basis vector.

By [32, Theorem 2], see also and [45], for an infinite dimensional Hilbert space $X$ the space $GL(X)$ of linear invertible maps have only one topological component in the operator norm topology and the set $GL(X)$ is contractible and hence path-connected. This implies that there are linear maps $A_t : X \to X$, $t \in [0, 1]$ such that $A_0 = Id$, and $A_1 \in GL(X)$ is arbitrary invertible linear map and that $t \to A_t \in GL(X)$ is continuous. Similarly, in an infinite dimensional real Hilbert space $X$ the space $O_\mathbb{R}(X)$ of the orthogonal operators $A : X \to X$ is path-connected in the operator norm topology, see [46, Section 4]. We recall that an orthogonal operator $A : X \to X$ is a bounded linear operator satisfying $A^* A = A A^* = I$, where $A^* : X \to X$ is the adjoint (i.e., the transpose) of the operator $A : X \to X$. Observe that in a finite dimensional space $\mathbb{R}^n$ the set $O_\mathbb{R}(\mathbb{R}^n)$ has two disjoint topological components, those which determinant is 1 and those which determinant is $-1$, and thus the set $O_\mathbb{R}(\mathbb{R}^n)$ is not connected.

A bounded linear operator $B : X \to X$ is called a rotation operator of the form $B = e^S$ where $S : X \to X$ is a linear, bounded, skew-symmetric operator, $S^* = -S$, and $e^S = I + S + \frac{1}{2!}S^2 + \frac{1}{3!}S^3 + \dots$. An orthogonal operator $A$ is called a reflection if it is not a rotation.

One of the fundamental reasons for the surprising property that the space $O_\mathbb{R}(X)$ of orthogonal operators in $X$ is path-connected is that in an infinite dimensional real Hilbert space $X$ every orthogonal operator $A : X \to X$ can be represented as a product of two rotation operators (that is no valid in the finite dimensional spaces), that is, $A = e^{S_1} e^{S_2}$ where $S_1, S_2 : X \to X$ are skew-symmetric linear operators. Thus any operator $A \in O_\mathbb{R}(X)$ can be connected to the identity operator $Id : X \to X$ via a path $\alpha : t \to e^{t S_1} e^{t S_2} \in O_\mathbb{R}(X)$, $t \in [0, 1]$, so that $\alpha(0) = Id$ and $\alpha(1) = A$.

As a motivating example on the fact that $O_\mathbb{R}(X)$ is connected in the operator norm topology, is to consider a similar, but easier result in the strong operator topology, that is, topology generated by the evaluation maps $A \to A(u)$, $u \in L^2(0, 1)$. Observe that the strong operator topology is weaker than the operator norm topology, but in finite dimensional space those are equivalent.. So, let us next show that the operators $Id$ and $-Id$ are in the same topological component of $O_\mathbb{R}(X)$ in the strong operator topology. As all infinite dimensional separable Hilbert spaces are isometric to $L^2(0, 1)$, let us consider the Hilbert space $L^2(\mathbb{R})$ and the orthogonal linear operators $A_s : L^2(0, 1) \to L^2(0, 1)$, $s \in [0, 1]$ that are given by

$$A_s u(t) = \operatorname{sign}(t - s) \cdot u(t), \tag{37}$$

where $u \in L^2(0, 1)$ and $\operatorname{sign}(t)$ is the sign of the real number $t$. Then, for every $u \in L^2(0, 1)$ the functions

$$a_u : s \to A_s(u),$$

are continuous functions $a_u : [0, 1] \to L^2(0, 1)$. As $A_0 = Id$ and $A_0 = -Id$, we see that

$$s \to A_s \in O_\mathbb{R}(L^2(0, 1)), \quad s \in [0, 1],$$

is a continuous path in the strong operator topology, that connects the operator $Id$ to $-Id$.

We recall that the continuity of the function $s \to A_s$ in the strong operator topology, means that for all $u \in L^2(0, 1)$ the map

$$s \to A_s u \in L^2(0, 1),$$

is continuous. As for $s > s_0$

$$\|A_s u - A_{s_0} u\|^2_{L^2(0,1)} = \int_{s_0}^s |2u(x)|^2 dx \to 0,$$

as $s \to s_0$, we see that $s \to A_s$ is continuous in the strong operator topology. All maps $A_s :$ $L^2(0,1) \to L^2(0,1)$ are invertible and $A_0 = Id$, $A_1 = -Id$. This has implications e.g. for Finite Element method analysis of partial differential equations (See Appendix A.5.2).

Note that when $X$ is any infinite dimensional separable Hilbert space with real scalars, there is an isometry $J : X \to L^2(0,1)$ (e.g., the linear operator mapping an orthogonal basis of $X$ to an orthogonal basis $L^2(0,1)$). Then, the maps $\tilde{A}_s : X \to X$ given by $\tilde{A}_s = J^{-1} \circ A_s \circ J : X \to X$, where $A_s : L^2(0,1) \to L^2(0,1)$ are given above, are continuous paths in $O_\mathbb{R}(X)$ from $\tilde{A}_0 = Id :$ $X \to X$ to $\tilde{A}_1 = -Id : X \to X$. As discussed above, the deep result that $Id$ and $-Id$ can be connected by a continuous path in the operator norm topology of $O_\mathbb{R}(X)$ is given in [32, Theorem 2], see also [45].

Let us first warm up by proving the claim in the case when an additional assumption (B) is valid:

(B) : When $V \subset X$ is finite dimensional discretization maps $F_V$ of linear invertible maps $F : X \to X$ are linear invertible maps and moreover, the set $S_0(X) = S(X)$ consists of all linear subspaces of $X$.

Under assumptions (A) and (B), we consider the case when $A_1 = -Id$ and denote by $F_t = A_t :$ $X \to X$ a family of linear maps such that $A_0 = Id$, and that $t \to A_t \in GL(X)$ is continuous path of operators connecting $Id$ to $A_1 = -Id$. Let $(X, S(X), (F_{t,V})_{V \in S(X)}) = \mathcal{A}(X, F_t)$, that is, $F_{t,V} : V \to V$ be the linear isomorphism that is the discretizations of the map $F_t : X \to X$. As the functor $\mathcal{A}$ is continuous, the map $t \to F_{t,V}$ is a continuous map from $[0,1]$ to the space of linear operators endowed with the topology of uniform convergence on compact sets, c.f. limit (7). As $F_{t,W}$ are linear, this implies that for all $t' \in [0,1]$,

$$\lim_{t \to t'} \|F_{t,W} - F_{t',W}\| = \lim_{t \to t'} \sup_{x \in V \cap \overline{B}_X(0,1)} \|F_{t,W}(x) - F_{t',W}(x)\|_X = 0. \tag{38}$$

and hence the map $t \to F_{t,W}$ is a continuous map $[0,1] \to GL(W)$.

Let $0 < \epsilon_0 < 1/2$ and $r > 1$. Let $t_0 = 0$ and $t_1 = 1$. Using Property (A) for operators $F_{t_j}$, $j = 0, 1$, we see that there are $W_0, W_1 \in S_0(X)$ such that for all $V \in S_0(X)$ satisfying $V \supset W_j$ we have

$$\sup_{x \in \overline{B}_X(0,r) \cap V} \|F_{t_j, W_j}(x) - F_{t_j}(x)\|_X < \epsilon_0, \quad \text{for } j \in \{0, 1\}.$$

Let $W \in S_0(X)$ be such a finite dimensional space which dimension is an odd integer and that $W_0 + W_1 \subset W$. Then,

$$\sup_{x \in \overline{B}_X(0,r) \cap W} \|F_{t_j, W}(x) - F_{t_j}(x)\|_X \le \epsilon_0, \quad \text{for } j \in \{0, 1\}, \tag{39}$$

Clearly, $A_0 = Id$ satisfies $A_0(W) = W$ and $A_0 : W \to W$ is an invertible linear map. Similarly, the map $A_1 = -Id$ satisfies $A_1(W) = W$ and $A_1 : W \to W$ is an invertible linear map. These observation and assumptions (A) and (B) imply that $F_{0,W} : W \to W$ and $F_{1,W} : W \to W$ are linear maps and by inequality (39),

$$\|F_{0,W} - A_0|_W\|_{W \to W} < \epsilon_0 < \frac{1}{2}, \quad \|F_{1,W} - A_1|_W\|_{W \to W} < \epsilon_0 < \frac{1}{2}, \tag{40}$$

and as the dimension of $W$ is odd, we have

$$\det(A_0|_W) = \det(Id|_W) = 1, \quad \det(A_1|_W) = \det(-Id|_W) = -1.$$

Let $B_{0,s} = (1-s)A_0|_W + sF_{0,W} : W \to W$ and $B_{1,s} = (1-s)A_1|_W + sF_{0,W} : W \to W$. As $\|(A_0|_W)^{-1}\|_{W \to W} = \|(A_1|_W)^{-1}\|_{W \to W} = 1$, we see using (40) that the maps $B_{0,s}^{-1} : W \to W$ and $B_{1,s}^{-1} : W \to W$ are invertible and that the functions $s \to B_{0,s} \in GL(W)$ and $s \to B_{1,s} \in GL(W)$ are continuous matrix-valued functions which determinants does not vanish and

$$\det(B_{0,s}) = \det(A_0|_W) > 0, \quad \det(B_{1,s}) = \det(A_1|_W) < 0,$$

for all $s \in [0,1]$. These imply that

$$\det(F_{0,W}) = \det(B_{0,1}) > 0, \quad \det(F_{1,W}) = \det(B_{0,1}) < 0, \tag{41}$$

However, as $t \to F_{t,W}$ is a continuous maps $[0,1] \to GL(W)$, see (38), we have that

$$t \to \det(F_{t,W}), \tag{42}$$

is a continuous function $[0,1] \to \mathbb{R}$ which does not obtain value zero and thus has a constant sign. However, this is in contradiction with formula (41).

Next we return to the main part of the proof where we do not assume that assumption (B) is valid, but we only assume that the assumption (A) is valid. In this case, we use degree theory instead of the determinants.

Next, let $F_t = A_t : X \to X$ be a family of linear maps such that $A_0 = Id$, but that $A_1(x) = B_e(x) = x - 2e\langle x, e\rangle_X$ is a reflection, where $e \in X$ is a unit vector, and $t \to A_t \in GL(X)$ is continuous. Again, let $(X, S_0(X), (F_{t,V})_{V \in S_0(X)}) = \mathcal{A}(X, F_t)$, that is, $F_{t,V} : V \to V$ be $C^1$-diffeomorphisms that are discretizations of the map $F : X \to X$. As the functor $\mathcal{A}$ is continuous, the map $t \to F_{t,V}|_{\overline{B}(0,R)} \in C(\overline{B}(0,R))$ is a continuous map for all $V \in S_0(X)$ and $R > 0$. Moreover, we note that by our assumptions on $\mathcal{B}$, the discretized map $F_{t,V} : V \to V$, is $C^1$-diffeomorphism of $V$.

As $V$ is a finite dimensional vector space, we can use finite dimensional degree theory and consider the degree $\deg(F_{t,V}, V \cap \overline{B}_X(0,r), p)$, that is the degree of the map $F_{t,V} : V \cap \overline{B}_X(0,r) \to V$ at the point $p$. We recall that when $\Omega \subset \mathbb{R}^d$ is open and bounded, $F : \overline{\Omega} \to \mathbb{R}^d$ is a $C^1$-smooth function and $p \in \mathbb{R}^d$ are such that $p \notin f(\partial\Omega)$ and for all $x \in f^{-1}(p)$ the derivative $Df(x)$ is an invertible matrix, the degree is defined to be

$$\deg(f, \Omega, p) = \sum_{x \in f^{-1}(p)} \mathrm{sgn}(\det(Df(x))),$$

where $\mathrm{sgn}(r)$ is sign of a real number $r$. Also, the map $f \to \deg(h, \Omega, p)$ is defined for a continuous function $h : \overline{\Omega} \to \mathbb{R}^d$ by approximating $h$ by $C^1$-smooth function, see [43], Definition 1.2.5 on details. Let us denote $\overline{B}_W(0,r) = W \cap \overline{B}_X(0,r)$.

Let $r > 0$. Recall that by assumption (A), for $F_0 = Id : X \to X$ and $F_1 = B_e : X \to X$ there are finite dimensional spaces $V_0 \in S_0(X)$ and $V_1 \in S_0(X)$ such that $e \in V_0$ and $e \in V_1$ and that when $V \in S_0(X)$ satisfies $V_0 \subset V$ and $V_1 \subset V$, then

$$\sup_{x \in \overline{B}_X(0,r) \cap V} \|F_{0,V}(x) - F_0(x)\|_X < r/4, \tag{43}$$

and

$$\sup_{x \in \overline{B}_X(0,r) \cap V} \|F_{1,V}(x) - F_1(x)\|_X < r/4. \tag{44}$$

Moreover, let $W \in S_0(X)$ be such a finite dimensional linear space that $V_0 \subset W$ and $V_1 \subset W$. Observe that as $e \in W$, we can decompose $W$ as

$$W = \mathrm{span}(e) \oplus \{x \in W : \langle x, e\rangle_X = 0\}. \tag{45}$$

and denote $W' = \{x \in W : \langle x, e\rangle_X = 0\}$. We see that

$$A_1 = B_e : \mathrm{span}(e) \to \mathrm{span}(e), \ B_e|_{\mathrm{span}(e)} = -Id, \tag{46}$$
$$A_1 = B_e : W' \to W', \ B_e|_{W'} = Id.$$

Next we use the facts that $F_0(0) = Id(0) = 0$ and $F_1(0) = B_e(0) = 0$. Let us define the maps

$$f_0 = F_0|_{\overline{B}_W(0,r)}, \quad f_1 = F_1|_{\overline{B}_W(0,r)}, \quad f_{0,W} = F_{0,W}|_{\overline{B}_W(0,r)}, \quad f_{1,W} = F_{1,W}|_{\overline{B}_W(0,r)}.$$

that are $C^1$-smooth maps $\overline{B}_W(0,r) \to W$. Moreover, for $j = 0, 1$, we have $f_0(\partial B_W(0,r)) = \partial B_W(0,r)$ and $f_1(\partial B_W(0,r)) = \partial B_W(0,r)$. Let $p_j := f_{j,W}(0)$. Then, by (43) and (44) we have

$$\|p_j\|_W = \|f_{j,W}(0)\|_W = \|f_{j,W}(0) - f_j(0)\|_W < \frac{1}{4}r, \tag{47}$$

and

$$\text{dist}(f_{j,W}(x), p_j) \geq \frac{1}{2}r, \quad \text{for all } x \in \partial B_W(0, r).$$

This implies that

$$\sup_{x \in \overline{B}_W(0,r)} \|f_{j,W}(x) - f_j(x)\| < \frac{1}{4}r < \frac{1}{2}r \leq \text{dist}(f_{j,W}(\partial B_W(0, r)), p_j). \tag{48}$$

Then, by [43], Definition 1.2.5, the formula (48) implies that

$$\deg(f_{j,W}, \overline{B}_W(0, r), p_j) = \deg(f_j, \overline{B}_W(0, r), p_j).$$

Moreover, as $F_1 = A_1 = B_e$ satisfies (46), and $p_j \in \overline{B}_W(0, r)$, we have

$$\deg(f_0, \overline{B}_W(0, r), p_0) = \deg(Id, \overline{B}_W(0, r), p_0) = \text{sgn}(\det(Id)) = 1,$$

and

$$\deg(f_1, \overline{B}_W(0, r), p_1) = \deg(A_1, \overline{B}_W(0, r), p_1) = \text{sgn}(\det(B_e|_W)) = -1.$$

Moreover, by our assumptions on $\mathcal{B}$, the discretized maps $F_{j,W} : W \to W$, $j = 0, 1$ are $C^1$-diffeomorphism. Hence, the inverse image $F_{j,W}^{-1}(\{p_j\})$ is a set containing only one point that coincides with $f_{j,W}^{-1}(\{p_j\})$ (that in fact is the set $\{0\}$). This and the above show that for any $R > r$, we have

$$\begin{aligned}
\deg(F_{j,W}, \overline{B}_W(0, R), p_j) &= \deg(F_{j,W}, \overline{B}_W(0, r), p_j) & (49) \\
&= \deg(f_{j,W}, \overline{B}_W(0, r), p_j) & (50) \\
&= \deg(f_j, \overline{B}_W(0, r), p_j) \\
&= (-1)^j.
\end{aligned}$$

Let us now consider the maps $F_{t,W} : W \to W$ where $t \in [0, 1]$, that are $C^1$-diffeomorphism $f_{t,W} : W \to W$. As $t \to F_t = A_t$ is a continuous map $[0, 1] \to GL(X)$ and $\mathcal{A} : \mathcal{D} \to \mathcal{B}$ is a continuous functor, the map $t \to F_{t,W}$ is a continuous map $[0, 1] \to C(\overline{B}_W(0, r); W)$.

Let us consider $\hat{t} \in [0, 1]$ and denote $p_t = F_{t,W}(0)$. As $F_{\hat{t},W} : W \to W$ is a $C^1$-diffeomorphism, the set $F_{\hat{t},W}(B_W(0, r))$ is open and hence there is $\epsilon > 0$ such that

$$B_W(p_{\hat{t}}, 5\epsilon) \subset F_{\hat{t},W}(B_W(0, r)). \tag{51}$$

This implies that

$$\text{dist}_W(p_{\hat{t}}, F_{\hat{t},W}(\partial B_W(0, r))) > 5\epsilon. \tag{52}$$

As $t \to F_{t,W}$ is a continuous map $[0, 1] \to C(\overline{B}_W(0, r); W)$, there is $\delta > 0$ such that when $|t - \hat{t}| < \delta$ then

$$\sup_{x \in V \cap \overline{B}_X(0,r)} \|F_{t,V}(x) - F_{\hat{t},V}(x)\|_X < \epsilon. \tag{53}$$

In particular, this implies that

$$\|F_{t,V}(0) - F_{\hat{t},V}(0)\|_X < \epsilon, \tag{54}$$

and thus $p_t = F_{t,V}(0)$ and $p_{\hat{t}} = F_{\hat{t},V}(0)$ are in the ball $B_W(p_{\hat{t}}, 5\epsilon)$ and hence by (51) these points are in the same topological component of by $W \setminus F_{\hat{t},W}(\partial B_W(0, r))$. Hence, it follows from [43], Theorem 1.2.6 (5), that for $|t - \hat{t}| < \delta$, we have

$$\deg(F_{\hat{t},W}, \overline{B}_W(0, r), p_t) = \deg(F_{\hat{t},W}, \overline{B}_W(0, r), p_{\hat{t}}). \tag{55}$$

Next, we observe that by (53), for any $t_1, t_2 \in [0, 1]$ satisfying $|t_1 - \hat{t}| < \delta$ and $|t_2 - \hat{t}| < \delta$,

$$\sup_{x \in V \cap \overline{B}_X(0,r)} \|F_{t_1,V}(x) - F_{t_2,V}(x)\|_X < 2\epsilon. \tag{56}$$

Inequalities (52), (54) and (56) imply that for any $t_1, t_2 \in [0, 1]$ satisfying $|t_1 - \hat{t}| < \delta$ and $|t_2 - \hat{t}| < \delta$ we have

$$
\begin{aligned}
\operatorname{dist}_W(p_{t_1}, F_{t_2,W}(\partial B_W(0, r))) &= \inf_{y \in \partial B_W(0,r)} \operatorname{dist}_W(p_{t_1}, F_{t_2,W}(y)) \\
&> \inf_{y \in \partial B_W(0,r)} \operatorname{dist}_W(p_{t_1}, F_{\hat{t},W}(y)) - \epsilon \\
&> \inf_{y \in \partial B_W(0,r)} \operatorname{dist}_W(p_{\hat{t}}, F_{\hat{t},W}(y)) - 2\epsilon - \epsilon \\
&= \operatorname{dist}_W(p_{\hat{t}}, F_{\hat{t},W}(\partial B_W(0, r))) - 3\epsilon - \epsilon > 5\epsilon - 3\epsilon = 2\epsilon.
\end{aligned}
\tag{57}
$$

As $t \to F_{t,W}$ is a continuous map $[0, 1] \to C(\overline{B}_W(0, r); W)$ and the formula (57) is valid, it follows from [43], Theorem 1.2.6 (3) that for $t_1$ and $t_2$ satisfying $|t_1 - \hat{t}| < \delta$ and $|t_2 - \hat{t}| < \delta$, we have

$$
\deg(F_{t_1,W}, \overline{B}_W(0, r), p_{t_2}) = \deg(F_{\hat{t},W}, \overline{B}_W(0, r), p_{t_2}).
\tag{58}
$$

This and (55) imply that

$$
\begin{aligned}
\deg(F_{t_1,W}, \overline{B}_W(0, r), p_{t_2}) &= \deg(F_{\hat{t},W}, \overline{B}_W(0, r), p_{t_2}) \\
&= \deg(F_{\hat{t},W}, \overline{B}_W(0, r), p_{\hat{t}}),
\end{aligned}
\tag{59}
$$

In particular, when $t_1 = t_2$ satisfy $|t_1 - \hat{t}| < \delta$, this implies

$$
\deg(F_{t_1,W}, \overline{B}_W(0, r), p_{t_1}) = \deg(F_{\hat{t},W}, \overline{B}_W(0, r), p_{\hat{t}}).
\tag{60}
$$

As $\hat{t} \in [0, 1]$ is above arbitrary, this implies that the function

$$
g : t \to \deg(F_{t,W}, \overline{B}_W(0, r), F_{t,W}(0)),
\tag{61}
$$

is a continuous integer valued function on $[0, 1]$ and thus it is constant on the interval $t \in [0, 1]$. However, by (49), $g(0) = 1$ is not equal to $g(1) = -1$ and hence $g(t)$ can not be constant function on the interval $t \in [0, 1]$. This contradiction shows that the required functor does not exists. $\qquad \square$

## A.5 Proofs from Sec. 3.2

### A.5.1 Proof of Lemma 1

*Proof.* Let $F : X \to X$ be strongly monotone and $C^1$-diffeomorphism. It is obvious that strongly monotonicity implies that strictly monotonicity.

We first show that the strongly monotonicity implies that the coercivity. Indeed, by the strongly monotonicity we have

$$
\langle F(x) - F(0), x - 0 \rangle_X \geq \alpha \|x\|_X^2,
$$

which implies that, as $\|x\|_X \to \infty$,

$$
\langle F(x), \frac{x}{\|x\|} \rangle_X \geq \langle F(0), \frac{x}{\|x\|} \rangle_X + \alpha \|x\|_X \to \infty.
$$

Therefore, $F : X \to X$ is coercive.

Let us consider the operator

$$
F_V := P_V F|_V : V \to V.
$$

From the definitions, $F_V : V \to V$ is $C^1$ and strongly monotones, and the (Fréchet) derivative $DF_V|_v$ at $v \in V$ is given by

$$
DF_V|_v = P_V(DF|_v)|_V,
$$

which is linear and continuous operator from $V$ to $V$. Since $F_V : V \to V$ is $C^1$ and strongly monotone, it is hemicontinuous, strictly monotones, and coercive. By the Minty-Browder theorem [10, Theorem 9.14-1], $F_V : V \to V$ is bijective, and then its inverse $F_V^{-1} : V \to V$ exists.

Next, let $v \in V$. As $F_V : V \to V$ is strongly monotones, we estimate that for all $h \in V$,

$$
\begin{aligned}
\langle \frac{F_V(v + \epsilon h) - F(x)}{\epsilon}, \frac{(x + \epsilon h) - x}{\epsilon} \rangle_X &= \frac{1}{\epsilon^2} \langle F(x + \epsilon h) - F(x), (x + \epsilon h) - x \rangle_X \\
&\geq \frac{\alpha}{\epsilon^2} \|(x + \epsilon h) - x\|_X^2 = \alpha \|h\|_X^2.
\end{aligned}
$$

Taking $\epsilon \to +0$, we have that
$$\langle DF_V|_v(h), h\rangle_X \geq \|h\|_X^2.$$
Therefore, $DF_V|_v : V \to V$ is injective for all $v \in V$. Then, it is bijective because $V$ is now finite dimensional. By the inverse function theorem, the inverse $F_V^{-1} : V \to V$ is $C^1$. $\qquad\square$

### A.5.2 Additional examples

Let us consider the elliptic (but not stongly elliptic) problem
$$P_s u := -\frac{d}{dt}\left(\text{sign}(t-s)\frac{d}{dt}u(t)\right) = f(t), \quad t \in [0,1], \tag{62}$$
with the boundary conditions
$$u(0) = 0, \quad \frac{d}{dt}u(t)\Big|_{t=1} = 0. \tag{63}$$
The FEM (Finite Element Method) matrix corresponding to this problem is $[p_{jk}(s)]_{j,k=1}^n$, where
$$p_{jk}(s) = \int_0^1 \text{sign}(t-s)\frac{d}{dt}\chi_j(t) \cdot \frac{d}{dt}\chi_k(t)dt, \quad j,k = 1,\ldots,m$$
and $\chi_j(t)$, $j = 1,2,\ldots$ are a complete basis of the Hilbert space $H_{D,N}^1(0,1) := \{v \in H^1(0,1) : v(0) = 0\}$ that correspond to the mixed Dirichlet and Neumann boundary conditions. Note that in the FEM approximation we use only a finite subset of the complete basis of the Hilbert space. Note for $u$ in the canonical domain of the above problem that makes the operator appearing in the above equation selfadjoint (the Friedrichs extension), that is, for
$$u \in \left\{v \in H^1(0,1) : \text{sign}(t-s)\frac{d}{dt}v(t) \in H^1(0,1), \ v(0) = 0, \ \frac{d}{dt}v(t)|_{t=1} = 0\right\},$$
we have
$$P_s u = -D_t^{(2)} A_s D_t^{(1)} u,$$
where $A_s$ is the multiplication with the function $\text{sign}(t-s)$, see (37), and $D_t^{(1)}$ and $D_t^{(2)}$ are the derivative operators $D_t^{(j)}u = \frac{d}{dt}u$ having the different domains
$$\mathcal{D}(D_t^{(1)}) = H_{D,N}^1(0,1) := \{v \in H^1(0,1) : v(0) = 0\}, \tag{64}$$
$$\mathcal{D}(D_t^{(2)}) = H_{N,D}^1(0,1) := \{v \in H^1(0,1) : v(1) = 0\}. \tag{65}$$
Observe that $D_t^{(1)}$ and $D_t^{(2)}$ have the inverse operators
$$(D_t^{(1)})^{-1}v(t) = \int_0^t v(t')dt', \tag{66}$$
$$(D_t^{(2)})^{-1}v(t) = -\int_t^1 v(t')dt', \tag{67}$$
that map $(D_t^{(j)})^{-1} : L^2(0,1) \to \mathcal{D}(D_t^{(j)})$. The eigenvalues of the matrix $s \to [p_{jk}(s)]_{j,k=1}^n$ change from positive values to negative values when $s$ moves from 0 to 1. Thus, for some value $s \in (0,1)$ the matrix $[p_{jk}(s)]_{j,k=1}^n$ has a zero eigenvalue and is no invertible even though all maps $D_t^{(1)}$, $D_t^{(2)}$, and $A_s : L^2(0,1) \to L^2(0,1)$ are invertible. In particular, there are no finite FEM basis in where the Galerkin discretizations of all operators $P_s$, $s \in [0,1]$ are invertible.

Let us consider even simpler example: Similarly to the above, if we consider the Galerkin discretizations of the operator $A_s : L^2(0,1) \to L^2(0,1)$, we see that the Galerkin matrix corresponding to $A_s : L^2(0,1) \to L^2(0,1)$ is $[a_{jk}(s)]_{j,k=1}^n$, where
$$a_{jk}(s) = \int_0^1 \text{sign}(t-s)\psi_j(t) \cdot \psi_k(t)dt, \quad j,k = 1,\ldots,m,$$
and $\psi_j(t)$, $j = 1,2,\ldots$ are a complete basis of the Hilbert space $L^2(0,1)$. Again, we see that the eigenvalues of the matrix $s \to [a_{jk}(s)]_{j,k=1}^n$ change from positive values to negative values when $s$ moves from 0 to 1. Thus, for some value $s \in (0,1)$ the matrix $[a_{jk}(s)]_{j,k=1}^n$ has a zero eigenvalue and is no invertible even though all maps $A_s : L^2(0,1) \to L^2(0,1)$ are invertible. Thus there are no finite basis in where the Galerkin discretizations of all operators $A_s$, $s \in [0,1]$ are invertible.

## A.6 Proofs from Sec. 3.4

### A.6.1 Proof of Lemma 4

*Proof.* From the assumption, it holds that

$$\||DF|_x - I\|_{X \to X} \leq \|T_1\|_{X \to X} \||DG|_x\|_{X \to X} \|T_2\|_{X \to X} \leq \frac{1}{2}.$$

Then, by the mean value theorem, there is $0 < t < 1$ such that

$$
\begin{aligned}
\langle F(x_1) - F(x_2), x_1 - x_2 \rangle_X &= \frac{\partial}{\partial t} \langle F(x_1 + t(x_2 - x_1)) - F(x_2), x_1 - x_2 \rangle_X \\
&= \frac{\partial}{\partial t} \langle DF \Big|_{x_1 + t(x_2 - x_1)} (x_2 - x_1), x_1 - x_2 \rangle_X \geq \frac{1}{2} \|x_1 - x_2\|_X^2.
\end{aligned}
$$

These imply that $F \colon X \to X$ is strongly monotone. $\qquad\square$

### A.6.2 Proof of Lemma 3

*Proof.* Observe that strongly monotone neural operators are coercive, that is,

$$\lim_{\|u\|_X \to \infty} \langle F(u), \frac{u}{\|u\|_X} \rangle_X = \lim_{\|u\|_X \to \infty} \frac{1}{\|u\|_X} \langle F(u) - F(0), u - 0 \rangle_X = \infty. \tag{68}$$

and therefore $F \colon X \to X$ is surjective by Browder-Minty theorem [28, Thm. 2.1], see also [14] considerations for neural operators. Moreover, the derivatives $DF|_x$ are linear strongly monotone operators for all $x \in X$, and therefore injective. Observe that the derivative is a linear operator of the form

$$DF|_x = I + T_2 \circ DG|_{T_1 x} \circ T_1 : X \to X,$$

and as $T_1$ and $T_2$ are compact and $DG|_{Tx}$ is bounded we see that $DF|_x$ is a Fredholm operator of index zero. Observe that

$$\langle DF|_x v, v \rangle_X = \frac{1}{h} \lim_{h \to 0} \langle \frac{F(x + hv) - F(x)}{h}, (x + hv) - x \rangle_X \geq \alpha \|v\|_X^2,$$

where $h \in \mathbb{R}$ and $v \in X$, and hence $DF|_x : X \to X$ is an injective linear operator. As $DF|_x$ is a Fredholm operator of index zero, this implies that the derivative $DF|_x : X \to X$ is a bijection. As the Hilbert space $X$ can be identified with it dual space and the operator $F \colon X \to X$ is continuous and hence hemi-continuous, it follows from [28], Theorem 3.1, that the map $F \colon X \to X$ is a homeomorphism. As it is $C^1$-smooth and its derivative is bijective operator $DF|_x : X \to X$ at all $x \in X$, it follows that $F \colon X \to X$ is a $C^1$-smooth diffeomorphism. $\qquad\square$

### A.6.3 Proof of Theorem 3

*Proof.* To show the well-definedness of the discretization functor $\mathcal{A}_{\text{lin}}$, we need to show that, for each strongly monotone $C^1$-function $F \colon X \to X$ that is of the form (4), $F_V = P_V F|_V : V \to V$ is still strongly monotone $C^1$-smooth of the form (4). This is given by Lemma 1. Moreover, such functions are $C^1$-smooth diffeomorphisms.

To verify assumption (A), let $r, \epsilon > 0$, and let $F \colon X \to X$ be a strongly monotone diffeomorphisms that is of the form $F = Id + T_2 \circ G \circ T_1 : X \to X$ where $T_1 \colon X \to X$ and $T_2 \colon X \to X$ are compact linear operators, and $G \colon X \to X$ is such that $G \in C^1(X)$. Since $T_2 \circ G \circ T_1$ is a compact mapping, there is a finite-dimensional subspace $V \subset X$, depending on $\epsilon > 0$ such that

$$\sup_{x \in \overline{B}_X(0,r)} \|(Id - P_V) T_2 G(T_1 x)\|_X \leq \epsilon,$$

which implies that

$$\sup_{x \in \overline{B}_X(0,r) \cap V} \|F_V(x) - F(x)\|_X = \sup_{x \in \overline{B}_X(0,r) \cap V} \|(Id - P_V) T_2 G(T_1 x)\|_X \leq \epsilon.$$

To prove the continuity in the sense of Definition 8 let $r > 0$ and let $V \in S_0(X)$. Assume that

$$\lim_{j \to \infty} \sup_{x \in \overline{B}_X(0,r)} \|F^{(j)}(x) - F(x)\|_X = 0,$$

where $F^{(j)}$ and $F$ are strongly monotone diffeomorphisms $F: X \to X$ that are of the form (4). Then, we see that, as $j \to \infty$,

$$\sup_{x \in V \cap \overline{B}_X(0,r)} \|F_V^{(j)}(x) - F_V(x)\|_X \leq \sup_{x \in V \cap \overline{B}_X(0,r)} \|P_V F^{(j)}(x) - P_V F(x)\|_X$$

$$\leq \sup_{x \in V \cap \overline{B}_X(0,r)} \|P_V\|_{\mathrm{op}} \|F^{(j)}(x) - F(x)\|_X \leq \sup_{x \in \overline{B}_X(0,r)} \|F^{(j)}(x) - F(x)\|_X \to 0.$$

$\square$

### A.7  Proofs from Sec. 3.5

#### A.7.1  Proof of Theorem 4

In the proof of Theorem 4, we prove first few auxiliary lemmas

**Lemma 5.** *A layer of a neural operator $F: X \to X$ is surjective. In particular, if $F: X \to X$ is bilipschitz, then $F: X \to X$ is a homeomorphism.*

*Proof.* By formula (4) a layer of neural operator $F: X \to X$ is of the form $F(u) = u + T_2 G(T_1 u)$ where $T_1 : X \to X$ and $T_2 : X \to X$ are compact linear operators and $G : X \to X$ is a function in $C^1(X)$. Let $c_0, c_1 > 0$ be such that $\|G\|_{L^\infty(X)} \leq c_0$ and $\mathrm{Lip}_{X \to X}(G) \leq c_1$.

Let $p \in X$ and $R_0 > \|T_2\|_{X \to X} c_0 + \|p\|_X$. Then $K_p : X \to X$, defined by the formula

$$K_p(x) = -T_2 \circ G \circ T_1(x) + p, \quad x \in X,$$

is a compact non-linear operator, see [43, Definition 2.1.11].

When $p$ satisfies $p \notin (Id - K_0)(\partial B(0, R_0))$, let $\deg(Id - K_0, B(0, R_0), p) = \deg(Id - K_p, B(0, R_0), 0)$ be the infinite dimensional (Leray-Schauder) degree of the operator $Id - K_0 : X \to X$ in the set $B(0, R_0)$ with respect to the point $p$, see [43, Definition 2.2.3]. Let $K_{p;t} : X \to X$ be the non-linear compact operators that depend on the parameter $t \in [0, 1]$, obtained by multiplying $K_p(x)$ by the number $t$, that is,

$$K_{p;t}(x) = t K_p(x), \quad x \in X. \tag{69}$$

As $\|K_{p;t}(x)\|_X \leq \|T_2\|_{X \to X} c_0 + \|p\|_X < R_0$ for all $t \in [0, 1]$ and $x \in X$, we see that

$$K_{p;t}(x) \neq x, \quad \text{for } x \in \partial B_X(R_0). \tag{70}$$

Then by the homotopy invariance of the Leray-Schauder degree, see [43, Theorem 2.2.4(3)], the function

$$d(t) = \deg(Id - K_{p;t}, B(0, R_0), 0), \quad t \in [0, 1], \tag{71}$$

is a constant function. Moreover, by [43, Theorem 2.2.4(1)],

$$\deg(Id - K_p, B(0, R_0), 0) = \deg(I - K_{p;1}, B(0, R_0), 0) = d(1)$$
$$= d(0) = \deg(I, B(0, R_0), 0) = 1.$$

By [43, Theorem 2.2.4(2)], this implies that the equation

$$x = K_p(x), \quad \text{or equivalently,} \quad x + T_2 \circ G \circ T_1(x) = p,$$

has a solution $x \in B_X(R_0)$. As $p \in X$ was above arbitrary, this implies that $F = Id + T_2 \circ G \circ T_1 : X \to X$ is surjection.

Finally, if $F: X \to X$ is bilipschitz, it is a bijection and its inverse function is Lipschitz function, and thus $F: X \to X$ is a homeomorphism. $\square$

The next lemma shows the existence of a (finite-dimensional) orthogonal subspace so that for each compact operator, projection onto the subspace is a perturbation of the identity under either pre or post composition.

**Lemma 6.** *Let $T_1, T_2 : X \to X$ be compact operators and $h > 0$. There is a finite dimensional space $W \subset X$ such that for the orthogonal projector $P_W : X \to X$ it holds that*

$$\|T_1(Id - P_W)\|_{X \to X} < h, \tag{72}$$
$$\|(Id - P_W)T_2\|_{X \to X} < h, \tag{73}$$
$$\|(Id - P_W)T_1(Id - P_W)\|_{X \to X} < h, \tag{74}$$
$$\|(Id - P_W)T_2(Id - P_W)\|_{X \to X} < h. \tag{75}$$

*Proof.* We use the singular value decomposition of compact operators: We can write

$$T_\ell x = \sum_{p=1}^{\infty} \omega_{\ell,p} \langle x, \psi_{\ell,p} \rangle \phi_{1,p},$$

where $\psi_{\ell,p}$ and $\phi_{\ell,p}$ are orthogonal families in $X$ and $\omega_{\ell,p} \geq 0$ satisfy $\omega_{\ell,p+1} \leq \omega_{\ell,p}$ and $\omega_{\ell,p} \to 0$ as $p \to \infty$. For all $h > 0$ we can choose $P > 0$ such that $\omega_{\ell,p} < h$ when $p \geq P$. Then

$$\|T_\ell - \sum_{p=1}^{P} \omega_{\ell,p} \langle \cdot, \psi_{\ell,p} \rangle \phi_{1,p}\|_{X \to X} < h.$$

If

$$V = V_P = \operatorname{span}\{\psi_{\ell,p}, \phi_{\ell,p}; \ \ell = 1, 2, \ p \leq P\}, \tag{76}$$

we see that if $W \subset X$ is a linear subspace such that $V \subset W$ then

$$(\sum_{p=1}^{P} \omega_{\ell,p} \langle \cdot, \psi_{\ell,p} \rangle \phi_{1,p}) \circ P_W = \sum_{p=1}^{P} \omega_{\ell,p} \langle \cdot, \psi_{\ell,p} \rangle \phi_{1,p},$$

$$P_W \circ (\sum_{p=1}^{P} \omega_{\ell,p} \langle \cdot, \psi_{\ell,p} \rangle \phi_{1,p}) = \sum_{p=1}^{P} \omega_{\ell,p} \langle \cdot, \psi_{\ell,p} \rangle \phi_{1,p},$$

and

$$P_W \circ (\sum_{p=1}^{P} \omega_{\ell,p} \langle \cdot, \psi_{\ell,p} \rangle \phi_{1,p}) \circ P_W = \sum_{p=1}^{P} \omega_{\ell,p} \langle \cdot, \psi_{\ell,p} \rangle \phi_{1,p},$$

and thus

$$\|T_\ell - T_\ell \circ P_W\|_{X \to X} < h, \quad \|T_\ell - P_W \circ T_\ell\|_{X \to X} < h,$$

and moreover,

$$\|T_\ell - P_W \circ T_\ell \circ P_W\|_{X \to X} < h.$$

$\square$

Let

$$F^W = Id + P_W \circ T_2 \circ G \circ T_1 \circ P_W \ : X \to X. \tag{77}$$

Observe that for $w \in W$ and $v \in W^\perp$ we have

$$F^W(w + v) = F^W(w) + v, \quad F^W(w) \in W, \tag{78}$$

and

$$F^W(W) \subset W, \tag{79}$$
$$F^W : W^\perp \to W^\perp, \quad F^W|_{W^\perp} = Id_{W^\perp}. \tag{80}$$

This means that $F^W = Id_{W^\perp} \oplus F^W|_W$, where $F^W|_W : W \to W$ is a function which maps the finite dimensional vector space $W$ to itself.

The lemma below shows that given an $F$ and $\epsilon > 0$, we may perturb it by a Lipschitz term $B$ so it becomes the operator $F^W$.

**Lemma 7.** *For any $\epsilon > 0$, there is a finite dimensional space $W \subset X$ such that $\mathrm{Lip}_{X \to X}(F - F^W) < \epsilon$ and for any ball $B_X(R) \subset X$, $R > 0$, the maps $F: B_X(R) \to X$ and $F^W : B_X(R) \to X$ satisfy $\|F - F^W\|_{L^\infty(B_X(R))} < \frac{1}{2}(1 + R)\epsilon$.*

*Proof.* Let us choose a finite dimensional space $W \subset X$ so that Lemma 6 is valid with

$$h = \frac{1}{4(1 + \|G\|_{C^1(X)}))(1 + \|T_1\|_{X \to X})(1 + \|T_2\|_{X \to X})}\epsilon.$$

The right hand side of (72) is chosen so that

$$
\begin{aligned}
\mathrm{Lip}_{X \to X}(F - F^W) &= \mathrm{Lip}(P_W \circ T_2 \circ G \circ T_1 \circ P_W(x) - T_2 \circ G \circ T_1) \\
&\leq \mathrm{Lip}(P_W \circ T_2 \circ G \circ T_1 \circ P_W - P_W \circ T_2 \circ G \circ T_1(x)) \\
&\quad + \mathrm{Lip}(P_W \circ T_2 \circ G \circ T_1(x) - T_2 \circ G \circ T_1) \\
&\leq \|T_2\|_{X \to X}\mathrm{Lip}(G|_{X \to X})\|T_1(I - P_W)\|_{X \to X} + \|(I - P_W)T_2\|_{X \to X}\mathrm{Lip}(G|_{X \to X})\|T_1\|_{X \to X} \\
&\leq \frac{1}{2}\varepsilon,
\end{aligned}
$$

and

$$
\begin{aligned}
\|F - F^W\|_{L^\infty(B(0,R))} &= \sup_{x \in B(0,R)} \|P_W \circ T_2 \circ G \circ T_1 \circ P_W(x) - T_2 \circ G \circ T_1(x)\|_X \\
&\leq \sup_{x \in B(0,R)} \|P_W \circ T_2 \circ G \circ T_1 \circ P_W(x) - P_W \circ T_2 \circ G \circ T_1(x)\|_X \\
&\quad + \sup_{x \in B(0,R)} \|P_W \circ T_2 \circ G \circ T_1(x) - T_2 \circ G \circ T_1(x)\|_X \\
&\leq \|T_2\|_{X \to X}\mathrm{Lip}(G|_{B(0,\|T_1\|_{X \to X}R)})\|T_1(I - P_W)\|_{X \to X} \cdot R \\
&\quad + \|(I - P_W)T_2\|_{X \to X}\|G\|_{L^\infty(B(0,\|T_1\|_{X \to X}R))} \\
&\leq \frac{1}{4}(R + 1)\epsilon.
\end{aligned}
$$

$\square$

**Lemma 8.** *For any $\epsilon > 0$, there are finite dimensional space $W \subset X$ and (possibly non-linear) functions $B : X \to X$ and $\tilde{B} : X \to X$ such that*

$$\mathrm{Lip}(B) < \epsilon, \quad \mathrm{Lip}(\tilde{B}) < \epsilon. \tag{81}$$

*and*

$$F^W = (Id + B) \circ F : X \to X, \tag{82}$$

$$F = (Id + \tilde{B}) \circ F^W : X \to X. \tag{83}$$

*Moreover, $B : X \to X$ is a compact non-linear operator of the form*

$$B = P_W \circ F_1 \circ P_W + T_2 \circ F_2 \circ P_W + P_W \circ F_3 \circ T_1 + T_2 \circ F_4 \circ T_1, \tag{84}$$

*where $F_1, F_2, F_3, F_4 : X \to X$ are functions in $C^1(X)$ and $\tilde{B} : X \to X$ is of the same form. In addition, the operator $Id + B : X \to X$ and $Id + \tilde{B} : X \to X$ are layers of neural operators.*

*Proof.* Below, let $C_0 > 0$ be such that

$$\|F(x) - F(y)\|_X \geq C_0\|x - y\|_X, \quad x, y \in X, \tag{85}$$

and let $W$ be the space in Lemma 6 with

$$h = \frac{1}{4(1 + \|G\|_{C^1(X)}))(1 + \|T_1\|_{X \to X})(1 + \|T_2\|_{X \to X})}\epsilon.$$

By Lemma 5, $F(x) = x + T_2 G(T_1(x))$ is by an invertible function $F : X \to X$. Thus we can define a non-linear operator

$$F^W \circ (F)^{-1} \tag{86}$$

$$= (Id + P_W \circ T_2 \circ G \circ T_1 \circ P_W) \circ (Id + T_2 \circ G \circ T_1)^{-1}$$

$$= Id + (P_W \circ T_2 \circ G \circ T_1 \circ P_W - T_2 \circ G \circ T_1) \circ (Id + T_2 \circ G \circ T_1)^{-1}.$$

Let

$$R = P_W \circ T_2 \circ G \circ T_1 \circ P_W - T_2 \circ G \circ T_1 \tag{87}$$

$$= \left( P_W \circ T_2 \circ G \circ T_1 \circ P_W - T_2 \circ G \circ T_1 \circ P_W \right) \tag{88}$$

$$+ \left( T_2 \circ G \circ T_1 \circ P_W - T_2 \circ G \circ T_1 \right). \tag{89}$$

Then for all $x, y \in X$

$$\begin{aligned} &\|R(x) - R(y)\|_X \\ \leq\ & \|(Id - P_W)T_2\|_{X \to X} \|G\|_{Lip(X,X)} \|T_1\|_{X \to X} \|x - y\|_X \\ & + \|T_2\|_{X \to X} \|G\|_{Lip(X,X)} \|T_1(Id - P_W)\|_{X \to X} \|x - y\|_X \\ \leq\ & \frac{1}{2C_0} \epsilon \|x - y\|_X. \end{aligned}$$

and thus,

$$\begin{aligned} & \|R \circ (Id + T_2 \circ G \circ T_1)^{-1}(x) - R \circ (Id + T_2 \circ G \circ T_1)^{-1}(y)\|_X \\ \leq\ & \frac{1}{2} \epsilon \|x - y\|_X. \end{aligned}$$

This implies that $F^W : X \to X$ satisfies for all $x, y \in X$

$$\|(F^W \circ (F)^{-1} - Id)(x) - (F^W \circ (F)^{-1} - Id)(y)\|_X \leq \frac{1}{2} \epsilon \|x - y\|_X,$$

and thus

$$\|(F^W \circ (F)^{-1} - (F^W \circ (F)^{-1}(y)\|_X \leq (1 + \frac{1}{2} \epsilon) \|x - y\|_X, \tag{90}$$

and

$$\mathrm{Lip}(F^W : X \to X) \leq (1 + \frac{1}{2} \epsilon) \cdot \mathrm{Lip}(F : X \to X). \tag{91}$$

Moreover, for all $x, y \in X$

$$\|F^W(x) - F^W(y)\|_X \geq (C_0 - \epsilon) \|x - y\|_X \geq \frac{1}{2} C_0 \|x - y\|_X. \tag{92}$$

Let $c_0 > 0$ be such that $\|G\|_{L^\infty(X)} \leq c_0$. Let $R_0 > 0$. As $\|T_2 \circ G\|_{L^\infty(X)} \leq \|T_2\| c_0$, finite dimensional degree theory, see using [43, Theorem 1.2.6], as above that

$$B_W(0, R_0) \subset F^W(B_W(0, R_1)), \tag{93}$$

when $R_1 > R_0 + \|T_2\| c_0 \geq R_0 + \|P_W \circ T_2 \circ G \circ T_1 \circ P_2\|_{L^\infty(X)}$. As $R_0 > 0$ above is arbitrary, formula (93) implies that $F^W : X \to X$ is surjective. Thus, we have shown that $F^W : X \to X$ is a bijective bilipschitz map.

Similarly to the above, by replacing (85) by (92) and changing the roles of $F^W$ and $F$, we see that for $x, y \in X$

$$\begin{aligned} & \|R \circ (I + P_W \circ T_2 \circ G \circ T_1 \circ P_W)^{-1}(x) - R \circ (I + T_2 \circ G \circ T_1)^{-1}(y)\|_X \\ \leq\ & \epsilon \|x - y\|_X, \end{aligned}$$

and

$$\|(F \circ (F^W)^{-1} - Id)(x) - (F \circ (F^W)^{-1} - Id)(y)\|_X \leq \epsilon \|x - y\|_X. \tag{94}$$

The above implies that

$$F^W = (Id + B) \circ F, \tag{95}$$

$$F = (Id + \tilde{B}) \circ F^W, \tag{96}$$

where

$$B = (F^W \circ F^{-1} - Id) : X \to X,$$
$$\tilde{B} = (F \circ (F^W)^{-1} - Id) : X \to X,$$

satisfy

$$Lip(B) < \frac{1}{2}\epsilon, \quad Lip(\tilde{B}) < \epsilon. \tag{97}$$

Next we use that

$$(Id + H)^{-1} = (Id + H)^{-1} \circ (Id + H - H)$$
$$= Id - (Id + H)^{-1} \circ H,$$

so that

$$(Id + T_2 \circ G \circ T_1)^{-1} = Id - (Id + T_2 \circ G \circ T_1)^{-1} \circ (T_2 \circ G \circ T_1).$$

Observe that by (86)

$$B = F^W \circ (F)^{-1} - Id \tag{98}$$
$$= (P_W \circ T_2 \circ G \circ T_1 \circ P_W - T_2 \circ G \circ T_1) \circ (Id + T_2 \circ G \circ T_1)^{-1}$$
$$= (P_W \circ T_2 \circ G \circ T_1 \circ P_W - T_2 \circ G \circ T_1)$$
$$\quad - (P_W \circ T_2 \circ G \circ T_1 \circ P_W - T_2 \circ G \circ T_1) \circ (Id + T_2 \circ G \circ T_1)^{-1} \circ (T_2 \circ G \circ T_1),$$

and as $P_W : X \to X$ is a finite rank operator and $T_2 : W \to W$ is a compact linear operator, we see that $B : X \to X$ is a compact non-linear operator of the form

$$B = P_W \circ F_1 \circ P_W + T_2 \circ F_2 \circ P_W + P_W \circ F_3 \circ T_1 + T_2 \circ F_4 \circ T_1,$$

where $F_1, F_2, F_3, F_4 : X \to X$ are functions in $C^1(X)$.

Moreover, similarly to (86), we see that

$$F \circ (F^W)^{-1} \tag{99}$$
$$= (Id + T_2 \circ G \circ T_1) \circ (Id + P_W \circ T_2 \circ G \circ T_1 \circ P_W)^{-1}$$
$$= Id + (T_2 \circ G \circ T_1 - P_W \circ T_2 \circ G \circ T_1 \circ P_W) \circ (I + P_W \circ T_2 \circ G \circ T_1 \circ P_W)^{-1},$$

and hence

$$\tilde{B} = F \circ (F^W)^{-1} - Id \tag{100}$$
$$= (T_2 \circ G \circ T_1 - P_W \circ T_2 \circ G \circ T_1 \circ P_W) \circ (Id + P_W \circ T_2 \circ G \circ T_1 \circ P_W)^{-1}$$
$$= (T_2 \circ G \circ T_1 - P_W \circ T_2 \circ G \circ T_1 \circ P_W)$$
$$\quad - (T_2 \circ G \circ T_1 - P_W \circ T_2 \circ G \circ T_1 \circ P_W)$$
$$\quad\quad\quad\quad \circ (Id + P_W \circ T_2 \circ G \circ T_1 \circ P_W)^{-1} \circ (P_W \circ T_2 \circ G \circ T_1 \circ P_W),$$

and again, as $P_W : X \to X$ is a finite rank operator and $T_2 : W \to W$ is a compact linear operator, we see that $\tilde{B} : X \to X$ is a compact non-linear operator of the form

$$\tilde{B} = P_W \circ \tilde{F}_1 \circ P_W + T_2 \circ \tilde{F}_2 \circ P_W + P_W \circ \tilde{F}_3 \circ T_1 + T_2 \circ \tilde{F}_4 \circ T_1,$$

where $\tilde{F}_1, \tilde{F}_2, \tilde{F}_3, \tilde{F}_4 : X \to X$ are functions in $C^1(X)$. For an infinite dimensional Hilbert space $X$ there is a linear isomorphism $J : X \to X \times X$, as the cardinality of Hilbert basis of the space $X$ is the same as the cardinality of the Hilbert basis of $X \times X$, see [26, Theorem 3.5]. Then, by writing the isomorphism $J$ as $J(x) = (J_1(x), J_2(x)) \in X \times X$, we see that

$$\tilde{B}(x) = (( \begin{array}{cc} P_W & T_2 \end{array} ) \circ J) \circ (J^{-1} \circ \begin{pmatrix} \tilde{F}_1 & \tilde{F}_3 \\ \tilde{F}_2 & \tilde{F}_4 \end{pmatrix} \circ J) \circ (J^{-1} \circ \begin{pmatrix} P_W \\ T_1 \end{pmatrix} )(x),$$

that is a composition of a linear compact operator $X \to X$, a non-linear operator $X \to X$, and a compact operator $X \to X$. Hence we see that $Id + \tilde{B}$, and similarly $Id + B$, are layers of neural operators. $\qquad\square$

Now we present our Proof of Theorem 4.

*Proof.* Let $F(x) = x + T_2G(T_1(x))$. Without loss of generality, we may assume that $0 < \epsilon < \min(\|T_2\|_{X \to X} C_1 \|T_1\|_{X \to X}, \frac{1}{2})$.

By (94)

$$\|B(x) - B(y)\|_X \leq \epsilon \|x - y\|_X, \tag{101}$$

and

$$\langle (Id + B)(x) - (Id + B)(y), x - y \rangle_X \geq \|x - y\|_X^2 - \epsilon \|x - y\|_X^2$$
$$= (1 - \epsilon) \|x - y\|_X^2, \tag{102}$$

which implies that

$$Id + B : X \to X, \tag{103}$$

is a strongly monotone operator. Similarly, we see that

$$Id + \tilde{B} : X \to X, \tag{104}$$

is a strongly monotone operator.

Recall that $F_W$ maps $W^\perp$ to itself and it is equal to the identity map in $W^\perp$, and moreover $F^W$ can be decomposed according to the formula (78). Thus we can write

$$F_W = Id_{W^\perp} \oplus (P_W \circ F_W \circ P_W) : W^\perp \oplus W \to W^\perp \oplus W.$$

Below, let us identify $W$ with $\mathbb{R}^n$ to clarify notations. We denote

$$f = F_W|_W = P_W \circ F_W \circ P_W : W \to W. \tag{105}$$

By (91) and (92) there are $c_1, c_0 > 0$ such that

$$c_0 |x - y| \leq |f(x) - f(y)| \leq c_1 |x - y|. \tag{106}$$

Define[4] for $0 < t \leq 1$

$$f_t(x) := \frac{1}{t}(f(tx) - f(0)) + tf(0). \tag{107}$$

For $t = 0$, we define

$$f_0(x) := Df|_0 x. \tag{108}$$

where $Df|_y : \mathbb{R}^n \to \mathbb{R}^n$ is the derivative of the map $f$ at the point $y$, that is considered as a linear map (or a matrix), and we denote its value at vector $v$ by $Df|_y(v) = Df|_y v$.

Then $f_1(x) = f(x)$ and

$$|f_t(x) - f_t(y)| = \frac{1}{t} |f(tx) - f(ty)|, \tag{109}$$

and as

$$\frac{1}{t} c_0 |tx - ty| \leq \frac{1}{t} |f(tx) - f(ty)| \leq \frac{1}{t} c_1 |tx - ty|,$$

we have

$$c_0 |x - y| \leq |f_t(x) - f_t(y)| \leq c_1 |x - y|.$$

Below, let

$$R_0 = |f(0)|, \tag{110}$$

and

$$R_1 = c_1 r_1 + R_0, \tag{111}$$

---

[4]The above used homotopy can be replace by a more explicit flow in the Lie group of diffeomorphism, see e.g. [42].

where $r_1$ appears in the claim of Theorem 4. Observe that then $f_0(B_{\mathbb{R}^n}(0,r_1)) \subset B_{\mathbb{R}^n}(0,R_1)$.

As $f \in C^2(\mathbb{R}^n, \mathbb{R}^n)$, we have by Taylor's series (with the remainder written in the integral form)

$$f(tx) = f(0) + Df|_0(tx) + \frac{1}{2} \int_0^t (t-s) Df|_{sx} x\, ds$$

$$= f(0) + tDf|_0 x + \frac{1}{2} t^2 \int_0^1 (1-s) Df|_{tsx} x\, ds. \tag{112}$$

Thus,

$$f_t(x) = Df|_0 x + \frac{1}{2} t \int_0^1 (1-s) Df|_{tsx} x\, ds + tf(0). \tag{113}$$

Observe that for all $x \in X$,

$$\|Df|_x\| \le c_1, \text{ and } \|(Df|_x)^{-1}\| \le c_0^{-1}.$$

By (113),

$$f_t((Df|_0)^{-1}x) - f_t((Df|_0)^{-1}y)$$

$$= x - y + \frac{1}{2} t \int_0^1 (1-s) \left( Df|_{ts\tilde{x}}(Df|_0)^{-1}x - Df|_{ts\tilde{y}}(Df|_0)^{-1}y \right) ds \bigg|_{\tilde{x}=(Df|_0)^{-1}x,\ \tilde{y}=(Df|_0)^{-1}y}$$

$$= x - y + \frac{1}{2} t \int_0^1 (1-s) \left( Df|_{ts\tilde{x}}(Df|_0)^{-1}(x - y) \right) ds \bigg|_{\tilde{x}=(Df|_0)^{-1}x,\ \tilde{y}=(Df|_0)^{-1}y}$$

$$+ \frac{1}{2} t \int_0^1 (1-s) \left( (Df|_{ts\tilde{x}} - Df|_{ts\tilde{y}})(Df|_0)^{-1}y \right) ds \bigg|_{\tilde{x}=(Df|_0)^{-1}x,\ \tilde{y}=(Df|_0)^{-1}y}. \tag{114}$$

Here, for $x, y \in B_{\mathbb{R}^n}(0, R_1)$,

$$\|Df|_{ts\tilde{x}} - Df|_{ts\tilde{y}}\|_{\mathbb{R}^n \to \mathbb{R}^n} \le \|f\|_{C^2(B_{\mathbb{R}^n}(0,R_1))},$$
$$\|Df|_{ts\tilde{x}}\|_{\mathbb{R}^n \to \mathbb{R}^n} \le c_1,$$
$$\|(Df|_0)^{-1}y\|_{\mathbb{R}^n} \le c_0^{-1}\|y\|_{\mathbb{R}^n}.$$

Hence,

$$f_t \circ (Df|_0)^{-1} = Id + Q_{t,0} : \mathbb{R}^n \to \mathbb{R}^n, \tag{115}$$

where

$$\mathrm{Lip}_{B_{\mathbb{R}^n}(0,R_1)}(Q_{t,0}) \le t \frac{1}{2c_0}(c_1 + \|f\|_{C^2(B_{\mathbb{R}^n}(0,R_1))} R_1). \tag{116}$$

Let us choose $t_1 \in (0,1)$ such that

$$t_1 < \frac{2c_0}{c_1 + \|f\|_{C^2(B_X(0,R_1))} R_1} \varepsilon, \tag{117}$$

so that

$$\mathrm{Lip}_{B_{\mathbb{R}^n}(0,R_1)}(Q_{t_1,0}) < \varepsilon. \tag{118}$$

Below, we will denote

$$H_{t_1,0}(x) = f_{t_1}(f_0^{-1}(x)) = f_{t_1}((Df|_0)^{-1}(x)). \tag{119}$$

Next we consider operators $f_t$ with $t > 0$.

Let us next consider $t_2, t_3, \ldots, t_{m+1} \in (0,1]$ such that $t_2 > t_1$, $t_{m+1} = 1$, and $t_{k+1} > t_k$.

We have

$$|f_t(0)| = t|f(0)| = tR_0,$$

and $\text{Lip}(f_t) \leq c_1$. Hence,

$$|f_t(x)| \leq c_1|x| + R_0. \tag{120}$$

Moreover, for $z = f(0)$ we have $f_t^{-1}(z) = 0$ and $|z| = R_0$, and as the $\text{Lip}(f_t^{-1}) \leq c_0^{-1}$,

$$|f_t^{-1}(x)| \leq |f_t^{-1}(x) - f_t^{-1}(z)| + |f_t^{-1}(z)| \leq c_0^{-1}|x - z| + 0 \leq c_0^{-1}|x| + c_0^{-1}|z|, \tag{121}$$

so that

$$|f_t^{-1}(x)| \leq c_0^{-1}|x| + c_0^{-1}R_0. \tag{122}$$

Also, as we can write

$$f_t(x) := \frac{1}{t}(f(tx) - f(0)) + tf(0) = \frac{1}{t}f(tx) + (t - \frac{1}{t})f(0), \tag{123}$$

we have for $t_k, t_{k+1} \in (0, 1]$, $k \geq 1$,

$$(f_{t_{k+1}} - f_{t_k})(x) = \frac{1}{t_{k+1}}f(t_{k+1}x) - \frac{1}{t_k}f(t_kx) - (\frac{1}{t_{k+1}} - \frac{1}{t_k})f(0) + (t_{k+1} - t_k)f(0)$$

$$= (\frac{1}{t_{k+1}} - \frac{1}{t_k})f(t_{k+1}x) + \frac{1}{t_k}(f(t_{k+1}x) - f(t_kx))$$

$$- (\frac{1}{t_{k+1}} - \frac{1}{t_k})f(0) + (t_{k+1} - t_k)f(0)$$

$$= (\frac{t_k - t_{k+1}}{t_k})\frac{1}{t_{k+1}}f(t_{k+1}x) + \frac{1}{t_k}(f(t_{k+1}x) - f(t_kx))$$

$$- (\frac{1}{t_{k+1}} - \frac{1}{t_k})f(0) + (t_{k+1} - t_k)f(0)$$

$$= (\frac{t_k - t_{k+1}}{t_k})\left(\frac{1}{t_{k+1}}f(t_{k+1}x) + (t_{k+1} - \frac{1}{t_{k+1}})f(0)\right) + \frac{1}{t_k}(f(t_{k+1}x) - f(t_kx))$$

$$- (\frac{t_k - t_{k+1}}{t_k})(t_{k+1} - \frac{1}{t_{k+1}})f(0) - (\frac{1}{t_{k+1}} - \frac{1}{t_k})f(0) + (t_{k+1} - t_k)f(0)$$

$$= (\frac{t_k - t_{k+1}}{t_k})f_{t_{k+1}}(x) + \frac{1}{t_k}(f(t_{k+1}x) - f(t_kx)) + C_k,$$

where

$$C_k = -(\frac{t_k - t_{k+1}}{t_k})(t_{k+1} - \frac{1}{t_{k+1}})f(0) - (\frac{1}{t_{k+1}} - \frac{1}{t_k})f(0) + (t_{k+1} - t_k)f(0) \tag{124}$$

$$= -(\frac{t_k - t_{k+1}}{t_k})(t_{k+1} - \frac{1}{t_{k+1}})f(0) + \frac{t_{k+1} - t_k}{t_kt_{k+1}}f(0) + (t_{k+1} - t_k)f(0) \tag{125}$$

$$= (t_{k+1} - t_k)\left(\frac{1}{t_k} + 1\right)f(0). \tag{126}$$

To analyze the above, we denote

$$k(x; t_k, t_{k+1}) := \frac{1}{t_k}(f(t_{k+1}x) - f(t_kx))$$

$$= \frac{1}{t_k}\int_{t_k}^{t_{k+1}} Df|_{t_k+s'}(x)ds'$$

$$= \frac{1}{t_k}\int_0^1 (t_{k+1} - t_k)Df|_{t_k+s(t_{k+1}-t_k)}(x)ds.$$

As $\|Df|_{t_k+s(t_{k+1}-t_k)}(x)\| \leq c_1\|x\|$, we have for all $R > 0$

$$\text{Lip}(k(\cdot; t_k, t_{k+1}) : B_{\mathbb{R}^n}(0, R) \to \mathbb{R}^n) \leq c_1\frac{t_{k+1} - t_k}{t_k}R.$$

and

$$|k(x; t_k, t_{k+1})| \leq |t_{k+1} - t_k| \cdot \frac{c_1}{t_k}|x|. \tag{127}$$

Moreover, for $x_1, x_2 \in B_{\mathbb{R}^n}(0, R)$,

$$|(f_{t_{k+1}} - f_{t_k})(x_2) - (f_{t_{k+1}} - f_{t_k})(x_1)| \le c_1 \frac{t_k - t_{k+1}}{t_k}|x_2 - x_1| + c_1 \frac{t_{k+1} - t_k}{t_k}R|x_2 - x_1|, \tag{128}$$

and for all $x \in \mathbb{R}^n$,

$$|(f_{t_{k+1}} - f_{t_k})(x)| \le \frac{|t_k - t_{k+1}|}{t_k}|f_{t_{k+1}}(x)| + |k(x; t_k, t_{k+1})| + |C_k| \tag{129}$$

$$\le \frac{|t_k - t_{k+1}|}{t_k}(c_1|x| + R_0) + |t_{k+1} - t_k| \cdot \frac{c_1}{t_k}|x| + |t_{k+1} - t_k|\left(\frac{1}{t_k} + 1\right)R_0 \tag{130}$$

$$\le \frac{|t_k - t_{k+1}|}{t_k}(2c_1|x| + 3R_0). \tag{131}$$

Moreover,

$$f_{t_{k+1}}(f_{t_k}^{-1}(x)) = f_{t_k}(f_{t_k}^{-1}(x)) + (f_{t_{k+1}} - f_{t_k}) \circ (f_{t_k}^{-1}(x)) \tag{132}$$

$$= x + (f_{t_{k+1}} - f_{t_k}) \circ f_{t_k}^{-1}(x). \tag{133}$$

Let

$$Q_{t_{k+1},t_k}(x) = f_{t_{k+1}}(f_{t_k}^{-1}(x)) - x \tag{134}$$

$$= (f_{t_{k+1}} - f_{t_k}) \circ f_{t_k}^{-1}(x). \tag{135}$$

Hence by (128), for $x_1, x_2 \in B_{\mathbb{R}^n}(0, R)$,

$$|Q_{t_{k+1},t_k}(x_2) - Q_{t_{k+1},t_k}(x_1)| \le \frac{1}{c_0}(c_1 \frac{t_k - t_{k+1}}{t_k} + c_1 \frac{t_{k+1} - t_k}{t_k}R)|x_2 - x_1|. \tag{136}$$

By (122) and (129), it hold that of all $x \in \mathbb{R}^n$

$$|Q_{t_{k+1},t_k}(x)| \le \frac{|t_k - t_{k+1}|}{t_k}\left(2c_1(\frac{1}{c_0}|x| + \frac{1}{c_0}R_0) + 3R_0)\right) \tag{137}$$

$$\le \frac{|t_k - t_{k+1}|}{t_k}(\frac{2c_1}{c_0}|x| + (\frac{2c_1}{c_0} + 3)R_0). \tag{138}$$

Let $R_2 \ge R_1$ and define for $k \ge 1$ a function that is a convex combination of the identity map Id and the map $f_{t_{k+1}} \circ f_{t_k}^{-1}$,

$$H_{t_{k+1},t_k}(x) = (1 - \phi(x)x + \phi(x)f_{t_{k+1}}(f_{t_k}^{-1}(x)) \tag{139}$$

$$= x + \phi(x)(f_{t_{k+1}}(f_{t_k}^{-1}(x)) - x) \tag{140}$$

$$= x + \phi(x)Q_{t_{k+1},t_k}(x), \tag{141}$$

where $\phi \in C_0^\infty(\mathbb{R}^n)$ is such that $\phi(x) = 1$ for $|x| \le R_2$, $\phi(x) = 0$ for $|x| \le 2R_2$, and $\|\nabla\phi(x)\| \le 2/R_2$. Then

$$H_{t_{k+1},t_k} = Id + P_{t_{k+1},t_k},$$

where

$$\text{Lip}_{\mathbb{R}^n}(P_{t_{k+1},t_k}) \le \|\phi\|_{L^\infty(B(0,2R_2))} \cdot \text{Lip}_{B(0,2R_2)}(Q_{t_{k+1},t_k}) + \|Q_{t_{k+1},t_k}\|_{L^\infty(B(0,2R_2))}\text{Lip}(\phi)$$

$$\le \frac{1}{c_0}(c_1 \frac{t_{k+1} - t_k}{t_k} + c_1 \frac{t_{k+1} - t_k}{t_k}R_2) \tag{142}$$

$$+ \frac{|t_k - t_{k+1}|}{t_k}(\frac{4c_1}{c_0}R_2 + (\frac{2c_1}{c_0} + 3)R_0)) \cdot \frac{2}{R_2}. \tag{143}$$

For $k \ge 1$ we have $t_k \ge t_1$, and thus

$$\text{Lip}_{\mathbb{R}^n}(P_{t_{k+1},t_k}) \le \frac{|t_{k+1} - t_k|}{t_1}\left(\frac{c_1}{c_0}(10 + R_2) + \frac{6}{R_2}(\frac{c_1}{c_0} + 1)R_0\right). \tag{144}$$

Then, when the condition $f_{t_{k-1}}(x) \in B_{\mathbb{R}^n}(0, R_2)$ it holds, we see that

$$
\begin{aligned}
H_{t_k, t_{k-1}} \circ f_{t_{k-1}}(x) &= f_{t_{k-1}}(x) + \phi(f_{t_{k-1}}(x))(f_{t_k}(f_{t_{k-1}}^{-1}(f_{t_{k-1}}(x))) - f_{t_{k-1}}(x)) \\
&= f_{t_{k-1}}(x) + 1 \cdot (f_{t_k}(f_{t_{k-1}}^{-1}(f_{t_{k-1}}(x))) - f_{t_{k-1}}(x)) \\
&= f_{t_k}(f_{t_{k-1}}^{-1}(f_{t_{k-1}}(x))) \\
&= f_{t_k}(x) \in B_{\mathbb{R}^n}(0, R_2).
\end{aligned}
$$

Observe that when $x \in B_{\mathbb{R}^n}(0, R_0)$ then $f_{t_0}(x) \in B_{\mathbb{R}^n}(0, R_1) \subset B_{\mathbb{R}^n}(0, R_2)$. Next, we denote $t_0 = 0$. Hence, by using induction, we see that for all $j = 1, 2, \ldots, m + 1$ it holds that

$$
\begin{aligned}
H_{t_j, t_{j-1}} \circ H_{t_{j-1}, t_{j-2}} \circ H_{t_1, t_0} \circ f_{t_0}(x) &= f_{t_j}(x), \text{ and} \\
H_{t_j, t_{j-1}} \circ H_{t_{j-1}, t_{j-2}} \circ H_{t_1, t_0} \circ f_{t_0}(x) &= f_{t_j}(x) \in B_{\mathbb{R}^n}(0, R_2).
\end{aligned}
$$

We recall that above we chose $t_1 > 0$ such that

$$
t_1 < \frac{2c_0}{c_1 + \|f\|_{C^2(B_{\mathbb{R}^n}(0, R_1))} R_1} \varepsilon. \tag{145}
$$

We now choose $m$ so that there are $t_k$, $k = 2, 3, \ldots, m + 1$ satisfying $|t_k - t_{k-1}| < \frac{1}{m}$ and

$$
\frac{1}{m} < \varepsilon \frac{t_1}{\left( \frac{c_1}{c_0}(10 + R_2) + \frac{6}{R_2}(\frac{c_1}{c_0} + 1)R_0 \right)}.
$$

This means that we can choose

$$
m > \frac{1}{\varepsilon} \frac{2}{t_1} \left( \frac{c_1}{c_0}(10 + R_2) + \frac{6}{R_2}(\frac{c_1}{c_0} + 1)R_0 \right). \tag{146}
$$

Then,

$$
\text{Lip}_{\mathbb{R}^n}(P_{t_{k+1}, t_k}) \leq \varepsilon. \tag{147}
$$

We define the map

$$
\alpha(t) = \begin{cases} f_t|_K, & \text{for } 0 < t \leq 1, \\ Df|_0, & \text{for } t = 0, \end{cases}
$$

is a continuous map $[0, 1] \to C^1(\mathbb{R}^n; \mathbb{R}^n)$.

The space $W$ is a finite dimensional Euclidean space, and let $GL(W)$ be the set of linear diffeomorphisms of it, that is, invertible linear operators $A : W \to W$. We consider $GL(W)$ with the topology given by the operator norm of linear maps. Then $GL(W)$ is a topological space with two path connected components – those matrices which have a positive determinant and those having a negative determinant.

The above implies that $Df|_0$ can be connected in $GL(\mathbb{R}^n)$ either to a matrix $B$ where

$$
B \text{ is either the identity map, or the diagonal matrix diag}(-1, 1, 1, \ldots, 1), \tag{148}
$$

with a continuous path $\beta_s \in GL(W)$, $s \in [0, 1]$ with $Df|_0 = \beta_1$ and $B = \beta_0$.

We need to consider the path $\beta_s$ in $GL(W)$ from $\beta_1 = f_0 = Df|_0$ to matrix $\beta_0 = B \in O(n)$. There are some explicit formulas in literature with relatively complicated formulas, see [41]. However, let us next consider estimates using a non-optimal but relatively explicit path. To do this we start from the PU-decomposition of the matrix $\beta_1 = Df|_0$, that we denote $\beta_1 = PU$ where $P = \text{diag}(\sigma_1, \ldots, \sigma_n)$ is positive matrix (given in a suitable basis) and $U$ is an orthogonal matrix. Then, we consider the path $t \to P_t U$, where $t \in [0, 1]$ and

$$
P_t = P^{1-t} = \exp((1 - t) \log(P)).
$$

This is a path from the matrix $Df|_0 = PU$ to the matrix $U$. Observe that

$$
P_{t_2} U = P^{t_1 - t_2} P_{t_1} U,
$$

where

$$P^{t_1-t_2} = \text{diag}\,(\sigma_1^{t_1-t_2}, \ldots, \sigma_n^{t_1-t_2}),$$

and when $|t_1 - t_2| \leq 1$, by mean value theorem we have

$$
\begin{aligned}
\|P^{t_1-t_2} - I\| &\leq \max_{j=1,\ldots,n}(\sigma_j^{t_1-t_2} - 1) \leq \max_{j=1,\ldots,n}(\sigma_j, \sigma_j^{-1}) \cdot (\max_{j=1,\ldots,n}|\log \sigma_j|) \cdot |t_1 - t_2| \\
&\leq (1 + c_1 + c_0^{-1})^2 \cdot |t_1 - t_2|.
\end{aligned}
$$

That is, using the norm of $Df|_0$ and the norm of inverse matrix $(Df|_0)^{-1}$ we can bound the length of the path $t \to P_t U$, $t \in [0, 1]$ (in the operator norm). After this, we need to consider the path from $U$ to $B = Id_k \times (-Id_{n-k})$ in $O(W)$. For this, we could use the structure of the Lie group $O(n)$, $n = \dim(W)$. By [15, Thm. 11.3.3], any matrix $U$ in $O(n)$ can be written as a tensor product of elements in $O(2)$ and possibly an operator $\pm Id : \mathbb{R} \to R$, that is, in a suitable orthogonal basis a matrix $U \in O(n)$ is a block diagonal matrix of $2 \times 2$ matrixes in $O(2)$ and one or two $1 \times 1$ matrices $\pm Id$, that is,

$$U = \begin{bmatrix} R_1 & & & & & \\ & \ddots & & & & 0 \\ & & R_k & & & \\ & & & \pm 1 & & \\ & 0 & & & \ddots & \\ & & & & & \pm 1 \end{bmatrix},$$

where the matrices $R_1 = R_1(\vartheta_1)$, ..., $R_k = R_1(\vartheta_k)$ are 2-by-2 rotation matrices in $SO(2)$,

Using this, we find a path form $U(s)$, $s \in [0, 1]$ either to a matrix $Id_k \times (-Id_{n-k})$,

$$U(s) = \begin{bmatrix} R_1(s\vartheta_1) & & & & & \\ & \ddots & & & & 0 \\ & & R_k(s\vartheta_k) & & & \\ & & & \pm 1 & & \\ & 0 & & & \ddots & \\ & & & & & \pm 1 \end{bmatrix}.$$

Observe that due the block diagonal form of this matrix, the operator norm satisfies

$$\|U(s_2) - U(s_1)\|_{\mathbb{R}^n \to \mathbb{R}^n} \leq \max_{j=1,2\ldots,k} \|R_j(s_2\vartheta_j) - R_j(s_1\vartheta_j)\|_{R^2 \to R^2} \leq C_*|s_2 - s_1|,$$

where $C_*$ is an absolute constant (that does not depend on the dimension $n$). We obtain the path $s \to \beta_s$ by concatenating the paths from $PU$ to $U$ in $GL(n)$ and from $U$ to $B$ in $SO(n)$. The length of the obtained path $\beta$ in the operator norm metric can be estimated and it is bounded $C(1 + c_1 + c_0^{-1})^2$ plus a constant. Moreover, the path $s \to \beta_s$ can be decomposed to a product of $(C(1 + c_1 + c_0^{-1})^2 + C)\varepsilon^{-1}$ matrices of the form $Id + B_j$ where $\|B_j\| \leq \varepsilon$.

Summarising the above analysis, we see that $J$ can bounded by

$$J \leq \frac{1}{\varepsilon}\frac{2}{t_1}\left(\frac{c_1}{c_0}(10 + R_2) + \frac{6}{R_2}(\frac{c_1}{c_0} + 1)R_0\right) + C(1 + c_1 + c_0^{-1})^2\varepsilon^{-1} + C\varepsilon^{-1}. \tag{149}$$

We recall that here

$$t_1 < \frac{2c_0}{c_1 + \|f\|_{C^2(B_{\mathbb{R}^n}(0,R_1))}R_1}\varepsilon, \tag{150}$$

radii $R_0$ and $R_1$ are given in formulas (110) and (111) and $c_0$ and $c_1$ are the bi-Lipschitz constants of $f$, see (106).

Note that $c_0, c_1, C > 0$ are constants independent of $\epsilon$, while $R_0, R_1, R_2 > 0$ depends on $|f(0)|$. We see that by $f = P_W F|_W$.

$$|f(0)| \leq \|F(0)\|_X,$$

thus, the upper bounds of $R_0, R_1, R_2$ are independent of $\epsilon$. By choosing $t_1$ as $t_1 = \frac{c_0}{c_1 + \|f\|_{C^2(B_{\mathbb{R}^n}(0,R_1))}R_1}\varepsilon$, we furthermore estimate that

$$J \lesssim \|f\|_{C^2(B_{\mathbb{R}^n}(0,R_1);\mathbb{R}^n)}\varepsilon^{-2}. \tag{151}$$

Here, if we assume that $F \in C^2(X, X)$, we see that

$$\|f\|_{C^2(B_{\mathbb{R}^n}(0,R_1);\mathbb{R}^n)} \le \|F\|_{C^2(W;X)} \le \|F\|_{C^2(X;X)} < \infty,$$

Thus, we obtain

$$J = \mathcal{O}(\varepsilon^{-2}).$$

Now we can finish the proof of Theorem 4. Denote $\tilde{f}_t = f_t \oplus Id_{W^\perp}$, $\tilde{H}_{t_1,t_0} = H_{t_1,t_0} \oplus Id_{W^\perp}$ $\tilde{\beta}_s = \beta_s \oplus Id_{W^\perp}$ and $A_0 = B \oplus Id_{W^\perp}$. Using these notations, we can write $F(x)$ as

$$F(x) = (F \circ (F^W)^{-1}) \circ \tilde{H}_{t_{m+1},t_m} \circ \tilde{H}_{t_m,t_{m-1}} \circ \dots$$
$$\dots \circ \tilde{H}_{t_1,t_0} \circ (\tilde{\beta}_1 \circ \tilde{\beta}_{s_m}^{-1}) \circ (\tilde{\beta}_{s_m} \circ \tilde{\beta}_{s_{m-1}}^{-1}) \circ \dots \circ (\tilde{\beta}_{s_1} \circ \tilde{\beta}_0^{-1}) \circ A_0, \tag{152}$$

where $t_j$ and $s_j$ are chosen so that $0 = t_0 < t_1 < \dots t_m < t_{m+1} = 1$ and $0 = s_0 < s_1 < \dots s_m < s_{m+1} = 1$ and that the Lipschitz constant of the maps $\tilde{H}_{t_{j+1},t_j} - Id$ and $\tilde{\beta}_{s_j} \circ \tilde{\beta}_{j-1}^{-1} - Id$ are less that $\epsilon$.

Moreover, recall that

$$F^W = Id + P_W \circ T_2 \circ G \circ T_1 \circ P_W : X \to X.$$

Observe that by writing $x \in X$ in the form $x = x_0 + x_1$, where $x_0 = (I - P_W)x$ and $x_1 = P_W x$, we see that

$$
\begin{aligned}
F^W(x_0 + x_1) &= (I + P_W \circ T_2 \circ G \circ T_1 \circ P_W)(x_0 + x_1) \\
&= x_0 + (I + P_W \circ T_2 \circ G \circ T_1 \circ P_W)(x_1) \\
&= (I - P_W)x + P_W\Big((I + P_W \circ T_2 \circ G \circ T_1 \circ P_W))(P_W x)\Big) \\
&= (I - P_W)x + P_W(F^W(P_W x)). \tag{153}
\end{aligned}
$$

Hence, $(F^W)^{-1} : X \to X$ can be written as

$$
\begin{aligned}
(F^W)^{-1} &= I - P_W + P_W \circ (F^W)^{-1} \circ P_W \\
&= I + P_W \circ (-I + (F^W)^{-1}) \circ P_W, \tag{154}
\end{aligned}
$$

and thus $(F^W)^{-1}$ is a layer of a neural operator by definition. Similarly, as

$$\tilde{f} = F^W = Id_X + P_W \circ (F_W - Id_W) \circ P_W : X \to X.$$

and

$$
\begin{aligned}
f_t(x) &= \frac{1}{t}(f(tx) - f(0)) + tf(0) \\
&= Id - P_W + P_W(Id + \frac{1}{t}(f(tP_W x) - f(0)) + tf(0)), \tag{155}
\end{aligned}
$$

for $0 < t \le 1$, and we see as above that $f_t$ and $f_t^{-1}$ are neural operators. Similarly, we see that $\tilde{\beta}_s^{-1}$ and $\tilde{\beta}_s^{-1}$ are neural operators. Hence, all factors in the product (152) are (strictly monotone) neural operators. $\qquad\square$

### A.7.2 Proof of Theorem 5

**Theorem 6** (Theorem 5 in the main text). *Let $X$ be a separable Hilbert space, and let $\varphi = \{\varphi_n\}_{n \in \mathbb{N}}$ be an orthonormal basis in $X$. Let $\delta \in (0, 1)$, and let $R > 0$, and let $F \colon X \to X$ be a layer of a bilipschitz neural operator, as in Def. 3. Then, for any $\epsilon \in (0, 1)$, there are $T, N \in \mathbb{N}$ and $G \in \mathcal{R}_{T,N,\varphi,ReLU}(X)$ that has the form*

$$G = (Id_X + D_N \circ NN_T \circ E_N) \circ \cdots \circ (Id_X + D_N \circ NN_1 \circ E_N),$$

*such that*

$$\sup_{x \in \overline{B}_X(0,R)} \|F(x) - G \circ A(x)\|_X \le \epsilon,$$

*where $A \colon X \to X$ is a linear invertible map that is either the identity map or a reflection operator $x \to x - 2\langle x, e \rangle_X e$ with some unit vector $e \in X$. Moreover, $G \circ A \colon B_X(0, R) \to G \circ A(B_X(0, R))$ is invertible, and there exists $R' > 0$ such that*

$$A(B_X(0, R)) \subset B_X(0, R'),$$

*and denoting by*

$$\Gamma_0 := B_X(0, R'), \quad \Gamma_1 := B(0, R'+\delta), \quad \cdots \quad \Gamma_T := B(0, R'+T\delta), \quad \Gamma_{T+1} := B(0, R'+(T+1)\delta),$$

*and*

$$\tilde{K}_0 := A(B_X(0, R)), \quad \tilde{K}_1 := (Id_X + D_N \circ NN_1 \circ E_N)\tilde{K}_0, \quad \tilde{K}_2 := (Id_X + D_N \circ NN_2 \circ E_N)\tilde{K}_1,$$

$$\cdots \quad \tilde{K}_T := (Id_X + D_N \circ NN_T \circ E_N)\tilde{K}_{T-1} = G \circ A(B_X(0, R)),$$

*we have that for each $t = 0, 1, ..., T$*

$$\tilde{K}_t \subset \Gamma_t$$

*and the mapping $G \circ A|_{B_X(0,R)} \colon B_X(0, R) \to G \circ A(B_X(0, R))$ is bijective and its inverse is given by*

$$\left(G \circ A|_{B_X(0,R)}\right)^{-1} = A^{-1} \circ \Phi,$$

*where $\Phi \colon G \circ A(B_X(0, R)) \to A(B_X(0, R))$ is defined by*

$$\Phi := L_1|_{\tilde{K}_1} \circ \cdots \circ L_{T-1}|_{\tilde{K}_{T-1}} \circ L_T|_{\tilde{K}_T}, \tag{156}$$

*where $L_t \colon \Gamma_t \to \Gamma_{t+1}$*

*is defined by*

$$L_t(y) := \lim_{n \to \infty} \pi_1 \circ \left(\bigcirc_{h=1}^{n} \phi_t\right) \circ e_1(y),$$

*where $\phi_t \colon \Gamma_{t+1} \times \Gamma_t \to \Gamma_{t+1} \times \Gamma_t$ is defined by*

$$\phi_t(x, y) = (y + D_N \circ NN_t \circ E_N(x), y),$$

*where $\pi_1(x, y) = x$ and $e_1(y) = (0, y)$.*

**Remark 2.** *Note that, in the proof, we have used the approximation result [22, Theorem 4.3] of Sobolev functions by neural networks with ReLU function $x \mapsto \max\{0, x\}$, which is not differentiable at 0. Alternatively, we can use approximation result [1, Theorem 4.1] by neural networks with ReCU function $x \mapsto \max\{0, x\}^3$, which are continuously differentiable. Then, each block $Id_X + D_N \circ NN_t \circ E_N$ in a obtained approximator $G$ is $C^1$ and strongly monotone on ball $\Gamma_t = B(0, R' + t\delta)$, that is, it holds that there is an $\alpha > 0$ such that*

$$\langle (Id_X + D_N \circ NN_t \circ E_N)x, x \rangle_X \ge \alpha \|x - y\|_X^2, \; x, y \in \Gamma_t,$$

*which implies that, by the same argument as in Lemma 3,*

$$(Id_X + D_N \circ NN_t \circ E_N) \colon \Gamma_t \to (Id_X + D_N \circ NN_t \circ E_N)(\Gamma_t),$$

*is a $C^1$-diffeomorphism.*

**Remark 3.** *From the proof, we can show that, for each $t = 1, ..., T$*

$$L_t(\tilde{K}_t) \subset \tilde{K}_{t-1}.$$

*Then, the mapping $\Phi$ defined in (156) is well-defined.*

*Proof.* Let $\delta, \epsilon \in (0, 1)$, and let $F \colon X \to X$ be a layer of a neural operator defined in (4). Assume that $F$ is bilipschitz, that is, it satisfies (85) and

$$\|F(x) - F(y)\|_X \leq C_1 \|x - y\|_X, \quad \text{for all } x, y \in X. \tag{157}$$

By Theorem 4, there is $T \in \mathbb{N}$ such that $F$ can represented in the form

$$F(x) = (Id_X - H_T) \circ (Id_X - H_{T-1}) \circ \cdots \circ (Id_X - H_1) \circ A(x), \quad \text{for } x \in B_X(0, R),$$

where $A : X \to X$ is a linear invertible map that is either the identity map or a reflection operator $x \to x - 2\langle x, e \rangle_X e$ with some unit vector $e \in X$, and $H_t : X \to X$ are global Lipschitz maps satisfying

$$\mathrm{Lip}_{X \to X}(H_t) \leq \delta/2. \tag{158}$$

for all $t$. Moreover, the operator $H_t : X \to X$ is a compact mapping. We choose $R' > 0$ such that

$$A(B_X(0, R)) \subset B_X(0, R').$$

We denote by

$$K_0 := A(B_X(0, R)), \quad K_1 := (Id_X - H_1)(K_0), \quad \cdots \quad K_T := (Id_X - H_T)(K_{T-1}).$$

$$\Gamma_0 := B_X(0, R'), \quad \Gamma_1 := B(0, R' + \delta), \quad \cdots \quad \Gamma_{T+1} := B(0, R' + (T+1)\delta).$$

We choose $\tilde{R} > 0$ such that for all $t = 0, ..., T+1$

$$E_N(K_t), E_N(\Gamma_t) \subset B_{\mathbb{R}^N}(0, \tilde{R}).$$

Since $H_t : X \to X$ is a compact mapping, for large enough $N \in \mathbb{N}$, we have that for all $t = 1, ..., T$,

$$\sup_{x \in B_X(R)} \|H_t(x) - P_{V_N} \circ H_t \circ P_{V_N}(x)\|_X \leq \frac{\epsilon}{2T(1+\delta)^T}. \tag{159}$$

Note that

$$P_{V_N} \circ H_t \circ P_{V_N} = D_N \circ \tilde{H}_{N,t} \circ E_N,$$

where $\tilde{H}_{N,t} : \mathbb{R}^N \to \mathbb{R}^N$ by

$$\tilde{H}_{N,t} := E_N \circ H_t \circ D_N.$$

From (158), we have

$$\tilde{H}_{N,t} \in W^{1,\infty}(B_{\mathbb{R}^N}(0, \tilde{R}); \mathbb{R}^N).$$

By approximation by ReLU neural networks in Sobolev spaces (see e.g., [22, Theorem 4.3]) , there is a ReLU neural network $NN_t : \mathbb{R}^N \to \mathbb{R}^N$, $NN_t \in \mathcal{R}_{1,N,\varphi,ReLU}(X)$ such that

$$\|\tilde{H}_{N,t} - NN_t\|_{W^{1,\infty}(B_{\mathbb{R}^N}(0,\tilde{R});\mathbb{R}^N)} \leq \min\left\{ \frac{\epsilon}{2T(1+\delta)^T}, \frac{\delta}{2} \right\}. \tag{160}$$

Then, we estimate that by (159) and (160)

$$\sup_{x \in K_{t-1}} \|H_t(x) - D_N \circ NN_t \circ E_N(x)\|_X \leq \frac{\epsilon}{T(1+\delta)^T}. \tag{161}$$

Also, we estimate that by (158) and (160)

$$\begin{aligned}
\mathrm{Lip}_{B_{\mathbb{R}^N}(0,\tilde{R}) \to \mathbb{R}^N}(NN_t) \quad &\leq \mathrm{Lip}_{B_{\mathbb{R}^N}(0,\tilde{R}) \to \mathbb{R}^N}(\tilde{H}_{N,t}) + \mathrm{Lip}_{B_{\mathbb{R}^N}(0,\tilde{R}) \to \mathbb{R}^N}(\tilde{H}_{N,t} - NN_t) \\
&\leq \mathrm{Lip}_{X \to X}(H_t) + \|\tilde{H}_{N,t} - NN_t\|_{W^{1,\infty}(B_{\mathbb{R}^N}(0,\tilde{R});\mathbb{R}^N)} \leq \delta. \quad (162)
\end{aligned}$$

We denote $G : X \to X$ by

$$G := (I - D_N \circ NN_T \circ E_N) \circ (I - D_N \circ NN_{T-1} \circ E_N) \circ \cdots \circ (I - D_N \circ NN_1 \circ E_N),$$

which belong to $\mathcal{R}_{T,N,\varphi,ReLU}(X)$. Then, by (161) and (162), we estimate that for all $x \in B_X(0,R)$,

$\|F(x) - G \circ A(x)\|_X$

$$\leq \sum_{1 \leq t \leq T} \| \left( \bigcirc_{h=t+1}^{T}(I - D_N \circ NN_h \circ E_N) \right) \circ \left( \bigcirc_{h=1}^{t}(I - H_h) \right) \circ A(x)$$

$$- \left( \bigcirc_{h=t}^{T}(I - D_N \circ NN_h \circ E_N) \right) \circ \left( \bigcirc_{h=1}^{t-1}(I - H_h) \right) \circ A(x)\|_X$$

$$\leq \sum_{1 \leq t \leq T} \prod_{h=t+1}^{T} \left( 1 + \mathrm{Lip}_{B_{\mathbb{R}^N}(0,\tilde{R}) \to \mathbb{R}^N}(NN_h) \right) \sup_{y \in K_{t-1}} \|(I - H_t)(y) - (I - D_N \circ NN_t \circ E_N)(y)\|_X$$

$$\leq \sum_{1 \leq t \leq T} \prod_{h=t+1}^{T} (1 + \delta) \sup_{y \in K_{t-1}} \|H_t(y) - D_N \circ NN_t \circ E_N(y)\|_X$$

$$\leq \sum_{1 \leq i \leq T} (1 + \delta)^T \frac{\epsilon}{T(1+\delta)^T} \leq \epsilon.$$

Next, as (160), we see that

$$(I - D_N \circ NN_1 \circ E_N)|_{\Gamma_0} : \Gamma_0 \to \Gamma_1, \quad (I - D_N \circ NN_2 \circ E_N)|_{\Gamma_1} : \Gamma_1 \to \Gamma_2, \quad \cdots$$

$$(I - D_N \circ NN_T \circ E_N)|_{\Gamma_{T-1}} : \Gamma_{T-1} \to \Gamma_T.$$

and

$$G|_{\Gamma_0} = (I - D_N \circ NN_T \circ E_N)|_{\Gamma_{T-1}} \circ (I - D_N \circ NN_{T-1} \circ E_N)|_{\Gamma_{T-2}} \circ \cdots \circ (I - D_N \circ NN_1 \circ E_N)|_{\Gamma_0},$$

which means that $G|_{\Gamma_0}$ maps from $\Gamma_0$ to $\Gamma_T$. For $t = 1, ..., T$, we define $L_t : \Gamma_t \to \Gamma_{t+1}$ by

$$L_t(y) := x^*, \quad y \in \Gamma_t,$$

where $x^* \in \Gamma_{t+1}$ is a unique fixed point of $h_{t,y} : \Gamma_{t+1} \to \Gamma_{t+1}$, where

$$h_{t,y}(x) := y + D_N \circ NN_t \circ E_N(x),$$

that is, $x^* \in \Gamma_{t+1}$ is a unique solution of

$$h_{t,y}(x) = x \quad \Longleftrightarrow \quad y = (I - D_N \circ NN_t \circ E_N)(x), \quad x \in \Gamma_{t+1}.$$

Indeed, there is a unique solution because we have for $x_1, x_2 \in \Gamma_{t+1}$, from (162)

$$\|h_{t,y}(x_1) - h_{t,y}(x_2)\|_X = \|D_N \circ NN_t \circ E_N(x_1) - D_N \circ NN_t \circ E_N(x_2)\|_X \leq \delta\|x_1 - x_2\|_X,$$

which implies that $h_{t,y} : \Gamma_{t+1} \to \Gamma_{t+1}$ is a contraction map. Then, we have for $x \in \Gamma_{t-1}$, $t = 1, ..., T$,

$$L_t \circ (I - D_N \circ NN_t \circ E_N)(x) = x. \tag{163}$$

Here, the solution $x^* \in \Gamma_{t+1}$ is given by

$$x^* = \lim_{n \to \infty} x_n,$$

where

$$x_0 = 0, \quad x_{n+1} = y + D_N \circ NN_t \circ E_N(x_n).$$

We define $\varphi_t : \Gamma_{t+1} \times \Gamma_t \to \Gamma_{t+1} \times \Gamma_t$ by

$$\varphi_t(x, y) = (y + D_N \circ NN_t \circ E_N(x), y).$$

Let $\pi_1(x, y) = x$ and $e_1(y) = (0, y)$ where $\pi_1 : X \times X \to X$ and $e_1 : X \to X \times X$ are linear maps. Then,

$$L_t(y) = x^* = \lim_{n \to \infty} \pi_1 \circ \left( \bigcirc_{h=1}^{n} \varphi_t \right) \circ e_1(y).$$

We define $\Phi_P : \Gamma_T \to \Gamma_0$ by

$$\Phi_P := P_{\Gamma_0} \circ L_1 \circ P_{\Gamma_1} \circ L_2 \circ \cdots \circ P_{\Gamma_{T-1}} \circ L_T,$$

where $P_{\Gamma_t} : X \to X$ is the projection onto the convex set $\Gamma_t = B(0, R' + t\delta)$. Then, by (163), we have for $x \in B_X(0, R')$

$$\Phi_P \circ G(x) = x. \tag{164}$$

Therefore, $\Phi_P$ is the left-inverse of $G$, that is, $G|_{A(B_X(0,R))} : A(B_X(0,R)) \to X$ is injective, and $G|_{A(B_X(0,R))} : A(B_X(0,R)) \to G \circ A(B_X(0,R))$ is bijective, and the inverse $(G|_{G \circ A(B_X(0,R))})^{-1} :$ $G \circ A(B_X(0,R)) \to A(B_X(0,R))$ is given by

$$(G|_{G \circ A(B_X(0,R))})^{-1} = \Phi,$$

where $\Phi : G \circ A(B_X(0,R)) \to A(B_X(0,R))$ is defined by $\Phi := \Phi_P|_{G \circ A(B_X(0,R))}$ and has the form

$$\Phi = L_1 \circ L_2 \circ \cdots \circ L_T|_{G \circ A(B_X(0,R))}.$$

Therefore, $G \circ A : B_X(0,R) \to G \circ A(B_X(0,R))$ is also bijective, and the inverse is given by

$$(G \circ A|_{B_X(0,R)})^{-1} = A^{-1} \circ \Phi.$$

Note that $G \circ A(B_X(0,R)) \subset \Gamma_T$. $\qquad\square$

### A.8 Production of Quantitative Universal Approximation Estimates

Here, we show how our framework may be used 'out of the box' to produce quantitative approximation results for discretization of neural operators.

Let $X$ be a separable Hilbert space. We will consider a Hilbert space $X$, endowed with its norm topology. Recall that $S_0(X) \subset S(X)$ is a partially ordered lattice of finite dimensional subspaces of $X$. Next, we consider quantified discretization of a continuous, possibly non-linear function $F \colon X \to X$, that is, how the discretizations $F_V \colon V \to V$ can be chosen (using e.g. neural networks $\mathcal{F}_V$ having a given architecture for each subspace $V \subset X$) so that the obtained discretization operator $\mathcal{A}_{\mathcal{F}}$ has the explicitly given error bounds $\varepsilon_V$.

Using quantitative approximation results for neural networks in $\mathbb{R}^d$, see e.g. [57] or [23], one obtains quantitative results for neural operators. An example of such result is given below.

**Proposition 5.** *Let $r > 0$ and $F \colon \overline{B}_X(0,r) \to X$ be a non-linear function satisfying $F \in Lip(\overline{B}_X(0,r); X)$, in $n = 1$, or $F \in C^n(\overline{B}_X(0,r); X)$, if $n \geq 2$. Let $\varepsilon_V > 0$ be numbers indexed by the linear subspaces $V \subset X$ such that $\varepsilon_V \to 0$ as $V \to X$. When $d = dim(V)$, the space $V$ is identified with $\mathbb{R}^d$ using an isometric isomorphism $J_V \colon V \to \mathbb{R}^d$. Then there is a feed forward neural network $F_{V,\theta} \colon \mathbb{R}^d \to \mathbb{R}^d$ with ReLU-activation functions with at most $C(n,d) \log_2((1+r)^n/\varepsilon_V)$ layers and $C(n,d) \varepsilon_V^{-d/n} \log_2((1+r)^n/\varepsilon_V)$ non-zero elements in the weight matrices such that $\mathcal{A}_{NN} \colon F \to (F_V)_{V \in S_0(X)}$, where $F_V = J_V^{-1} \circ F_{V,\theta} \circ J_V \colon V \to V$, is an $\vec{\varepsilon}$-approximation operation in the ball $B_X(0,r)$.*

*Proof.* Let $\text{Lip}(F \colon \overline{B}_X(0,r) \to X) \leq M$, in $n = 1$, or $\|F\|_{C^n(\overline{B}_X(0,r); X)} \leq M$, if $n \geq 2$. Let $V \subset X$ be a linear subspace of dimension $d$ and $\varepsilon_V > 0$. Let us use some orthogonal basis of $V$ to identify $V$ and $\mathbb{R}^d$ and denote the identifying isomorphism by $J \colon \mathbb{R}^d \to V$. The function $\hat{F} \colon \mathbb{R}^d \to \mathbb{R}^d$, given by $\hat{F} = J^{-1} \circ P_V \circ F \circ J$ satisfies $\text{Lip}(\hat{F} \colon \overline{B}_{\mathbb{R}^d}(0,r)) \to X) \leq M$ if $n = 1$, or $\|\hat{F}\|_{C^n(\overline{B}_{\mathbb{R}^d}(0,r))} \leq M$ if $n \geq 2$. Then, by applying [57, Theorem 1] or [23], we see that there exists a feed forward neural network $F_{V,\theta} \colon \mathbb{R}^d \to \mathbb{R}^d$ with ReLU-activation functions with at most $C(n,d) \log_2((1+r)^n/\varepsilon_V)$ layers and $C(n,d) \varepsilon_V^{-d/n} \log_2((1+r)^n/\varepsilon_V)$ non-zero elements in the weight matrices such that

$$\|F_{V,\theta}(x) - \hat{F}(x)\|_{\mathbb{R}^d} \leq M\varepsilon_V. \tag{165}$$

Due to Def. 1, this yields the claim. $\qquad\square$

## B  Invertible residual network on separable Hilbert spaces

In this section, we consider the approximation of diffeomorphisms by globally invertible residual networks on Hilbert spaces. From the viewpoint of no-go theorem, as the class of diffeomorphisms is too large, we focus on the class of strongly monotone $C^1$-diffeomorphisms with compact support[5].

---

[5]We denote the support of $F : X \to X$ by $\text{supp}(F) := \overline{\{x \in X : F(x) \neq x\}}$.

We first obverse a following similar result with Theorem 3 that strongly monotone $C^1$-diffeomorphism $F : X \to X$ with compact support has the property that any linear discretizations $P_V F|_V$ are still strongly monotone $C^1$-diffeomorphism with compact support. The proof can be given by the same argument in Theorem 3 because $F$ has the form of $F = Id + B$ where $B := F - Id$ is a compact mapping.

**Proposition 6.** *Let $\mathcal{A}_{\text{lin}}$ be the discretization functor that maps $F$ to $P_V F|_V$ for each finite subspace $V \subset X$. Let $\mathcal{D}_{smc}$ and $\mathcal{B}_{smc}$ be categories where $F : X \to X$ and $F_V : V \to V$ are strongly monotone $C^1$-diffeomorphisms with compact support. Then, the functor $\mathcal{A}_{\text{lin}} : \mathcal{D}_{smc} \to \mathcal{B}_{smc}$ satisfies assumption (A), and it is continuous in the sense of Definition 8.*

Under the same setting and notations in Section 3.5, we define the class of invertible residual networks in the separable Hilbert space by, for $T, N \in \mathbb{N}$ and $\delta \in (0,1)$,

$$\mathcal{R}_{T,N,\varphi,\delta,\sigma}^{inv}(X) := \Big\{ G \in \mathcal{R}_{T,N,\varphi,\delta,\sigma}(X) : G = \bigcirc_{t=1}^{T}(Id_X + D_N \circ NN_t \circ E_N),$$

$$\text{Lip}_{X \to X}(D_N \circ NN_t \circ E_N) \leq \delta \ (t = 1, ..., T) \Big\}, \tag{166}$$

which is a subset of $\mathcal{R}_{T,N,\varphi,\delta,\sigma}^{inv}(X)$ defined in (10). The smallness of Lipschitz constants $\text{Lip}_{X \to X}(D_N \circ NN_t \circ E_N)$ implies that $\mathcal{R}_{T,N,\varphi,\delta,\sigma}^{inv}(X)$ is included in the class of homeomorphisms. The following lemma shows this fact.

**Lemma 9.** *Let $\delta \in (0,1)$, and let $F \in \mathcal{R}_{T,N,\varphi,\delta,\sigma}^{inv}(X)$. If $\sigma : \mathbb{R} \to \mathbb{R}$ is Lipschitz continuous, then $F : X \to X$ is homeomorphism. Moreover, if $\sigma : \mathbb{R} \to \mathbb{R}$ is $C^1$, then, $F : X \to X$ is $C^1$-diffeomorphism.*

*Proof.* Assume that $\sigma$ is Lipschitz continuous. Let $G \in \mathcal{R}_{T,N,\varphi,\delta,\sigma}^{inv}(X)$, that is,

$$G = (Id_X + D_N \circ NN_T \circ E_N) \circ \cdots \circ (Id_X + D_N \circ NN_1 \circ E_N),$$

where $\text{Lip}_{X \to X}(D_N \circ NN_t \circ E_N) \leq \delta$. It is enough to show that for all $t = 1, ..., L$

$$Id_X + D_N \circ NN_t \circ E_N : X \to X,$$

is homeomorphism. Indeed, we have for $u, v \in X$

$$\langle (Id_X + D_N \circ NN_t \circ E_N)(u) - (Id_X + D_N \circ NN_t \circ E_N)(v), u - v \rangle_X$$
$$= \|u - v\|_X^2 + \langle D_N \circ NN_t \circ E_N(u) - D_N \circ NN_t \circ E_N(v), u - v \rangle_X \geq (1 - \delta)\|u - v\|_X^2, \tag{167}$$

that is, $Id_X + D_N \circ NN_t \circ E_N : X \to X$ is strongly monotone. By the same argument in Lemma 1, we can show that $(Id_X + D_N \circ NN_t \circ E_N) : X \to X$ is coercive. By the Minty-Browder theorem [10, Theorem 9.14-1], $(Id_X + D_N \circ NN_t \circ E_N) : X \to X$ is bijective, and then its inverse exists. Denoting by $H_t := Id_X + D_N \circ NN_t \circ E_N$, we see that by substituting $u = H^{-1}(u)$ and $v = H^{-1}(v)$ for (167)

$$\|H_t^{-1}(u) - H_t^{-1}(v)\|_X^2 \leq \frac{1}{1 - \delta}\langle H_t \circ H_t^{-1}(u) - H_t \circ H_t^{-1}(v), H_t^{-1}(u) - H_t^{-1}(v) \rangle_X$$

$$\leq \frac{1}{1 - \delta}\|u - v\|_X \|H_t^{-1}(u) - H_t^{-1}(v)\|_X,$$

which means that its inverse is continuous. Therefore, $(Id_X + D_N \circ NN_t \circ E_N) : X \to X$ is homeomorphism.

For the second statement, we assume that $\sigma$ is $C^1$. By the same argument in Lemma 1, we can show that the derivative $DH_t|_u : X \to X$ at $u \in X$ is given by

$$DH_t|_u = I_X + D_N \circ D(NN_t)|_{E_N(u)} \circ E_N,$$

and it is injective. Since $D_N \circ D(NN_t)|_{E_N(u)} \circ E_N : X \to X$ is a finite dimensional linear operator, it is compact operator. Then by the Fredholm theorem, $DH_t|_u : X \to X$ is bijective. By the inverse function theorem, the inverse $H_t^{-1} : X \to X$ is $C^1$, which implies that $H_t : X \to X$ is $C^1$-diffeomorphism. $\qquad \square$

In what follow, we employ $\sigma$ as the GroupSort activation having a group size of 2 (see [3, Section 4]). As sort activation is 1-Lipschitz, from Lemma 9, $\mathcal{R}^{inv}_{L,N,\varphi,\delta,\sigma}(X)$ is a subset of the class of homeomorphisms. We finally show that, in this case, $\mathcal{R}^{inv}_{L,N,\varphi,\delta,\sigma}(X)$ is an universal approximator for the class of the strongly monotone diffeomorphisms with compact support.

**Theorem 7.** *Let $R > 0$, and let $F : X \to X$ be strongly monotone $C^1$-diffeomorphism with compact support, and let $\sigma$ be the GroupSort activation having a group size of 2. Then, for any orthonormal basis $\{\varphi_n\}_{n \in \mathbb{N}} \subset X$, $\delta \in (0, 1)$, and $\epsilon \in (0, 1)$, there are $T, N \in \mathbb{N}$, and $G \in \mathcal{R}^{inv}_{T,N,\varphi,\delta,\sigma}(X)$ such that*

$$\sup_{u \in \overline{B}_X(0,R)} \|F(u) - G(u)\|_X \le \epsilon.$$

*Moreover, the inverse $G^{-1} : X \to X$ of $G \in \mathcal{R}^{inv}_{T,N,\varphi,\delta,\sigma}(X)$ is given by*

$$G^{-1} = \tilde{L}_1 \circ \cdots \circ \tilde{L}_{T-1} \circ \tilde{L}_T, \tag{168}$$

*where $\tilde{L}_t : X \to X$ is defined by*

$$L_t(y) := \lim_{n \to \infty} \pi_1 \circ \left( \bigcirc_{h=1}^n \tilde{\phi}_t \right) \circ e_1(y),$$

*where $\tilde{\phi}_t : X \times X \to X \times X$ is defined by*

$$\tilde{\phi}_t(x, y) = (y + D_N \circ NN_t \circ E_N(x), y),$$

*where $\pi_1(x, y) = x$ and $e_1(y) = (0, y)$.*

*Proof.* We denote by $\mathrm{Diff}^1(X)$ the class of $C^1$-diffeomorphisms between $X$, and $\mathrm{Diff}^1_{sm}(X)$ the class of strongly monotone $C^1$-diffeomorphisms between $X$. We also denote the support of $F \in \mathrm{Diff}^1(X)$ by $\mathrm{supp}(F) := \overline{\{x \in X : F(x) \ne x\}}$. We say that $F \in \mathrm{Diff}^1_{sm,c}(X)$ if $F \in \mathrm{Diff}^1_{sm}(X)$ has a compact support.

Let $F \in \mathrm{Diff}^1_{c,sm}(X)$. We define $F_1 : X \to X$ by $F_1 := Id_X + P_{V_N}(F - Id_X)P_{V_N}$, and we see that

$$F_1 = Id_X + P_{V_N}(F - Id_X)P_{V_N} = P_{V_N^\perp} + P_{V_N}FP_{V_N} = P_{V_N^\perp} + D_N E_N F D_N E_N.$$

We can show that for large $N \in \mathbb{N}$

$$\sup_{u \in \overline{B}_X(0,R)} \|F(u) - F_1(u)\|_X \le \sup_{u \in \overline{B}_X(0,R)} \|P_{V_N}(Id_X - F)P_{V_N}(u)\|_X \le \epsilon, \tag{169}$$

as $Id_X - F : X \to X$ is a compact mapping. By the same argument in Lemma 1, we can show that $F_N := E_N F D_N \in \mathrm{Diff}^1_{sm,c}(\mathbb{R}^N)$, and $DF_N|_0$ is a positive definite matrix, which is connected to $Id_{\mathbb{R}^N}$. By the similar argument in the proof of Theorem 4, see (105)–(152), we can construct continuous paths $f : [0,1] \to \mathrm{Diff}^1_{sm,c}(\mathbb{R}^N)$ and $\beta : [0,1] \to GL(\mathbb{R}^N)$ with $f_0 = DF_N|_0$, $f_1 = F_N$, $\beta_0 = Id_{\mathbb{R}^N}$, and $\beta_1 = DF_N|_0$ such that

$$F_N = (f_1 \circ f_{t_m}^{-1}) \circ (f_{t_m} \circ f_{t_{m-1}}^{-1}) \circ \ldots$$
$$\cdots \circ (f_{t_1} \circ f_0^{-1}) \circ (\beta_1 \circ \beta_{s_m}^{-1}) \circ (\beta_{s_m} \circ \beta_{s_{m-1}}^{-1}) \circ \cdots \circ (\beta_{s_1} \circ \beta_0^{-1}), \tag{170}$$

where $t_j$ and $s_j$ are chosen so that $0 = t_0 < t_1 < \ldots t_m < t_{m+1} = 1$ and $0 = s_0 < s_1 < \ldots s_m < s_{m+1} = 1$ and that the Lipschitz constant of the maps $f_{t_j} \circ f_{t_{j-1}}^{-1} - Id_{\mathbb{R}^N}$ and $\beta_{s_j} \circ \beta_{j-1}^{-1} - Id_{\mathbb{R}^N}$ are less that $\delta$.

Then, there is $T \in \mathbb{N}$ such that $F_N$ has the form

$$F_N = (Id_{\mathbb{R}^N} - H_T) \circ (Id_{\mathbb{R}^N} - H_{T-1}) \circ \cdots \circ (Id_{\mathbb{R}^N} - H_1),$$

where for each $t = 1, \ldots, T$,

$$\mathrm{Lip}_{\mathbb{R}^N \to \mathbb{R}^N}(H_t) \le \delta.$$

Then, remarking that $E_N D_N = Id_{\mathbb{R}^N}$ and $D_N E_N = P_{V_N}$, we see that

$$F_1 = P_{V_N^\perp} + D_N(Id_{\mathbb{R}^N} - H_L) \circ \cdots \circ (Id_{\mathbb{R}^N} - H_1)E_N$$
$$= (Id_X - D_N \circ H_T \circ E_N) \circ \cdots \circ (Id_X - D_N \circ H_1 \circ E_N).$$

For each $t = 1, ..., T$, by using [3, Theorem 3 and Observation], $\delta$-Lipschitz function $H_t : \mathbb{R}^N \to \mathbb{R}^N$ can be approximated by a neural network $NN_t : \mathbb{R}^N \to \mathbb{R}^N$ with GroupSort activation $\sigma$ having a group size of 2 in $L^\infty$-norm on any compact set, and $NN_t : \mathbb{R}^N \to \mathbb{R}^N$ is $\delta$-Lipschitz continuous.

We denoting by

$$G := (Id_X - D_N \circ NN_T \circ E_N) \circ \cdots \circ (Id_X - D_N \circ NN_1 \circ E_N) \in \mathcal{R}^{inv}_{T,N,\varphi,\delta,\sigma}(X).$$

Note that $\mathrm{Lip}_{X \to X}(D_N \circ NN_\ell \circ E_N) \le \delta$. Then, we can show that, by similar way in the first half of proof of Theorem 5,

$$\sup_{u \in \overline{B}_X(0,R)} \|F_1(u) - G(u)\|_X \le \epsilon. \tag{171}$$

With (169) and (171), we obtain that

$$\sup_{u \in \overline{B}_X(0,R)} \|F(u) - G(u)\|_X \le \sup_{u \in \overline{B}_X(0,R)} \|F(u) - F_1(u)\|_X + \sup_{u \in \overline{B}_X(0,R)} \|F_1(u) - G(u)\|_X \le 2\epsilon.$$

The representation of inverse $G^{-1}$ can be given by the same argument in the proof of Theorem 5. $\quad\square$

Similarly in Corollary 1, Theorem 7 and Lemmas 10 and 11 have the following corollary.

**Corollary 2.** *Under the same setting and assumptions with Corollary 1, let $\mathcal{RNO}^{inv}_{T,N,\varphi,\delta,\sigma}(L^2(D;\mathbb{R}))$ be the class of invertible residual neural operators defined in (180). Then, the statement replacing $X$ with $L^2(D;\mathbb{R})$ and $G \in \mathcal{R}^{inv}_{T,N,\varphi,\delta,\sigma}(X)$ with $G \in \mathcal{RNO}^{inv}_{T,N,\varphi,\delta,\sigma}(L^2(D;\mathbb{R}))$ in Theorem 7 holds.*

## C   Neural operators

### C.1   Examples of generalized neural operators

In the main text, we defined generalized neural operators on Hilbert spaces. Here, we give several examples to show that they are extensions of classical neural operators [33, 30].

**Example 1.** *Let $X$ be a Hilbert space. Let $G_\ell : X \to X$ be a classical neural operator having the form*

$$G_\ell := (W_{T_\ell} + K_{T_\ell}) \circ \sigma(W_{T_\ell-1} + K_{T_\ell-1}) \circ \cdots \sigma(W_1 + K_1),$$

*where $W_\ell : X \to X$ are linear bounded operators corresponding to the local term and $K_\ell : X \to X$ are compact operators corresponding to the non-local term (e.g., integral operators having a smooth kernel or smooth basis). We assume that activation function $\sigma$ is $C^1$. Then, we have $G_\ell \in C^1(X;X)$. Let $T_1 = K_{lift} : X \to X$ and $T_2 = K_{proj} : X \to X$ be compact linear operators, which corresponds to lifting and projection, respectively. We denoting by*

$$F_\ell := I + T_1 \circ G_\ell \circ T_2,$$

*which is one block of classical neural operators with skip-connection. We also denote by $A_\ell = Id$ and $\sigma = Id$. Then, the generalized neural operator $H = F_L \circ \cdots \circ F_1$ corresponds to classical neural operators with skip-connections. In this paper, the skip-connection (ie., the structure of the identity plus some compact mapping) is so important to preserve bijectivity, discussed in Section 3.*

**Example 2.** *We show that classical neural operators , for example*

$$F_{clas} : u \mapsto (W_2 + K_2) \circ (\sigma(W_1 u + K_1(u)),$$

*see [30, 33], can be written in the form of the generalized neural operators that we consider of the form*

$$H(x) = A_k \circ \sigma \circ F_k \circ A_{k-1} \circ \sigma \circ F_{k-1} \circ \cdots \circ A_1 \circ \sigma \circ F_1 \circ A_0,$$

*where $A_j : X \to X$ are linear operators which may not be bijective and $F_j = Id + T_{j,1} \circ G \circ T_{j,2} : X \to X$.*

*We could start with the observations that for an infinite dimensional Hilbert space $X$ there is a linear isomorphism $J_{m,n} : X^n \to X^m$, where $X^m = X \times \cdots \times X$. The reason for this is that the cardinality of Hilbert basis of the space $X$ is the same as the cardinality of the Hilbert basis of $X^n$, see [26, Theorem 3.5].*

*First we observe that we can write an operator*

$$H(x) = \tilde{A}_k \circ \tilde{\sigma}_k \circ \tilde{F}_k \circ \tilde{A}_{k-1} \circ \tilde{\sigma}_{k-1} \circ \tilde{F}_{k-1} \circ \cdots \circ \tilde{A}_1 \circ \tilde{\sigma} \circ \tilde{F}_1 \circ \tilde{A}_0,$$

*where $\tilde{A}_j : X^{n_j} \to X^{n_{j+1}}$ are linear operators which may not be bijective and $\tilde{F}_j = Id + \tilde{T}_{j,1} \circ \tilde{G} \circ \tilde{T}_{j,2} : X^{n_j} \to X^{n_j}$ and $\tilde{\sigma}_k : X^{n_k} \to X^{n_k}$ is $\tilde{\sigma}(x_1, x_2, \ldots, x_{i_k}, x_{i_k+1}, \ldots, x_{n_k}) = (x_1, x_2, \ldots, x_{i_k}, \sigma(x_{i_k+1}), \ldots, \sigma(x_{n_k}))$ in the form*

$$H(x) = (J_{1,n_{k+1}} \circ \tilde{A}_k \circ J_{n_k,1}) \circ (J_{1,n_k} \circ \tilde{\sigma} \circ \tilde{F}_k \circ J_{n_k,1}) \circ (J_{1,n_k} \circ \tilde{A}_{k-1} \circ J_{n_{k-1},1}) \circ \cdots \circ (J_{1,n_1} \circ \tilde{A}_0 \circ J_{n_0,1}),$$

*where e.g. $(J_{1,n_{k+1}} \circ A_k \circ J_{n_k,1}) : X \to X$ and $(J_{1,n_k} \circ \tilde{\sigma}_k \circ \tilde{F}_k \circ J_{n_k,1}) : X \to X$. This means that in our formalism we can replace e.g. operators $A_k$ by matrices of operators $A_k$.*

*Next we go to the second step of the construction:*

*As an example, let us consider the classical neural operator*

$$F_{clas} : u \to (W_2 + K_2) \circ \sigma \circ (W_1 u + K_1(u)),$$

*where $W_1$ and $W_2$ are invertible matrices and $K_1$ and $K_2$ are compact operators, can be written as an generalized neural operator*

$$u \to L_4 \circ F_3 \circ L_2 \circ \tilde{\sigma} \circ L_1 \circ F_0(u),$$

*where $F_0 : X \to X$ is a layer of neural operator*

$$F_0(v) = v + W_1^{-1} K_1(v),$$

*and $L_1 : X \to X$ is the invertible linear operator*

$$L_1(u) = W_1 u,$$

*and $\tilde{\sigma}(u) = \sigma(u)$ and $L_2 : X \to X \times X$ is a non-invertible linear operator*

$$L_2(w) = (w, w),$$

*and and $F_3 : X \times X \to X \times X$ is a layer neural operator*

$$F_3(w_1, w_2) = (w_1, w_2 + W_2^{-1} K_2(w_2)),$$

*and $L_4 : X \times X \to X$ is the non-invertible linear operator*

$$L_4(u_1, u_2) = W_2(u_2 - u_1).$$

*Finally, we point out that if a non-invertible activation function $\sigma : X \to X$ satisfies $Lip(\sigma) \leq \lambda$, it can be written as*

$$u \to L_2 \circ \hat{\sigma} \circ L_1(u),$$

*where*

$$L_1 : u \to (u, u),$$

*and $\hat{\sigma}$ is an invertible function*

$$\hat{\sigma} : (u_1, u_2) \to (u_1, 2\lambda u_2 + \sigma(u_2)),$$

*and*

$$L_1 : (w_1, w_2) \to w_2 - 2\lambda w_1,$$

*Note that equation $2\lambda u + \sigma(u) = w$ can be written as a fixed point equation*

$$u = g_w(u) := (2\lambda)^{-1} w - (2\lambda)^{-1} \sigma(u),$$

*where $Lip(g_w) \leq 1/2$. Hence, $g_w : X \to X$ is a contraction and the equation $u = g_w(u)$ has a unique solution for all $w$ by Banach fixed point theorem. Hence, $u \to 2\lambda u + \sigma(u)$ is invertible.*

**Example 3.** *Let $D \subset \mathbb{R}^d$ be a domain, and let $L^2(D; \mathbb{R})$ be the real-valued $L^2$-function space on $D$, and let $\varphi = \{\varphi_n\}_{n \in \mathbb{N}} \subset L^2(D; \mathbb{R})$ be an orthonormal basis in $L^2(D; \mathbb{R})$. We consider the case when $X = L^2(D; \mathbb{R}^h) = L^2(D; \mathbb{R})^h$, and $A_j = W^{(j)}$ where $W^{(j)} \in \mathbb{R}^{h \times h}$ are invertible matrices, and $F_j = Id + P_{V_N^h} \circ K_N^{(j)} \circ P_{V_N^h} W^{(j)-1}$ ) (corresponding to $T_1 = P_{V_N^h}$, $T_2 = P_{V_N^h} W^{(j)-1}$, $G = K_N^{(j)}$)*

*where* $V_N := \text{span}\{\varphi_n\}_{n \leq N}$, *and* $K_N^{(j)} : L^2(D;\mathbb{R})^h \to L^2(D;\mathbb{R})^h$ *is a finite rank operator defined by*

$$K_j(u)(x) := \sum_{p,q \leq N} K_{p,q}^{(j)} \langle u, \varphi_p \rangle_{L^2(D;\mathbb{R})} \varphi_q(x), \quad x \in D,$$

*Then,* $H : L^2(D;\mathbb{R})^h \to L^2(D;\mathbb{R})^h$ *can be written by*

$$H = (W^{(k)} + K^{(k)}) \circ \sigma \circ (W^{(k-1)} + K^{(k-1)}) \circ \sigma \circ \cdots \circ (W^{(1)} + K^{(1)}) \circ \sigma \circ (W^{(0)} + K^{(0)}),$$

*which coincides with classical neural operators (assuming that all local weight matrices are invertible) used in e.g., [33]. See Definition 9 as well.*

**Example 4.** *Let us consider an example of an example of an generalized neural operator which is obtained by composition of non-linear integral operators and smooth activation functions. Let* $D \subset \mathbb{R}^d$ *be be a bounded domain with a smooth boundary. We consider the case when* $X = H^1(D;\mathbb{R}^h) = H^1(D;\mathbb{R})^h$ *are Sobolev spaces, and* $A_j = W^{(j)}$ *where* $W^{(j)} \in \mathbb{R}^{h \times h}$ *are invertible matrices, and* $F_j = Id_{H^1} + i_{H^2 \to H^1} \circ \tilde{K}^{(j)} \circ i_{H^1 \to L^2} W^{(j)-1}$ *(corresponding to* $T_1 = i_{H^1 \to L^2}$, $T_2 = i_{H^2 \to H^1}(W^{(j)})^{-1}$, $G = \tilde{K}^{(j)}$*) where* $\tilde{K}^{(j)} : L^2(D;\mathbb{R})^h \to H^1(D;\mathbb{R})^h$ *is non-linear integral operator*

$$\tilde{K}^{(j)}(u)(x) := \int_D k^{(j)}(x, y, u(y))u(y)dy, \quad x \in D,$$

*where kernel satisfies* $k^{(j)} \in C^3(\overline{D} \times \overline{D} \times \mathbb{R}^h; \mathbb{R}^{h \times h})$ *has uniformly bounded three derivatives. Also, let* $\sigma \in C^1(\mathbb{R})$ *be an activation function which derivative is uniformly bounded and we denote* $\sigma_* f = \sigma \circ f$. *Then,* $H : H^1(D;\mathbb{R})^h \to H^1(D;\mathbb{R})^h$, *defined by*

$$\begin{aligned} H \quad &= \quad (W^{(k)} + i_{H^2 \to H^1} \circ \tilde{K}^{(j)} \circ i_{H^1 \to L^2}) \circ \sigma_* \circ (W^{(k-1)} + i_{H^2 \to H^1} \circ \tilde{K}^{(j)} \circ i_{H^1 \to L^2}) \circ \sigma_* \\ &\quad \circ \cdots \circ (W^{(1)} + K^{(1)}) \circ \sigma_* \circ (W^{(0)} + i_{H^2 \to H^1} \circ \tilde{K}^{(j)} \circ i_{H^1 \to L^2}), \end{aligned}$$

*is an generalized neural operator. Here,* $i_{H^1 \to L^2}$ *and* $i_{H^2 \to H^1}$ *are compact emending from* $H^1(D)$ *to* $L^2(D)$ *and* $H^2(D)$ *to* $H^1(D)$, *respectively. Non-linear integral operator in neural operator has been used in [30].*

**Example 5.** *Let us consider an example of a typical neural operator which is a finite composition of layers of neural operators similar to those introduced in [30, 33],* $F : X \to X$ *of the form*

$$F : u \to \sigma \circ (Wu + T_2(G(T_1u))), \tag{172}$$

*where* $X = H^m(\Omega)$, $\Omega \subset \mathbb{R}^d$ *is a bounded set with a smooth boundary and* $W : X \to X$ *is a linear operator. In the above,* $Y = C(\overline{\Omega})$ *and* $Z = C^{m+1}(\overline{\Omega})$, *where* $m > d/2$. *Moreover,* $G : Y \to Z$ *is a nonlinear integral operator*

$$G(u)(x) = \int_\Omega k_\theta(x, y, u(y))u(y)dy,$$

*where* $k_\theta, \partial_u k_\theta \in C^{m+1}(\overline{\Omega} \times \overline{\Omega} \times \mathbb{R})$ *is a kernel given by a feed-forward neural network with sufficiently smooth activation functions. Here, we assume that* $\sigma_j \in C^\ell(\mathbb{R})$, $\ell \geq m + 2$ *and that the kernel* $k_\theta$ *is of the form*

$$k_\theta(x, y, t) = \sum_{j=1}^J c_j(x, y, \theta)\sigma_j(a_j(x, y, \theta)t + b_j(x, y, \theta)).$$

*Moreover,* $T_1 : X \to Y$ *and* $T_2 : Z \to X$ *are the identical embedding operators*

$$T_j(u) = u.$$

*Due to the choice of function spaces* $X, Y$ *and* $Z$, *the maps* $T_1$ *and* $T_2$ *are compact operators. Summarizing,* $F = F_{\sigma,W,k}$, *where*

$$F_{\sigma,W,k}(u)(x) = \sigma((Wu)(x) + \int_\Omega k_\theta(x, y, u(y))u(y)dy). \tag{173}$$

*Furthermore, by choosing $k_\theta(x, y, u(y)) = k_\theta(x - y)$ and $\Omega = \mathbb{T}^d$ as the convolutional kernel and the torus, the map $F$ takes the form of an FNO [39].*

*Here the activation functions $\sigma$ and $\sigma_j$ are in different roles as the functions $\sigma_j$ appear inside the integral operator (or between the compact operators $T_1$ and $T_2$). The function $\sigma$ is useful in obtaining universal approximation results for neural operators, but as this question is somewhat technical, we postpone this discussion to the end.*

*The appearance of the compact operators $T_1$ and $T_2$ makes the discretization of activation function $\sigma$ and the activation functions inside $G$ in Definition 4 different, and this is the reason why we have introduced both $\sigma$ and $G$. To consider invertible neural operators, we will below assume that $\sigma$ is an invertible function, for example, the leaky Relu function. In the above operator, the nonlinear function $G$ inside compact operators in the operation*

$$N : u \to u + T_2(G(T_1 u)),$$

*and the compact operators map weakly converging sequences to norm converging sequences. This is essential in the proofs of the positive results for approximation functors as discussed in the paper. However, we do not have general results on how the operation*

$$G_\sigma : u \to \sigma \circ u,$$

*can be approximated by finite dimensional operators in the norm topology, but only in the weak topology in the sense of Definition 11 of the Weak Approximation Functor. However, one can overcome this difficulty in two ways. The first way is to use in the discretization a suitable finite dimensional space $V$ that satisfies $G_\sigma(V) \subset V$. For example, when $X = L^2(\Omega)$ and $\sigma$ is a leaky relu function, one can choose $V$ to be a space that consists of piecewise constant functions. The second way is choosing a composition of layers of the form*

$$N_j : u \to W_j u + T_{1,j}(G_j(T_{2,j} u)),$$

*and*

$$G_\sigma : u \to \sigma \circ u,$$

*in different finite dimensional spaces $V_j$. For example, we can consider these operations as maps*

$$N_1 : V_1 \to V_1, \tag{174}$$
$$G_\sigma : V_1 \to V_2 := G_\sigma(V_1), \tag{175}$$
$$N_2 : V_2 \to V_2, \tag{176}$$
$$G_\sigma : V_2 \to V_3 := G_\sigma(V_2). \tag{177}$$

*This makes the composition*

$$G_\sigma \circ N_2 \circ G_\sigma \circ N_1 : V_1 \to V_3,$$

*well defined. The maps, $G_\sigma$, are clearly invertible and one can use Theorems 4 and 5 to analyze when $N_j : V_j \to V_j$ are invertible functions.*

*Finally, we return to the question whether the activation function $\sigma$ is useful in universal approximation results. If the activation function $\sigma$ is removed (that is, it is the identical map $\sigma_{id} : s \to s$), the operator $F$ is a sum of a linear operator and a compact nonlinear integral operator,*

$$F_{W,k}(u)(x) = (Wu)(x) + \int_\Omega k_\theta(x, y, u(y))u(y)dy. \tag{178}$$

*Moreover, if we compose above operators $F_j$ of the above form, the obtained operator, $G : X \to X$ is also a sum of a linear operator and a compact operator,*

$$G(u) = \tilde{W}u + \tilde{K}(u).$$

*Moreover, the Frechet derivative of $G$ at $u_0$, denoted $DG|_{u_0}$ is the map*

$$DG|_{u_0} : v \to \tilde{W}v + D\tilde{K}|_{u_0}v,$$

*and, due to the above assumptions on kernel $k_\theta(x, y, u)$, the derivative is a linear operator*

$$D\tilde{K}|_{u_0} : H^m(\Omega) \to H^{m+1}(\Omega).$$

By the Sobolev embedding theorem it is a compact operator $D\tilde{K}|_{u_0} : X \to X$. This means that the Fredholm index of the derivative of $DG|_{u_0}$ is constant

$$Ind(DG|_{u_0}) = Ind(W),$$

that is, independent of the point $u_0$ where the derivative is computed. In particular, this means that one cannot approximate an arbitrary $C^1$-function $G : X \to X$ in compact subsets of $X$ by neural operators which are compositions of layers (178). Indeed, for a general $C^1$-function $G : X \to X$ the Fredholm index may be a varying function of $u_0$. Thus, $\sigma$ appears to be relevant for obtaining universal approximation theorems for neural operators.

## C.2 Residual neural operators

Let $D \subset \mathbb{R}^d$ be a domain. In what follows, we consider the real-valued $L^2$-function space $L^2(D;\mathbb{R})$ [6]. Let $\varphi = \{\varphi_n\}_{n \in \mathbb{N}} \subset L^2(D;\mathbb{R})$ be an orthonormal basis in $L^2(D;\mathbb{R})$.

**Definition 9** (Neural operators [33]). *We define a neural operator $G : L^2(D;\mathbb{R}) \to L^2(D;\mathbb{R})$ by*

$$G : u_0 \mapsto u_{L+1},$$

*where $u_{L+1}$ is give by the following steps:*

$$u_{\ell+1}(x) = \sigma\left(W^{(\ell)}u_\ell(x) + (K_N^{(\ell)}u_\ell)(x) + b^{(\ell)}\right), \quad x \in D, \quad 0 \le \ell \le L-1,$$

$$u_{L+1}(x) = W^{(L)}u_L(x) + (K_N^{(L)}u_L)(x) + b^{(L)}, \quad x \in D,$$

*where $\sigma : \mathbb{R} \to \mathbb{R}$ is a non-linear activation operating element-wise, and $W^{(\ell)} \in \mathbb{R}^{d_{\ell+1} \times d_\ell}$ and $b^{(\ell)} \in \mathbb{R}^{d_{\ell+1}}$ and*

$$K_N^{(\ell)}(v)(x) = \sum_{p,q \le N} K_{p,q}^{(\ell)} \langle v, \varphi_p \rangle_{L^2(D;\mathbb{R})} \varphi_q(x), \quad x \in D,$$

*where $K_{p,q}^{(\ell)} \in \mathbb{R}^{d_{\ell+1} \times d_\ell}$ ($\ell = 0, ..., L$, $p, q = 1, ..., N$, $d_0 = d_{L+2} = 1$). Here, we use the notation for $v = (v_1, ..., v_{d_\ell}) \in L^2(D;\mathbb{R})^{d_\ell}$*

$$\langle v, \varphi_p \rangle_{L^2(D;\mathbb{R})} = \left(\langle v_1, \varphi_p \rangle_{L^2(D;\mathbb{R})}, ..., \langle v_{d_\ell}, \varphi_p \rangle_{L^2(D;\mathbb{R})}\right) \in \mathbb{R}^{d_\ell}.$$

*We denote by $\mathcal{NO}_{L,N,\varphi,\sigma}(L^2(D;\mathbb{R}))$ the class of neural operators $G : L^2(D;\mathbb{R}) \to L^2(D;\mathbb{R})$ defined above, with depths $L$, rank $N$, orthonormal basis $\varphi$, and activation function $\sigma$.*

**Definition 10** (Residual Neural Operator). *Let $T, N \in \mathbb{N}$ and let $\delta \in (0,1)$. We define by*

$$\mathcal{RNO}_{T,N,\varphi,\sigma}(L^2(D;\mathbb{R}))$$
$$:= \{G : L^2(D;\mathbb{R}) \to L^2(D;\mathbb{R}) : G = (Id_{L^2(D;\mathbb{R})} + G_T) \circ \cdots \circ (Id_{L^2(D;\mathbb{R})} + G_1),$$
$$G_t \in \mathcal{NO}_{L_t,N,\varphi,\sigma}(L^2(D;\mathbb{R})), \ L_t \in \mathbb{N}, \ t = 1, ..., T\}, \tag{179}$$

$$\mathcal{RNO}_{T,N,\varphi,\delta,\sigma}^{inv}(L^2(D;\mathbb{R})) := \{G : L^2(D;\mathbb{R}) \to L^2(D;\mathbb{R}) :$$
$$G = (Id_{L^2(D;\mathbb{R})} + G_T) \circ \cdots \circ (Id_{L^2(D;\mathbb{R})} + G_1), G_t \in \mathcal{NO}_{L_t,N,\varphi,\sigma}(L^2(D;\mathbb{R})),$$
$$\mathrm{Lip}_{L^2(D;\mathbb{R}) \to L^2(D;\mathbb{R})}(G_t) \le \delta, \ L_t \in \mathbb{N}, \ t = 1, ..., T\}, \tag{180}$$

**Lemma 10.** *Let $\delta \in (0,1)$, and let $F \in \mathcal{RNO}_{T,N,\varphi,\delta,\sigma}^{inv}(L^2(D;\mathbb{R}))$. Let $\sigma : \mathbb{R} \to \mathbb{R}$ be Lipschitz continuous. If $\sigma : \mathbb{R} \to \mathbb{R}$ is Lipschitz continuous, then $F : X \to X$ is homeomorphism. Moreover, if $\sigma : \mathbb{R} \to \mathbb{R}$ is $C^1$, then, $F : X \to X$ is $C^1$-diffeomorphism.*

The proof is given by the same argument in Lemma 9.

**Lemma 11.** *Assume that the orthonormal basis $\varphi$ include the constant function. Then, we have the following inclusion:*

$$\mathcal{R}_{T,N,\varphi,\sigma}(L^2(D;\mathbb{R})) \subset \mathcal{RNO}_{T,N,\varphi,\sigma}(L^2(D;\mathbb{R})), \tag{181}$$

$$\mathcal{R}_{T,N,\varphi,\delta,\sigma}^{inv}(L^2(D;\mathbb{R})) \subset \mathcal{RNO}_{T,N,\varphi,\delta,\sigma}^{inv}(L^2(D;\mathbb{R})). \tag{182}$$

---

[6]We will discuss the function space $L^2(D;\mathbb{R})$ for easier reading, but all arguments can be replaced with real-valued function space $\mathcal{U}(D;\mathbb{R})$ that is a separable Hilbert space.

*Proof.* Let $G \in \mathcal{R}_{T,N,\varphi,\sigma}(L^2(D;\mathbb{R}))$ such that

$$G = (Id_{L^2(D;\mathbb{R})} + D_N \circ NN_T \circ E_N) \circ \cdots \circ (Id_{L^2(D;\mathbb{R})} + D_N \circ NN_1 \circ E_N).$$

Since $\varphi = \{\varphi_n\}_{n\in\mathbb{N}} \subset L^2(D;\mathbb{R})$ has the constant basis, denoting it by $\varphi_0 := \frac{\mathbf{1}_D(x)}{\|\mathbf{1}_D\|_{L^2(D;\mathbb{R})}}$, where $\mathbf{1}_D(x) = 1$ for $x \in D$, we see that for $\alpha = (\alpha_1,...,\alpha_N) \in \mathbb{R}^N$

$$D_N\alpha(x) = \sum_{n\leq N} \alpha_n\varphi_n(x) = \sum_{n\leq N} \alpha_n\langle\varphi_1,\varphi_1\rangle_{L^2(D;\mathbb{R})}\varphi_n(x)$$

$$= \sum_{n\leq N} \frac{1}{\|\mathbf{1}_D\|_{L^2(D;\mathbb{R})}}\langle\alpha_n\cdot\mathbf{1}_D,\varphi_1\rangle_{L^2(D;\mathbb{R})}\varphi_n(x).$$

We define $\tilde{D}_N : L^2(D;\mathbb{R})^N \to L^2(D;\mathbb{R})$ by for $u \in L^2(D;\mathbb{R})^N$

$$\tilde{D}_N u := \sum_{p,q\leq N} \tilde{D}_{p,q}\langle u,\varphi_p\rangle_{L^2(D;\mathbb{R})}\varphi_q,$$

where $\tilde{D}_{p,q} \in \mathbb{R}^{1\times N}$ is defined by

$$\tilde{D}_{p,q} = \begin{cases} \left(0,...,0,\underbrace{\frac{1}{\|\mathbf{1}_D\|_{L^2(D;\mathbb{R})}}}_{q-th},0,...,0\right), & p = 1 \\ O, & p \neq 1. \end{cases}$$

Then, we have that

$$D_N \circ NN_t \circ E_N(u) = \tilde{D}_N\left(NN_t \circ E_N(u)\cdot\mathbf{1}_D\right). \tag{183}$$

Next, we see that

$$E_N(u) = \left(\langle u,\varphi_1\rangle_{L^2(D;\mathbb{R})},...,\langle u,\varphi_N\rangle_{L^2(D;\mathbb{R})}\right) = \|\mathbf{1}\|_{L^2(D;\mathbb{R})}\left(\langle u,\varphi_1\rangle_{L^2(D;\mathbb{R})},...,\langle u,\varphi_N\rangle_{L^2(D;\mathbb{R})}\right)\varphi_1(x),$$

We define $\tilde{E}_N : L^2(D;\mathbb{R}) \to L^2(D;\mathbb{R})^N$ by

$$\tilde{E}_N(u) := \sum_{p,q\leq N} \tilde{E}_{p,q}\langle u,\varphi_p\rangle_{L^2(D;\mathbb{R})}\varphi_q(x),$$

where $\tilde{E}_{p,q} \in \mathbb{R}^{N\times 1}$ is defined by

$$\tilde{E}_{p,q} = \begin{cases} \left(0,...,0,\underbrace{\langle\mathbf{1}_D,\varphi_n\rangle_{L^2(D;\mathbb{R})}}_{p-th},0,...,0\right)^T, & q = 1 \\ O, & q \neq 1. \end{cases}$$

Then we have that

$$\tilde{D}_N\left(NN_t \circ E_N(u)\cdot\mathbf{1}_D\right) = \tilde{D}_N \circ NN_t \circ \tilde{E}_N(u). \tag{184}$$

With (183) and (184), we see that

$$D_N \circ NN_t \circ E_N = \tilde{D}_N \circ NN_t \circ \tilde{E}_N \in \mathcal{NO}_{L_t,N,\varphi,\sigma}(L^2(D;\mathbb{R})),$$

where $L_t \in \mathbb{N}$ is the depth of $NN_t$. Therefore, $G$ has the form

$$G = (Id_{L^2(D;\mathbb{R})} + \tilde{D}_N \circ NN_T \circ \tilde{E}_N) \circ \cdots \circ (Id_{L^2(D;\mathbb{R})} + \tilde{D}_N \circ NN_1 \circ \tilde{E}_N) \in \mathcal{RNO}_{T,N,\varphi,\sigma}(L^2(D;\mathbb{R})).$$

(182) can be proved by the same argument. $\qquad\square$

## D  Generalizations

### D.1  Generalization of the no-go Theorem 2 using weak topology

We can generalize the no-go Theorem 2 for the case when approximations and continuity of approximations are considered in the weak topology of the Hilbert space $X$. This can be done when the approximations of the identity map and the minus one times the identity operator satisfy additional assumptions and the partially ordered subset $S_0(X)$ is the set of all all linear subspaces $S(X)$ of $X$.

In the case when $S_0(X) = S(X)$ and Definition 7 the condition (A) can generalized as follows:

**Definition 11** (Weak Approximation Functor). *When $S_0(X) = S(X)$ we define the* weak approximation functor, *that we denote by $\mathcal{A}\colon \mathcal{D} \to \mathcal{B}$, as the functor that maps each $(X, F) \in \mathcal{O}_\mathcal{D}$ to some $(X, S(X), (F_V)_{V \in S(X)}) \in \mathcal{O}_\mathcal{B}$ so that the Hilbert space $X$ stays the same. The approximation functor maps all morphisms $a_\phi$ to $A_\phi$ and morphisms $a_{X_1, X_2}$ to $A_{X_1, X_2}$, and has the following properties*

(A') *For all $r > 0$, all $(X, F) \in \mathcal{O}_\mathcal{D}$ and all $y \in X$, it holds that*

$$\lim_{V \to X} \sup_{x \in \overline{B}_X(0,r) \cap V} \langle F_V(x) - F(x), y \rangle_X = 0. \tag{185}$$

*Moreover, when $F : X \to X$ is the operator $Id : X \to X$ or $-Id : X \to X$, then $F_V$ is the operator $Id_V : V \to V$ or $-Id_V : V \to V$, respectively.*

Moreover, Definition 8 can generalized as follows so that it uses the weak topology.

**Definition 12.** *We say that the approximation functor $\mathcal{A}$ is continuous in the weak topology if the following holds: Let $(X, F), (X, F^{(j)}) \in \mathcal{O}_\mathcal{D}$ be such that the Hilbert space $X$ is the same for all these objects and let $(X, S(X), (F_V)_{V \in S(X)}) = \mathcal{A}(X, F)$ be approximating sequences of $(X, F)$ and $(X, S(X), (F_{j,V})_{V \in S(X)}) = \mathcal{A}(X, F^{(j)})$ be approximating sequences of $(X, F^{(j)})$. Moreover, assume that for $r > 0$ and all $y \in X$*

$$\lim_{j \to \infty} \sup_{x \in \overline{B}_X(0,r)} |\langle F^{(j)}(x) - F(x), y \rangle_X| = 0. \tag{186}$$

*Then, for all $V \in S(X)$ the approximations $F_V^{(j)}$ of $F^{(j)}$ and $F_V$ of $F$ satisfy for all $y \in X$*

$$\lim_{j \to \infty} \sup_{x \in V \cap \overline{B}_V(0,r)} |\langle F_V^{(j)}(x) - F_V(x), y \rangle_X| = 0. \tag{187}$$

The theorem below states a negative result, namely that there does not exist continuous approximating functors for diffeomorphisms.

**Theorem 8.** *(No-go theorem for discretization of general diffeomorphisms) There exists no functor $\mathcal{D} \to \mathcal{B}$ that satisfies the property (A') of a weak approximation functor and is continuous in the weak topology.*

The proof of Theorem 8 is analogous to Theorem 2 by replacing $A_1$ by the map $-Id$ and considering a linear space $V \in S(X)$ having an odd dimension, in which case $\deg(A_1 : V \to V) = -1$. Let $F_0 = Id : X \to X$ and $F_1 = -Id : X \to X$. Assume that $(X, S(X), (F_{t,V})_{V \in S(X)}) = \mathcal{A}(X, F_t)$ are approximations of the map $F : X \to X$ in the weak topology and that $\mathcal{A}$ is continuous in the weak topology. Recall that then $F_{t,V} : V \to V$ are $C^1$-diffeomorphisms that are discretizations of $F : X \to X$. Then, by condition (A'), $F_{0,V} = Id_V$ and $F_{1,V} = -Id_V$ so that $\deg(F_{0,V}) = 1$ and $\deg(F_{1,V}) = -1$. Observing that when the condition (187) is applied for $y_1, y_2, \ldots, y_n \in V$ that form a basis of the space $V$ having the dimension $\dim(V) = n$, we see that the condition (187) implies the condition (7). Using these observations, Theorem 8 is follows similarly to the proof of Theorem 2.

