# OpenReview forum: "Can neural operators always be continuously discretized?"
_NeurIPS.cc/2024/Conference — NeurIPS 2024 poster_

### Official Review · Reviewer_47x8 · 2024-06-17

**Soundness:** 3
**Presentation:** 3
**Contribution:** 3
**Rating:** 6
**Confidence:** 2

**Summary:**

This paper studies the question of whether a neural operator (or a general diffeomorphism on an infinite dimensional Hilbert space) can be continuously discretized through the lens of category theory. It first proves that there does not exist a continuous approximation scheme for all diffeomorphisms. Then, it shows that neural operators with strongly monotone layers can be continuously discretized, followed by a proof that a biliptschitz neural operator can be approximated by a deep one with strongly monotone layers. Some further consequences are discussed.

**Strengths:**

1. The paper studies the discretization of a continuous operator, which is a source of error and an important issue in operator learning if one is not careful.
2. The paper is fairly comprehensive, encompassing both positive and negative results. The study of positive results contains theorems of different flavors.
3. Although the theory is based on the category theory, the presentation and explanation of the results are relatively clear and accessible for people who are unfamiliar with it.

**Weaknesses:**

1. Most results presented in the paper are purely theoretical and lack a quantitative or asymptotic estimate. For example,
   * the notion of continuous approximation functor in Definition 8 does not care about the rate of convergence, and
   * there is no estimate for the number of layers $J$ in Theorem 4.
2. No empirical results are there to support the theory. While this is a theoretical paper, some toy experiments that exemplify the theory would be very helpful.

**Questions:**

1. In general, is there any assumption of the Hilbert space studied in this paper? For example, are the Hilbert spaces assumed to be separable?
2. What is the definition of the convergence of finite-dimensional subspaces used in this paper? Strong convergence of the projection operator? I do not think there is a standard definition for this in elementary functional analysis so it would be helpful to say it explicitly in the paper.
3. Have you studied the role of the bilipschitz constant in your theorems? For example, how does the number of layers $J$ in Theorem 4 depend on it?

**Limitations:**

None.

---

> ### Author Rebuttal · Authors · 2024-08-06
>
> We would like to thank the reviewer for the detailed comments and fair criticisms. We address all of these below:
>
> 1. "Most results presented in the paper are purely theoretical and lack a quantitative or asymptotic estimate."
>
> The proof of Theorem 4 makes it possible to estimate the number $J$ of layers as a function of $\epsilon$, the Lipschitz constants of the map $F$ and $F^{-1}$, the $C^2$-norm of $F:B(0,2r_1)\to X$ in  a ball having double the radius $r_1$ where we consider the approximation of $F$. More precisely, $$ J\leq C\epsilon^{-2} $$ where $C$ depends on the radius $r_1$ of the ball where $F$ is approximated and the
> $C^2$-norm of the map $F$ in a ball of radius $2r_1$. We will add an explicit formula for $C_0(r_1,\|F\|_{B(0,2r_1);X)})$ and its proof in the final version of the paper.
>
> The main steps of the proof are the following: First, we use spectral theory of the compact operators $T_1$ and $T_2$ to find a finite dimensional subspace $W$ and the projection $P_W$ onto it so that $$\hat F(x)=(I-P_W)x+P_WF(P_WF) $$ is a diffeomorphism that is close to the operator $F:X\to X$. Let $f:W\to W$ be the restricition of $\hat F$ to $W$. After this we deform  to the invertible linear map $A_1:W\to W$ is the derivative of $f$at $x=0$, along the path $t\to f_t,$ $$f_t(x):=\frac 1t(f(tx)-f(0))+tf(0),\quad\hbox{for }t>0,$$ $$ f_t(x):=A_1,\quad\hbox{for }t=0.$$ All operators $f_t:W\to W$ are bi-Lipschitz maps, $f_0(x)=f(x)$ and $f_0(x)=A_1$. We consider the values $t_1=c_1\epsilon$ and  $t_j=t_1+jh$, for $j=2,\dots,J_1$, and the operators $$ Id+B_j=f_{t_{j+1}}\circ f_{t_j}^{-1}\ : W\to W,\quad j=0,1,2,\dots,J_1.$$ We show that in the ball $B_W(0,2r_1)$ $$\hbox{Lip}(B_0)=\hbox{Lip}(f_{t_1}\circ A_1^{-1}-Id)\le C_2t_1=C_2c_1\epsilon.$$ Here, $C_2$ depends on $r_1$ and $C^2$-norm of $F$ as well as on the Lipschitz constants of $F$ and $F^{-1}$ and $c_1$ is chosen to be sufficiently small.
>
> Moreover, we show that for $j\ge 1$, $$\hbox{Lip}(B_j)\leq C\frac 1{t_j}(t_{j+1}-t_j)=C\frac 1{t_j}h,$$ where the factor $\frac 1{t_j}$ appears due to the multiplier $\frac 1t$ in the definition of $f_t$. To obtain $\hbox{Lip}(B_j)\le \epsilon$, we choose $h\le c_2\epsilon^2$. This causes the factor $\epsilon^{-2}$ in the bound for $J$.
>
> In addition to this, we consider paths on the Lie group $O(n)$, $n=\dim(W)$ and show that there are sequence of invertible matrixes $A_j\in O(n)$, $j=1,2,\dots,J_2$ such that  $$ Id+B_{J_1+j}=A_{j+1}\circ A_j^{-1}\ :W\to W$$ where $\hbox{Lip}(B_j)<\epsilon$ and $A_j$, where $j=J_2$, is either the identity operator $Id:W\to W$ or a reflection operator $B_e:x\to x-2\langle x,e\rangle e$. Here, we can choose the number of steps to be  $J_2\leq c_3\epsilon^{-1}.$ Combining the  operators $Id+B_j$ with $j=1,\dots, J_1+J_2$, we see that $F$ can be deformed to a linear operator or a reflection operator by combining $J=J_1+J_2$ operators of the form $Id+B_j$. This yields the bound for $J$.
>
> 2. "While this is a theoretical paper, some toy experiments that exemplify the theory would be very helpful."
>
> We much appreciate the reviewer's comment.  In Appendix A.1 we had given an example where we approximate the solution operator $x(t)\to u(t)$ of the nonlinear elliptic equation $$ \partial_t^2 u(t)-g(u(t))=x(t),\quad t\in (0,1), $$ with $u=0$ on the boundary using a discretization that is based on Finite Element Method. When $g$ is convex and the source term $x(t)$ is represented in the form $$x(t)=\frac {d^2}{dt^2}h(t),$$ the map $F:h\to u$ is a diffeomorphism in the Sobolev space $H^2$ with the Dirichlet boundary values. The approximation $F\to F_V$ can be obtained by Galerkin method. We will expand upon this example in the final version of the manuscript. We will give an example on no-go theorem using the elliptic (but not not strongly elliptic) problem $$B_su:=-\frac d{dt}\bigg(\hbox{sign}(t-s)\frac d{dt} u(t)\bigg)=f(t),\quad t\in [0,1],$$ $$u(0)=0,\ \ \frac d{dt} u(1)=0.$$ For all $0\le s\le 1$ these equations are uniquely solvable, but we show that when we use FEM to approximate those, some of the obtained finite dimensional problems has also a zero eigenvalue and is not solvable.
>
> 3. "Do the Hilbert spaces need to be separable?"
>
> The no-go theorem applies also to non-separable Hilbert spaces. Naturally, for such spaces the partially ordered set $S(X)$ of finite dimensional linear subspaces of $X$, that is used as an index set, is huge. In most of our positive results on existence of approximation operations we have assumed that the Hilbert space is separable as we use finite rank neural operators as approximators, and have used this in the orthogonal projectors $P_n$ form $X$ to $\hbox{span}(\phi_1,\dots,\phi_n)$, where $\phi_j,$ $j \in \mathbb Z_+$, is an enumerable orthonormal basis. However, it seems to us that our results can be generalized to non-separable Hilbert spaces $X$ that have a non-enumerable orthonormal bases. We will check carefully if this generalization is possible.
>
> 5. "What is the definition of the convergence of finite-dimensional subspaces used in this paper?"
>
> After Definition 7, line 223, we defined that the limit $$ \lim_{V\to X} y_V=y. $$ This limit can be defined also by endowing the set $S_0(X)\cup \{X\}$ by the topology associated to  the partial ordering of  $S_0(X)\cup \{X\}$, that is, the topology generated by the sets $U_V:=\{W\in S_0(X):\ X\supset V\\}\cup \{X\}$.
>
> 6. "Have you studied the role of the bi-Lipschitz constant in your theorems? For example, how does the number of layers $J$ in Theorem 4 depend on it?"
>
> The bi-Lipschitz constraint (and form of neural operator layers) enable us to decompose the map into strongly monotone neural operator layers $G_k=Id+B_k$ (Theorem 4), where $\mathrm{Lip}(B_k)<\epsilon$. Here, $\epsilon \in (0,1)$ is arbitrary. The bi-Lipschitz constant appears in the proof of the estimate of $J$ (under 1.) We will mention this observation in the final version of the manuscript.

---

> > ### Comment · Reviewer_47x8 · 2024-08-12
> >
> > Thank you for the detailed response. Since I am not absolutely familiar with category theory and other related work, I am unable to further raise my score, but I acknowledge that I have read through the rebuttal and it appears to be a nice paper overall.

---

> ### Comment · Reviewer_KEed · 2024-08-13
>
> Well, I felt sorry for the authors because a reviewer of NeurIPS, once well-known for a top theory-oriented conference of machine leaning, **cannot raise his/her score** simply because he/she is **not familiar with the category theory**. I even feel this is symbolic of the current state of a theory-oriented machine learning conference. It should not be the problem of the individual reviewer him/herself, but the problem of conference's matching systems that mistakenly assign a perfect amateur of the category theory for reviewing a category theoretic research.
>
> This is a suggestion for chairs for future avoidance of mismatches, that the reviewers should be examined if they have fundamental knowledge/background/understandings in the field. I am an expert of expressive power analysis, but not at all of category theory and tropical geometry. Unfortunately, this kind of mismatch happens every year, so I am usually skeptical to any mathematical ''theorems'' published in machine learning conferences.

---

### Official Review · Reviewer_K5mB · 2024-06-26

**Soundness:** 3
**Presentation:** 3
**Contribution:** 3
**Rating:** 7
**Confidence:** 4

**Summary:**

This paper focuses on the continuous discretization in operator learning. This is a very important question since it involves reducing the infinite-dimensional space to a finite-dimensional space in operator learning. The authors present cases where discretization is continuous and cases where it is not. The results are interesting and can be applied to design methods in operator learning.

**Strengths:**

The proof is solid, and the paper is well-written and organized. I appreciate the results presented in this paper.

**Weaknesses:**

Since this paper is submitted to NeurIPS and not a mathematical journal, I hope the authors can provide some practical examples, such as solving the Poisson equation \(\Delta u = f\) to learn the operator relationship between \(f\) and \(u\). By using methods like DeepONet and FNO, it would be beneficial to determine whether the discretization in these methods is continuous or not. I believe this could make the paper more accessible to a broader audience.

**Questions:**

Mentioned in the Weakness.

**Limitations:**

All right.

---

> ### Author Rebuttal · Authors · 2024-08-06
>
> We thank the reviewer for the suggestion to include an example based on the discretization of simple differential equations as it surely helps the readers to quickly understand the essential features of the no-go theorem of the approximation of invertible operators.
>
> On the positive results, in Appendix A of our paper, we have considered nonlinear discretiation of the operators $u\to -\Delta u+G(u)$; see the reply to reviewer mJde under point 3. To exemplify the negative result, we will add in the appendix of the paper the following example on the solution operation of differential equations and the non-existence of approximation by diffeomorphic maps: We consider the elliptic (but not not strongly elliptic) problem (below, called as "PDE1")
> $$
> B_su:=-\frac d{dt}\bigg((1+t)\hbox{sign}(t-s)\frac d{dt} u(t)\bigg)=f(t),\quad t\in [0,1],
> $$
> with the Dirichlet and Neumann boundary conditions
> $$
> u(0)=0,\quad \frac d{dt} u(1)=0.
> $$
> Here, $0\le s\le 1$ is a parameter of the coeffient function and $\hbox{sign}(t-s)=1$ if $t>s$ and $\hbox{sign}(t-s)<0$ if $t<s$. We consider the weak solutions of PDE1 in the space
> $$
> u\in H^1_{D,N}(0,1):=\\{v\in H^1(0,1):\ v(0)=0\\}.
> $$
> We can write
> $$
> B_su=-D_t^{(2)}A_sD_t^{(1)}u,
> $$
> where
> $$
> A_sv(t)=(1+t)\hbox{sign}(t-s)v(t),
> $$
> parametrized by $0\le s\leq 1$, are multiplication operations that are invertible operators, $A_s:L^2(0,1)\to L^2(0,1)$ (this invertibility makes the equation PDE1 elliptic). Moreover, $D_t^{(1)}$ and $D_t^{(2)}$ are the operators $v\to \frac {d}{dt}v$ with the Dirichlet boundary  condition $v(0)=0$ and $v(1)=0$, respectively. We consider the Hilbert space $X=H^1_{D,N}(0,1)$; to generate an invertible operator $G_s:X\to X$ related to PDE1, we write the source term using an auxiliary function $g$,
> $$
> f(t)=Qg:=-\frac {d^2}{dt^2}g(t)+g(t).
> $$
> Then the equation,
> $$
> B_su=Qg ,
> $$
> defines a continuous and invertible operator,
> $$
> G_s:X\to X,\quad G_s:g\to u.
> $$
> In fact, $G_s=B_s^{-1}\circ Q$ when the domains of $B_s$ and $Q$ are chosen in a suitable way. The Galerkin method (that is, the standard approximation based on the Finite Element Method) to approximate the equation PDE1  involves introducing a complete basis $\chi_j(t)$, $j=1,2,\dots$ of the Hilbert space $X$, the orthogonal projection
> $$
> P_n:X\to X_n:= \hbox{span}\\{\chi_j:\ j=1,2,\dots,n\\} ,
> $$
> and approximate solutions of PDE1 through solving
> $$
> P_nB_sP_nu_n=P_nQP_ng_n,\quad u_n\in X_n, \ g_n=P_ng.
> $$
> This means that operator $B_s^{-1}Q:g\to u$ is approximated by $(P_nB_sP_n)^{-1}P_nQP_n:g_n\to u_n$, when $P_nB_sP_n:X_n\to X_n$ is invertible.
>
> The above corresponds to the Finite Element Method where the matrix defined by the operator $P_nB_sP_n$
> is  $m(s)= [b_{jk}(s)]_{j,k=1}^n\in \mathbb R^{n\times n}$, where
> $$
> m(s)=\int_0^1 (1+t)\hbox{sign}(t-s)\frac d{dt} \chi_j(t)\cdot \frac d{dt} \chi_k(t)dt,\quad j,k=1,\dots,n.
> $$
> Since we used the  mixed Dirichlet and Neumann boundary conditions in the above boundary value problem, we see that for $s=0$ all eigenvalues of the matrix $m(s)$ are strictly positive, and when $s=1$ all eigenvalues  are strictly negative. As the function $s \to m(s)$ is a continuous matrix-valued function, we see that there exists $s\in (0,1)$ such that  the matrix $m(s)$ has a zero eigenvalue and is no invertible. Thus, we have a situation where all operators $B_s^{-1}Q:g\to u$, $s\in [0,1]$ are invertible (and thus define diffeomorphisms $X\to X$) but for any basis $\chi_j(t)$ and any $n$ there exists $s\in (0,1)$ such that the finite dimensional approximation $m(s):\mathbb R^n\to \mathbb R^n$ is not invertible. This example shows that there is no FEM-based discretization method for which the finite dimensional approximations of all operators $B_s^{-1}Q$,  $s\in (0,1)$, are invertible. The above example also shows a key difference between finite and infinite dimensional spaces. The operator $A_s:L^2(0,1)\to L^2(0,1)$ has only continuous spectrum and not eigenvalues nor eigenfunctions whereas the finite dimensional matrices have only point spectrum (that is, eigenvalues). The  continuous spectrum makes it possible to deform the  positive operator $A_s$ with $s=0$  to a negative operator $A_s$ with $s=1$ in such a way that all operators $A_s$, $0\le s\le 1$, are invertible but this is not possible to do for finite dimensional matrices. We point out  that the map $s\to A_s$ is not continuous in the operator norm topology but only in the strong operator topology and the fact that $A_0$ can be deformed to $A_1$ in the norm topology by a path that lies in the set of invertible operators is a deeper result. However, the strong operator topology is enough to make the FEM matrix  $m(s)$ to depend continuously on $s$.

---

> > ### Comment · Reviewer_K5mB · 2024-08-07
> >
> > Thanks for the reply. I will keep my score.

---

### Official Review · Reviewer_KEed · 2024-07-12

**Soundness:** 3
**Presentation:** 3
**Contribution:** 3
**Rating:** 6
**Confidence:** 3

**Summary:**

This paper investigates theoretical limitations of discretizing neural operators on infinite-dimensional Hilbert spaces. The authors first prove a "no-go theorem" (Theorems 1,2) showing that diffeomorphisms between infinite-dimensional Hilbert spaces cannot generally be continuously approximated by finite-dimensional diffeomorphisms. Then, they provide positive results for certain classes of operators such as strongly monotone (Theorem 3) and bilipschitz neural operators (Theorem 4). They finally provide concrete example of approximation by finite residual ReLU networks (Theorem 5).

**Strengths:**

- The universality of neural networks has been demonstrated in various settings. However, research on the approximation abilities of operators is relatively scarce. Particularly, the characterization of classes that cannot be approximated is intriguing. This study is important as it succinctly demonstrates the differences between finite-dimensional and infinite-dimensional properties in the manageable setting of Hilbert spaces.

- Moreover, the novel approach of expressing approximation sequences in terms of category theory is noteworthy.

**Weaknesses:**

- On the other hand, the proofs are based on conventional analytical arguments rather than category-theoretic arguments. Therefore, the "category theory" framework might be somewhat exaggerated. It is expected that with refinement of notation and sentence structure, the description could become more perspicuous in the future.

- There is concern that the categorical description may have obscured the contributions typically seen in __traditional approximation theory__ papers. As the authors likely recognize, various topologies are used in function approximation, and this study focuses __only__ on approximation in the norm topology of Hilbert spaces, and does not negate "all considerable approximation sequences". So, the impossibility theorem presented here might simply be due to the norm topology being too strong. While the Hilbert structure sounds natural as a generalization of Euclidean structure, in reality, concepts like L2 convergence of Fourier series are quite technical and not necessarily an inevitable notion of convergence. It seems that in pursuit of an elegant categorical description, the diversity of function approximation may have been compromised.

**Questions:**

In Definition 3, why $\sigma$ is imposed besides $G$?

**Limitations:**

The authors did not discuss the validity of assumptions.

---

> ### Author Rebuttal · Authors · 2024-08-06
>
> We appreciate the detailed suggestions, criticisms and endorsement of the reviewer. We address all of these below:
>
> 1. "On the other hand, the proofs are based on conventional analytical arguments rather than category-theoretic arguments."
>
> The proofs are indeed based on analytical arguments. We used category theory as a formalism (similar to the one for object oriented programming) to describe approximation operations in all Hilbert spaces (including non-separable ones). As the collection of all Hilbert spaces cannot be considered as a set (cf. Russel's paradox) but can as a category, we chose to use the language of category theory. In the beginning of the paper, we considered an "approximation operation" to avoid difficulties related to formal category theory.
>
> 2. "The impossibility theorem presented here might simply be due to the norm topology being too strong."
>
> We much appreciate the issue raised by the reviewer. We will include the analysis and discussion below in the revised manuscript, in an appendix on generalizations.
>
> We formulated the approximation functor using norm topology as uniform convergence in compact sets is extensively studied in the theory of neural networks. Norm topology also makes it possible to consider quantitative error estimates. However, we agree with the referee that it is important to understand no-go results in weaker topologies. It turns out that our results can be generalized to a setting where the norm topology is partially replaced by the weak topology. Definition 7 is replaced by the following
>
> Definition [Weak Approximation Functor]
> When $S_0(X)=S(X)$  we define the _weak approximation functor_, that we denote by $\mathcal A: \mathcal D\to\mathcal B$, as the functor that maps each$(X,F)\in\mathcal O_{\mathcal D}$ to some$(X,S(X),(F_V)_{V \in S(X)})$ and has the following  the properties
>
> (A')
> For all $r>0$, all $(X,F)\in O_{\mathcal D}$ and all $y\in X$, it holds that$$\lim_{V\to X}\ \sup_{x\in B(0,r)\cap V} \ \langle F_V(x)-F(x),y\rangle_X=0.$$Moreover, when $F:X\to X$ is the operator $Id:X\to X$ or  $-Id:X\to X$, then $F_V$ is  the operator $Id_V:V\to V$ or  $-Id V:V\to V$, respectively.
>
> In (A') we added conditions on the approximation on the operators $Id$ and $-Id$. Similarly, the continuity of the approximation functors can be generalized in the case where the convergence in the norm topology is replaced by the weak topology. The proof of the no-go theorem generalizes also to this setting; we will add in the Appendix a theorem which states that there are no weak approximation functors that are continuous in the weak topology.
>
> 3. In Definition 3, why $\sigma$ is imposed besides $G$?
>
> This definition needs to be interpreted with care, which we will clarify in the revised manuscript.  The appearance of the compact operators $T_1$ and $T_2$ makes the discretization of activation function $\sigma$ and the activation functions inside $G$ in Definition 3 different, and this is one reason why we have introduced both $\sigma$ and $G$. To consider invertible neural operators, we will below assume that $\sigma$ is an invertible function, for example, the leaky Relu function. In the  operation $$N:u\to u +T_2(G(T_1u)),$$ the nonlinear function $G$ is sandwiched between compact operators $T_1$ and $T_2$. The compact operators map weakly converging sequences to norm converging sequences. This is essential in the proofs of the positive results for approximation functors as discussed in the paper. However, we do not have general results on how the operation $$u\to \sigma \circ u$$ can be approximated by finite dimensional operators in the norm topology, but only in the weak topology in the sense of the above
> {Definition} 1 of the Weak Approximation Functor. Nonetheless, one can overcome this difficulty, for example, for using the explicit form of the activation function and choosing different finite dimensional spaces $V_j$ in each layer of the neural operator.
>
> We address the question whether the activation function $\sigma$ is relevant in universal approximation results. If the activation function $\sigma$ is removed, the operator $F$ becomes a sum of a (local) linear operator and a compact (nonlocal) nonlinear integral operator. Moreover, if we compose above operators of the above form, the resulting operator, $H$ say, is also a sum of a (local) linear operator, $W$, and a compact (nonlocal) operator, $K$. The Fr\'{e}chet derivative of $H$ at $u_0$ is equal to $W$ and a compact linear operator. This means that the Fredholm index of the derivative of $H$ at $u_0$ equal to the index of $W$ is constant, that is, independent of the point $u_0$ where the derivative is computed. In particular, this means that one cannot approximate an arbitrary $C^1$-function $X\to X$ in compact subsets of $X$ by such neural operators. Indeed, for a general $C^1$-function, the Fredholm index may be a varying function of $u_0$. Thus, $\sigma$ appears to be relevant for obtaining universal approximation theorems for neural operators. Again, we will add this analysis to the final version of the manuscript.
>
> 4. "The authors did not discuss the validity of assumptions."
>
> We appreciate this criticism and will address it in the final version of the manuscript. The key assumption is that the neural operator is bilipschitz while being of the general form (4). the expressability properties and applicability in designing generative models is discussed in global comments. We also point of that the strong monotonicity used as an assumption in several lemmas and theorems is an intermediate assumption that is absorbed in Theorem 4 where we consider approximation of bi-Lipschitz neural operators.
>
> In Theorem 5, finite-rank residual neural operators appear as explicit natural approximators of bilipschitz neural operators. Such a perspective has been empirically studied as in [Behrmann, et al   PMLR 2019, pp. 573-582], although in the finite-dimensional case.

---

> > ### Comment · Reviewer_KEed · 2024-08-12
> >
> > Thank you for detailed clarifications. I would like to keep my score as is.
> >
> > > The proofs are indeed based on analytical arguments.
> >
> > If so, I recommend the authors to reconsider the following phrases in the abstract and conclusion:
> >
> > > Using category theory, we give a no-go theorem
> > > We used tools from category theory to produce a no-go theorem
> >
> > It would be much impactful and significant if the authors could more directly point out any incorrectness of the proof or inappropriateness of the assumption in the previous studies.

---

> > > ### Author Response · Authors · 2024-08-13
> > >
> > > Thank you for your response. We would will address your points in the following way.
> > >
> > > - If so, I recommend the authors to reconsider the following phrases in the abstract and conclusion: "Using category theory, we give a no-go theorem." "We used tools from category theory to produce a no-go theorem"
> > >
> > > We appreciate the advice, and will follow it. We will replace
> > >
> > > "Using category theory, we give a no-go theorem"
> > >
> > > with
> > >
> > > "Using analytical arguments, we give a no-go theorem framed with category theory."
> > >
> > > and replace
> > >
> > > "We used tools from category theory to produce a no-go theorem"
> > >
> > > with
> > >
> > > "We give a no-go theorem framed with category theory"
> > >
> > > in the abstract and conclusion.
> > >
> > > - It would be much impactful and significant if the authors could more directly point out any incorrectness of the proof or inappropriateness of the assumption in the previous studies.
> > >
> > > There are several papers which use continuous functions (either as elements of infinite dimensional function spaces or metric spaces) to model images or signal and apply statistical methods and invertible neural networks or maps modeling diffeomorphisms. Often in these papers one derives theoretical results in the continuous models and presents numerical results using a finite dimensional approximations. In this process the errors are caused by the discretization and the effect of changing the dimension of the approximate models. We believe that our work meaningfully addresses these questions as applied to injective/bijective neural operators, an important architecture. We hope that our paper inspires further study these points. We can include citations to the following papers, related to these issues.
> > >
> > > The below papers which combine neural networks and approximation of diffeomorphisms, as applied to imaging.
> > >
> > > - Elena Celledoni · Helge Glöckner · Jørgen N. Riseth, Alexander Schmeding
> > > Deep neural networks on diffeomorphism groups for
> > > optimal shape reparametrization.
> > > BIT Numerical Mathematics (2023) 63:50
> > >
> > > - GradICON: Approximate Diffeomorphisms via Gradient Inverse Consistency
> > > Lin Tian · Hastings Greer · François-Xavier Vialard · Roland Kwitt · Raúl San José Estépar · Richard Jarrett Rushmore · Nikolaos Makris · Sylvain Bouix · Marc Niethammer
> > > West Building Exhibit Halls ABC 153
> > >
> > > The below papers combine invertible neural networks and statistical models, especially for solving inverse problems (including imaging problems).
> > >
> > > - Alexander Denker , Maximilian Schmidt , Johannes Leuschner and Peter Maass
> > > Conditional Invertible Neural Networks for Medical Imaging. Journal of Imaging 2021, 7(11), 243
> > >
> > > - Ardizzone, L.; Kruse, J.; Rother, C.; Köthe, U. Analyzing Inverse Problems with Invertible Neural Networks. In Proceedings of
> > > the 7th International Conference on Learning Representations (ICLR 2019), New Orleans, LA, USA, 6–9 May 2019.
> > >
> > > - Anantha Padmanabha, G.; Zabaras, N. Solving inverse problems using conditional invertible neural networks. J. Comput. Phys.
> > > 2021, 433, 110194
> > >
> > > - Denker, A.; Schmidt, M.; Leuschner, J.; Maass, P.; Behrmann, J. Conditional Normalizing Flows for Low-Dose Computed
> > > Tomography Image Reconstruction. In Proceedings of the ICML Workshop on Invertible Neural Networks, Normalizing Flows,
> > > and Explicit Likelihood Models, Vienna, Austria, 18 July 2020.
> > >
> > > - Hagemann, P.; Hertrich, J.; Steidl, G. Stochastic Normalizing Flows for Inverse Problems: A Markov Chains Viewpoint.
> > > SIAM/ASA Journal on Uncertainty QuantificationVol. 10, Iss. 3 (2022) 10.1137
> > >
> > > - Papamakarios, G.; Nalisnick, E.T.; Rezende, D.J.; Mohamed, S.; Lakshminarayanan, B. Normalizing Flows for Probabilistic
> > > Modeling and Inference. Journal of Machine Learning Research 22 (2021) 1-64

---

### Official Review · Reviewer_mJde · 2024-07-15

**Soundness:** 2
**Presentation:** 1
**Contribution:** 2
**Rating:** 4
**Confidence:** 2

**Summary:**

The paper addresses the problem of discretizing neural operators, maps between infinite dimensional Hilbert spaces that are trained on finite-dimensional discretizations.  Using tools from category theory, the authors provide a no-go theorem showing that diffeomorfisms between Hilbert spaces may not admit continuous approximations by diffeomorfisms on finite spaces. This highlights the fundamental differences between infinite-dimensional Hilbert spaces and finite-dimensional vector spaces. Despite these challenges, the authors provide positive results, showing that strongly monotone diffeomorphism operators can be approximated in finite dimensions and that bilipschitz neural operators can be decomposed into strongly monotone operators and invertible linear maps. Finally, they observe how such operators can be locally inverted through an iteration scheme.

**Strengths:**

- The paper provides  theoretical results addressing the challenging problem of discretizing inherently infinite-dimensional objects (neural operators)

**Weaknesses:**

- The text and presentation require significant polishing.  It contains numerous typos, poorly formulated sentences, and instances of missing or repeated words
- While the paper's theoretical focus is valuable, it lacks  examples of specific neural operator structures that meet the theorems or remarks
- A more detailed  discussion on the  practical impact of this work, accompanied by examples, would be beneficial for the audience

Please note that my review should be taken with  caution, as I am not  familiar with category theory and did not thoroughly check the mathematical details. My feedback primarily focuses on the presentation and potential impact of the results rather than a rigorous validation of the theoretical content.

**Questions:**

- Neural operators are typically defined between Banach spaces. Why does your theory focus on maps between Hilbert spaces instead?
- Comment: The work in [1] might have been relevant to cite as well.
- The main neural operator paper [2] develops theoretical results on the universal approximation theory of neural operators. How do your results relate to the ones in that paper?

[1] F. Bartolucci, E. de Bézenac, B. Raonić, R. Molinaro, S. Mishra, R. Alaifari, Representation Equivalent Neural Operators: a Framework for Alias-free Operator Learning, NeurIPS 2023.

[2] Nikola B. Kovachki, Zongyi Li, Burigede Liu, Kamyar Azizzadenesheli, Kaushik Bhattacharya, Andrew M. Stuart, and Anima Anandkumar. "Neural operator: Learning maps between function spaces with applications to PDEs," J. Mach. Learn. Res., 24(89):1–97, 2023.

**Limitations:**

The paper lacks  examples of applications of the theorems to specific neural operator structures, and some further discussion on the practical impact of the results with examples

---

> ### Author Rebuttal · Authors · 2024-08-06
>
> We appreciate the valuable comments and constructive feedback of the reviewer. We are pleased to address all of these below.
>
> 1. "The text and presentation require significant polishing."
>
> We agree and sincerely regret this, and have already made many corrections to the manuscript.
>
> 2. "While the paper's theoretical focus is valuable, it lacks examples of specific neural operator structures that meet the theorems or remarks."
>
> In our approximation results (Theorem 5 and Corollary 1), while we consider a large class of bijective neural operators, approximators can be obtained through  finite rank neural operators (see Def. 10). Finite-rank neural operators are encountered, e.g., as FNOs [37], wavelet neural operators  [Tripura and Chakraborty, Wavelet Neural Operator for solving parametric partial differential equations in computational mechanics problems,  Comp. Meth. Appl. Mech. 2023], and Laplace neural operators [Chen et al, arXiv:2302.08166v2, 2023].
>
> The neural operators,$$F:u\to \sigma\circ (u+T_2(G(T_1u))),$$studied in our paper include neural operators that are close to those introduced in Kovachki-Lanthaler-Mishra (KLM) [28, 31]. This is discussed in the comments for all reviewers.
>
> 3. "A more detailed discussion on the practical impact of this work, accompanied by examples, would be beneficial for the audience."
>
> We much appreciate this suggestion. In Appendix A.1 we had given an example where we approximate the solution operator $x(t)\to u(t)$ of the nonlinear elliptic equation $$\partial_t^2 u(t)-g(u(t))=x(t),\quad t\in \Omega=(0,1),$$with $u=0$ on$\partial \Omega,$using a discretization that is based on Finite Element Method.In the case when $g$ is a convex function and the source term $x(t)$ is represented in the form $$x(t)=\frac {d^2}{dt^2}h(t),$$ the map $F:h\to u$ is a diffeomorphism in the Sobolev space $H^2$ with the Dirichlet boundary values $u(0)=0$ and $u(1)=0$. The approximation $F\to F_V$ can be obtained by Galerkin method. We will expand upon this example in the final version of the manuscript.
>
> 4. "Neural operators are typically defined between Banach spaces. Why does your theory focus on maps between Hilbert spaces instead?"
>
> Via our general framework, we found that strong monotonicity is one of the key ingredients to obtain a "positive" result, that is, preserving invariant discretization. Strong monotonicity is defined by using inner products, which is why we have focused on Hilbert spaces.
>
> However, the no-go theorem which states that diffeomorphisms of Hilbert spaces cannot be continuously approximated by finite dimensional diffeomorphisms implies directly that the same "negative" result holds for general Banach spaces.
>
> The main challenge for using general Banach spaces for the "positive" result is that a map $P_Y : X \to Y$, that maps a point to the closest points in the subspace $Y \subset X$, may be set-valued, that is, there may be several nearest points. Nonetheless, several of our results can be generalized to uniformly convex Banach spaces, $X$. For these,
> $$\|x\| = \|y\| = 1\hbox{ and }x\not = y\quad \implies  \|(x+y)/2\|<1.$$In such a space, for a closed subspace $Y\subset X$ and $x \in X$, there is a unique closest point $y\in Y$ to $x$. (In fact, uniformly convex Banach spaces are strictly convex Banach spaces where the above inequality  is given in a quantitative form). This makes, e.g., the linear discretization $F \to F_V = P_V F|_V$ well defined. We will include a detailed discussion in the revision on generalizations to  strictly convex spaces.
>
> 5. "Comment: The work in [1] might have been relevant to cite as well.
>
> We thank the reviewer for bringing this paper to our attention. We will add [1] to our references.
>
> 6. "The main neural operator paper [2] develops theoretical results on the universal approximation theory of neural operators. How do your results relate to the ones in that paper?"
>
> We agree with the reviewer that this is an important point. The approximation result in [2] is the universality of neural operators, i.e., to approximate any continuous map by a neural operator. Diffeomorphisms are contained in this result, which holds in the function space setting. However,  even though a general diffeomorphism, $F :\ X \to X$, can be approximated by a neural operator, $F^{NO} :\ X \to X$, and the neural operator can be approximated by a finite dimensional operator, $F^{NO}_V :\ V  \to V$, the proof of the no-go theorem implies that either the approximating infinite dimensional neural operators $F^{NO} : X \to X$ are not diffeomorphisms or that the approximation of neural operators  by finite dimensional operators, that is, operation $F^{NO}\to F^{NO}_V$, is not continuous. Note that the present universal approximation results for neural operators have mainly analyzed the approximation of functions  $F : X \to X$ by neural operators in norms of the spaces $C(K)$, where $K\subset X$ is compact, but not in the $C^1$-norms.
>
> We will add this discussion to the Introduction.

---

> > ### Comment · Reviewer_mJde · 2024-08-13
> >
> > Thank you for your detailed response. As I mentioned in my initial review, my understanding of category theory is somewhat limited. My feedback has mainly focused on the presentation and potential impact of the results rather than an in-depth validation of the theoretical content.  I am not in a position to increase my score.

---

### Author Rebuttal · Authors · 2024-08-06

We thank the reviewers for their valuable comments and detailed questions. We will provide replies to the individual reviewers below, but first would like to make some general statements addressing a few issues raised by all the reviewers.

Common questions: Practical impact/examples of this work?

A practical implication of our result is the description of  bi-Lipschitz neural operators as a composition of discretization invariant layers of invertible finite dimensional neural operators (i.e. neural networks). Such neural operators are useful in generative models where a probability distribution $\mu_0$ supported on a given model manifold $M_0$ is pushed forward by a map $F_\theta$ to a distribution that one would like to be close to an empirical target distribution $\mu_{data}$ supported on some submanifold $M_{data}$ of the Hilbert space $X$. (Here $\mu_{data}$ and $M_{data}$ are unknown and $\theta$ are optimized with samples from $\mu_{data}$). Suppose that we know a priori the topology of the data manifold $M_{data}$ and there is a diffeomorphism $f_0:M_0\to M_{data}.$ As all smooth finite dimensional submanifolds of a Hilbert space are close to some finite dimensional subspace $V$, one can start by assuming that there exists an embedding $f_1:M_0\to P_V(M_{data})$ that is close to $f_0$, where $P_V$ is a finite dimensional orthoprojection onto $V$. By considering the model manifold $M_0$ as a subset of $V$, we can extend the embedding $f_1:M_0\to V$ to a diffeomorphism $F_0:V\to V$. This can be done when the dimension of $V$ is sufficiently large [Puthawala et al., ICML 2022].

Furthermore, $F_0$ can be extended to a diffeomorphism $$F_{ext}=F_0\times Id_{V^\perp}:X\to X,$$where $F_{ext}$ maps $x=v+w\in V\oplus V^\perp$ to$$F_{ext}(v+w)=F_0(v)+w.$$The map $F_{ext}$ can be written as$$F_{ext}=Id+P_V\circ G\circ P_V, $$where $P_V$ is a compact linear operator and $G=F_{ext}-Id$. By definition the map $F_{ext}$ is a neural operator diffeomorphism. Thus, diffeomorphic neural operators can be used to obtain generative models. As the finite dimensional subspace $V$ is not a priori known, and its dimension depends on the accuracy required for the generative model, it is natural to consider infinite dimensional neural operators $F:X\to X$ and study their approximation properties.

In our paper we show, in Theorem 3, that strongly monotone neural operators can be approximated continuously by finite dimensional neural operators that are diffeomorphisim (so, invertible). In Theorem 4 we show that any bi-Lipschitz neural operator (not necessarily strongly monotone) can locally be represented as a composition of strongly monotone neural operator layers. This implies that bi-Lipschitz neural operators can be approximated by a composition of invertible, finite dimensional neural networks in a continuous way. This makes invertible neural operators a class that behaves well in finite dimensional approximations. Our results can also be summarized by stating that neural operators conditionally serve as a class of diffeomorphisms of function spaces that are simple enough for well-working approximations but still sufficiently expressive (and may model a rich variety of deformations).

The neural operators$$F:u\to \sigma\circ (u+T_2(G(T_1u)))$$we study include the neural operators that are close to those introduced in Kovachki-Lanthaler-Mishra (KLM) [28, 31].  We have assumed that operators $T_1$ and $T_2$ are compact linear operators; in several cases these can be chosen to be identity embeddings that are maps between different function spaces so that these embeddings are compact.

Consider a KLM neural operator $F:X\to X$ of the form$$F:u\to \sigma\circ (u+S_2(H(S_1u))),$$where $X=H^m(D)$ and $D\subset \mathbb R^d$ is a bounded set. Moreover, let  $Y = C(\overline D)$ and $Z = C^{m+1}(\overline D)$, where $m>d/2$. Let $h:Y \to X$ be a nonlinear (integral) operator,$$H(u)(x)=\int_D k_\theta(x,y,u(y))u(y)dy,$$where $k_\theta$ is a kernel given by a neural network with sufficiently smooth activation functions $\sigma_j$ of the form$$k_\theta(x,y,t)=\sum_{j=1}^{J} c_{j}(x,y,\theta)\sigma_j(a_{j}(x,y,\theta)t+b_j(x,y,\theta)),$$and $S_1 : X \to Y$ and $S_2  : Z \to X$ the identity embedding operators mapping between function spaces,$$S_j(u)=u.$$Thus,$$F(u)(x)=\sigma(u(x)+\int_{D}k_\theta(x,y,u(y))u(y)dy).$$The Hilbert spaces $X,Y$, and $Z$ are isomorphic and by writing, e.g.,$$S_2\circ H=T_2\circ G,$$where$$T_2=S_2\circ J_Z^{-1},\ G=J_Z\circ h,$$where $J_Z : Z \to X$ is an isomorphism, we can write $F$in the form$$F(u)=\sigma\circ (u+T_2(G(T_1u))),$$in which $T_1:X\to X$ and $T_2:X\to X$ are compact linear operators and $G:X\to X$ a continuous nonlinear operator. In this way, the  KLM-operator $F$ can be written in the form  studied in our paper.

Furthermore, by choosing $k_{\theta}(x,y,u(y)) = k_{\theta}(x - y)$ and $D = \mathbb{T}^d$ as the convolutional kernel and the torus, the map $F$ takes the form of an FNO [37].

---

### Decision · Program_Chairs · 2024-09-25

**Decision:**

Accept (poster)

**Comment:**

The reviewers find this work studying an important problem and encourage the authors to incorporate the suggestions and developments during the review process.